# ADVECTIVE DIFFUSION TRANSFORMERS FOR TOPOLOGICAL GENERALIZATION IN GRAPH LEARNING

## ABSTRACT

Graph diffusion equations are intimately related to graph neural networks (GNNs) and have recently attracted attention as a principled framework for analyzing GNN dynamics, formalizing their expressive power, and justifying architectural choices. One key open questions in graph learning is the generalization capabilities of GNNs. A major limitation of current approaches hinges on the assumption that the graph topologies in the training and test sets come from the same distribution. In this paper, we make steps towards understanding the generalization of GNNs by exploring how graph diffusion equations extrapolate and generalize in the presence of varying graph topologies. We first show deficiencies in the generalization capability of existing models built upon local diffusion on graphs, stemming from the exponential sensitivity to topology variation. Our subsequent analysis reveals the promise of non-local diffusion, which advocates for feature propagation over fully-connected latent graphs, under the assumption of a specific data-generating condition. In addition to these findings, we propose a novel graph encoder backbone, Advective Diffusion Transformer (ADiT), inspired by advective graph diffusion equations that have a closed-form solution backed up with theoretical guarantees of desired generalization under topological distribution shifts. The new model, functioning as a versatile graph Transformer, demonstrates superior performance across a wide range of graph learning tasks. Source codes will be made publicly available.

## 1 INTRODUCTION

Learning representations for non-Euclidean data is essential for geometric deep learning. Graph-structured data in particular has attracted increasing attention, as graphs are a very popular mathematical abstraction for systems of relations and interactions that can be applied from microscopic scales (e.g. molecules) to macroscopic ones (social networks). The most common framework for learning on graphs is graph neural networks (GNNs), which operate by propagating information between adjacent nodes of the graph networks (Scarselli et al., 2008; Gilmer et al., 2017; Kipf & Welling, 2017). GNNs are intimately related to graph diffusion equations (Atwood & Towsley, 2016; Klicpera et al., 2019; Chamberlain et al., 2021a) and can be seen as discretized versions thereof. Considering GNNs as diffusion equations offers powerful tools from the domain of partial differential equations (PDEs) allowing to study the expressive power (Bodnar et al., 2022), behaviors such as over-smoothing (Rusch et al., 2023; Di Giovanni et al., 2022) and over-squashing (Topping et al., 2022), the settings of missing features (Rossi et al., 2022), and guide architectural choices (Di Giovanni et al., 2022).

While significant efforts have been devoted to understanding the expressive power of GNNs and similar architectures for graph learning, the generalization capabilities of such methods are largely an open question. In many important real-world settings, the training and testing graph topologies can be generated from different distributions (a phenomenon referred to as *"topological shift"*) (Koh et al., 2021; Hu et al., 2021; Bazhenov et al., 2023; Zhang et al., 2023).

Generalization to testing data with new unseen topological patterns can be highly challenging when training observations are insufficient. One of the established principles by prior works resorts to the invariant underlying mechanism (Rojas-Carulla et al., 2018; Arjovsky et al., 2019; Schölkopf et al., 2021) that governs the shared data-generating process and enables generalization across environments. However, unlike in Euclidean space, in the case of graphs, the invariant topological features can be more abstract and complex, making it hard to come up with a single model to resolve the challenge.

**Contributions** We explore how graph diffusion equations (and derived GNN architectures) generalize in the presence of topological shifts. We show that current models relying on local graph diffusion suffer from undesirable sensitivity to variations in graph structure, making it difficult to achieve stable and reliable predictions and potentially tampering generalization. Extending the diffusion operators to latent fully-connected graphs in principle allows ideal generalization if the ground-truth labels are independent of the observed graphs in data generation, which is however often violated in practice.

To overcome this problem, we introduce a novel method for learning graph representations based on *advective diffusion* equations. We connect advective diffusion with a Transformer-like architecture particularly designed for the challenging topological generalization: the non-local diffusion term (instantiated as global attention) aims to capture invariant latent interactions that are insensitive to the observed graphs; the advection term (instantiated as local message passing) accommodates the observed topological patterns specific to environments. We prove that the closed-form solution of this new diffusion system possesses the capability to control the rate of change in node representations w.r.t. topological variations at arbitrary orders. This further produces a guarantee of the desired level of generalization under topological shifts.

For efficiently calculating the solution of the diffusion equation, we use the numerical scheme based on the Padé-Chebyshev theory (Golub & Van Loan, 1989). Experiments show that our model, which we call *Advective Diffusion Transformer (ADiT)*, offers superior generalization across a broad spectrum of graph ML tasks in diverse domains, including social and citation networks, molecular screening, and protein interactions.

## 2 BACKGROUND AND PRELIMINARIES

As building blocks of our methodology, we first recaptulate diffusion equations on manifolds (Freidlin & Wentzell, 1993; Medvedev, 2014) and its established connection with graph representations.

**Diffusion on Riemannian manifolds.** Let $\Omega$ denote an abstract domain, which we assume here to be a Riemannian manifold (Eells & Sampson, 1964). A key feature distinguishing an $n$-dimensional Riemannian manifold from a Euclidean space is the fact that it is only *locally* Euclidean, in the sense that at every point $u \in \Omega$ one can construct $n$-dimensional Euclidean *tangent space* $T_u\Omega \cong \mathbb{R}^n$ that locally models the structure of $\Omega$. The collection of such spaces (referred to as the *tangent bundle* and denoted by $T\Omega$) is further equipped with a smoothly-varying inner product (*Riemannian metric*).

Now consider some quantity (e.g., temperature) as a function of the form $q : \Omega \to \mathbb{R}$, which we refer to as a *scalar field*. Similarly, we can define a *(tangent) vector field* $Q : \Omega \to T\Omega$, associating to every point $u$ on a manifold a tangent vector $Q(u) \in T_u\Omega$, which can be thought of as a local infinitesimal displacement. We use $\mathcal{Q}(\Omega)$ and $\mathcal{Q}(T\Omega)$ to denote the functional spaces of scalar and vector fields, respectively. The *gradient* operator $\nabla : \mathcal{Q}(\Omega) \to \mathcal{Q}(T\Omega)$ takes scalar fields into vector fields representing the local direction of the steepest change of the field. The *divergence* operator is the adjoint of the gradient and maps in the opposite direction, $\nabla^* : \mathcal{Q}(T\Omega) \to \mathcal{Q}(\Omega)$.

A manifold diffusion process models the evolution of a quantity (e.g., temperature or chemical concentration) due to its difference across spatial locations on $\Omega$. Denoting by $q(u, t) : \Omega \times [0, \infty) \to \mathbb{R}$ the quantity over time $t$, the process is described by a PDE (*diffusion equation*) (Romeny, 2013):

$$\frac{\partial q(u, t)}{\partial t} = \nabla^* \left( S(u, t) \odot \nabla q(u, t) \right), \ \ t \geq 0, u \in \Omega \ \ \text{with initial conditions} \ \ q(u, 0) = q_0(u), \ \ (1)$$

and possibly additional boundary conditions if $\Omega$ has a boundary. $S$ denotes the *diffusivity* of the domain. It is typical to distinguish between an *isotropic* (location-independent diffusivity), *non-homogeneous* (location-dependent diffusivity $S = s(u) \in \mathbb{R}$), and *anisotropic* (location- and direction-dependent $S(u) \in \mathbb{R}^{n \times n}$) settings. In the cases studied below, we will assume the dependence of the diffusivity on the location is via a function of the quantity itself, i.e., $S = S(q(u, t))$.

**Diffusion on Graphs.** Recent works leverage diffusion equations as a foundation principle for learning graph representations (Chamberlain et al., 2021a;b; Thorpe et al., 2022; Bodnar et al., 2022; Choi et al., 2023; Rusch et al., 2023), employing analogies between calculus on manifolds and graphs. Let $\mathcal{G} = (\mathcal{V}, \mathcal{E})$ be a graph with nodes $\mathcal{V}$ and edges $\mathcal{E}$, represented by the $|\mathcal{V}| \times |\mathcal{V}|$ *adjacency matrix* $\mathbf{A}$. Let $\mathbf{X} = [\mathbf{x}_u]_{u \in \mathcal{V}}$ denote a $|\mathcal{V}| \times D$ matrix of node features, analogous to scalar fields on manifolds. The graph gradient $(\nabla \mathbf{X})_{uv} = \mathbf{x}_v - \mathbf{x}_u$ defines edge features for $(u, v) \in \mathcal{E}$, analogous to a vector field on a manifold. Similarly, the graph divergence of edge features $\mathbf{E} = [\mathbf{e}_{uv}]_{(u,v) \in \mathcal{E}}$, defined as the adjoint $(\nabla^* \mathbf{E})_u = \sum_{v:(u,v) \in \mathcal{E}} \mathbf{e}_{uv}$, produces node features.

Diffusion-based approaches replace discrete GNN layers with continuous time-evolving node embeddings $\mathbf{Z}(t) = [\mathbf{z}_u(t)]$, where $\mathbf{z}_u(t) : [0, \infty) \to \mathbb{R}^D$ is driven by the graph diffusion equation,

$$\partial \mathbf{Z}(t)/\partial t = \nabla^* \left( \mathbf{S}(\mathbf{Z}(t), t; \mathbf{A}) \odot \nabla \mathbf{Z}(t) \right), \ \ t \geq 0, \ \ \text{with initial conditions } \mathbf{Z}(0) = \phi_{enc}(\mathbf{X}), \ \ (2)$$

where $\phi_{enc}$ is a node-wise MLP encoder and w.l.o.g., the diffusivity $\mathbf{S}(\mathbf{Z}(t), t; \mathbf{A})$ over the graph can be defined as a $|\mathcal{V}| \times |\mathcal{V}|$ matrix-valued function dependent on $\mathbf{A}$, which measures the rate of information flows between node pairs. With the graph gradient and divergence, Eqn. 2 becomes

$$\partial \mathbf{Z}(t)/\partial t = (\mathbf{C}(\mathbf{Z}(t), t; \mathbf{A}) - \mathbf{I})\mathbf{Z}(t), \ \ 0 \leq t \leq T, \ \ \text{with initial conditions } \mathbf{Z}(0) = \phi_{enc}(\mathbf{X}), \ \ (3)$$

where $\mathbf{C}(\mathbf{Z}(t), t; \mathbf{A})$ is a $|\mathcal{V}| \times |\mathcal{V}|$ coupling matrix associated with the diffusivity. Eqn. 3 yields a dynamics from $t = 0$ to an arbitrary given stopping time $T$, where the latter gives node representations for prediction, e.g., $\hat{\mathbf{Y}} = \phi_{dec}(\mathbf{Z}(T))$. The coupling matrix determines the interactions between different nodes in the graph, and its common instantiations include the normalized adjacency (non-parametric) and learnable attention matrix (parametric), in which cases the finite-difference numerical iterations for solving Eqn. 3 correspond to the discrete propagation layers of common GNNs (Chamberlain et al., 2021a) and Transformers (Wu et al., 2023) (see Appendix A for details).

It is typical to tacitly make a closed-world assumption, i.e., the graph topologies of training and testing data are generated from the same distribution. The challenge of generalization arises when the testing graph topology is different from the training one. In such an open-world regime, it still remains unexplored how graph diffusion equations extrapolate and generalize to new unseen structures.

## 3    CAN GRAPH DIFFUSION GENERALIZE?

As a prerequisite for analyzing the generalization behaviors of graph diffusion models, we need to characterize how topological shifts happen in nature. In general sense, extrapolation is impossible without any exposure to the new data or prior knowledge about the data-generating mechanism. In our work, we assume testing data is strictly unknown during training, in which case structural assumptions become necessary for authorizing generalization.

### 3.1    PROBLEM FORMULATION: GRAPH DATA GENERATION

We present the underlying data-generating mechanism of graph data in Fig. 1, inspired by the graph limits (Lovász & Szegedy, 2006; Medvedev, 2014) and random graph models (Snijders & Nowicki, 1997). In graph theory, the topology of a graph $\mathcal{G} = (\mathcal{V}, \mathcal{E})$ can be assumed to be generated by a *graphon* (or continuous graph limit), a random symmetric measurable function $W : [0, 1]^2 \to [0, 1]$, which is an unobserved latent variable. In our work, we generalize this data-generating mechanism to include alongside graph adjacency also node features and labels, as follows:

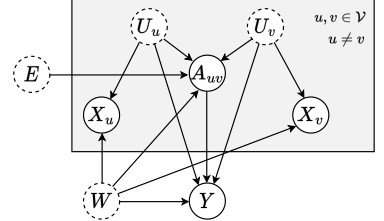

Figure 1: The data-generating mechanism with topological shifts caused by environment $E$. The solid (resp. dashed) nodes represents observed (resp. latent) random variables.

**i)** Each node $u \in \mathcal{V}$ has a latent i.i.d. variable $U_u \sim U[0, 1]$. The *node features* are a random variable $X = [X_u]$ generated from each $U_u$ through a certain node-wise function $X_u = g(U_u; W)$. We denote by matrix $\mathbf{X}$ a particular realization of the random variable $X$.

**ii)** Similarly, the *graph adjacency* $A = [A_{uv}]$ is a random variable generated through a pairwise function $A_{uv} = h(U_u, U_v; W, E)$ additionally dependent on the *environment* $E$. The change of $E$ happens when it transfers from training to testing, resulting in a different distribution of $A$. We denote by $\mathbf{A}$ a particular realization of the adjacency matrix.

**iii)** The *label $Y$* can be specified in certain forms. In graph-level tasks (as we assume in below), $Y$ is generated by a function over sets, $Y = r(\{U_{v \in \mathcal{V}}\}, A; W)$. Denote by $\mathbf{Y}$ a realization of $Y$.

The above process formalizes the data-generating mechanism behind various data of inter-dependent nature. It boils down to finding parameters $\theta$ of a parametric function $\Gamma_\theta(\mathbf{A}, \mathbf{X})$ that establishes the predictive mapping from observed node features $\mathbf{X}$ and graph adjacency $\mathbf{A}$ to the label $\mathbf{Y}$. $\Gamma_\theta$ is typically implemented as a GNN, which is expected to possess sufficient *expressive power* (in the sense that $\exists \theta$ such that $\Gamma_\theta(\mathbf{A}, \mathbf{X}) \approx \mathbf{Y}$) as well as *generalization capability* under topological distribution shift (i.e., when the observed graph topology varies from training to testing, which in our model amounts to the change in $E$). While significant attention in the literature has been devoted to the former property (Morris et al., 2019; Xu et al., 2019; Bouritsas et al., 2023; Papp et al., 2021; Balcilar et al., 2021; Bodnar et al., 2022); the latter is largely an open question.

## 3.2 Graph Diffusion under Topological Shifts

Building upon the connection between GNNs and diffusion equations, we next study the behavior of diffusion equation (i.e., Eqn. 3) under topological shifts, which will shed lights on GNN generalization. The effect of $\mathbf{A}$ on node representations (solution of the diffusion equation $\mathbf{Z}(T)$) stems from the coupling matrix $\mathbf{C}(\mathbf{Z}(t), t; \mathbf{A})$. Thereby, the output of the diffusion process can be expressed as $\mathbf{Z}(T) = f(\mathbf{Z}(0), \mathbf{A})$. We are interested in the extrapolation behavior of graph diffusion models that can be reflected by the change of $\mathbf{Z}(T)$ w.r.t. small perturbation centered at $\mathbf{A}$.

*Linear Diffusion.* We first consider the constant diffusivity setting inducing $\mathbf{C}(\mathbf{Z}(t), t; \mathbf{A}) = \mathbf{C}$. In this case, Eqn. 3 becomes a linear diffusion equation with a closed-form solution $\mathbf{Z}(t) = e^{-(\mathbf{I}-\mathbf{C})t}\mathbf{Z}(0)$. In this case, using the numerical scheme to solve the PDE would induce the discrete propagation layers akin to SGC (Wu et al., 2019), where the non-linearity in-between layers is omitted for acceleration (see more illustration on this connection in Appendix A). The following proposition shows that the variation magnitude of $\mathbf{Z}(T)$ can be significant for small change of input graphs.

**Proposition 1.** *If the coupling matrix $\mathbf{C}$ is set as the normalized adjacency $\tilde{\mathbf{A}} = \mathbf{D}^{-1}\mathbf{A}$ or $\tilde{\mathbf{A}} = \mathbf{D}^{-1/2}\mathbf{A}\mathbf{D}^{-1/2}$, where $\mathbf{D}$ denotes the diagonal degree matrix of $\mathbf{A}$, then the change of $\mathbf{Z}(T; \tilde{\mathbf{A}})$ given by Eqn. 3 w.r.t. a small perturbation $\Delta\tilde{\mathbf{A}}$ is $\|\mathbf{Z}(T; \tilde{\mathbf{A}} + \Delta\tilde{\mathbf{A}}) - \mathbf{Z}(T; \tilde{\mathbf{A}})\|_2 = \mathcal{O}(\exp(\|\Delta\tilde{\mathbf{A}}\|_2 T))$.*

The consequence of this result is that the label prediction $\hat{\mathbf{Y}} = \phi_{dec}(\mathbf{Z}(T; \tilde{\mathbf{A}}))$ can be highly (exponentially) sensitive to the change of the graph topology. Under the assumption of our graph generation model in which the graph adjacency is a realization of a random variable $A = h(U_u, U_v; W, E)$ dependent on a varying environment $E$, this may result in poor generalization.[1] Proposition 1 can be extended to the multi-layer model comprised of multiple piece-wise diffusion dynamics with feature transformations (e.g., neural networks) in-between layers (see Appendix B.2).

*Non-Linear Diffusion.* In a more general setting, the diffusivity can be time-dependent. The analogy in GNN architectures e.g. GAT (Velickovic et al., 2018) is layer-wise propagation that can aggregate neighbored nodes' signals with adaptive strengths across edges. Consider the time-dependent case used in (Chamberlain et al., 2021a), where $\mathbf{C}(t)$ depends on $\mathbf{Z}(t)$ throughout the diffusion process:

$$\mathbf{C}(\mathbf{Z}(t); \mathbf{A}) = [c_{uv}(t)]_{u,v \in \mathcal{V}}, \quad c_{uv}(t) = \mathbb{I}[(u,v) \in \mathcal{E}] \cdot \frac{\eta(\mathbf{z}_u(t), \mathbf{z}_v(t))}{\sum_{w,(u,w)\in\mathcal{E}} \eta(\mathbf{z}_u(t), \mathbf{z}_w(t))}, \quad (4)$$

where $\eta : \mathbb{R}^d \times \mathbb{R}^d \to \mathbb{R}$ denotes a pairwise function ("attention"). While such a non-linear diffusion equation has no closed-form solution anymore, we can generalize our previous result as follows:

**Proposition 2.** *For arbitrary time limit $T$ and bounded function $\eta$, the change of $\mathbf{Z}(T)$ by the diffusion model Eqn. 3 with $\mathbf{C}(\mathbf{Z}(t); \mathbf{A})$ by Eqn. 4 w.r.t. a small perturbation $\Delta\mathbf{A}$ is $\mathcal{O}(\exp(\|\Delta\mathbf{A}\|_2 T))$.*

The analysis so far suggests the common limitation of local graph diffusion equations with different instantiations, i.e., the sensitivity of the output states w.r.t. the change of graph topology. This implies the potential failure of such a model class for the challenge of generalization where the graph topology varies from training to testing. Moreover, the analysis enlightens that the crux of the matter lies in the diffusion operators which determine the effect of graph structures throughout the diffusion process.

## 3.3 Non-Local Graph Diffusion and Generalization with Conditions

We proceed to extend our discussion to another class of neural diffusion models that resort to non-local diffusion operators allowing instantaneous information flows among arbitrary locations (Chasseigne et al., 2006). In the context of learning on graphs, the non-local diffusion can be seen as generalizing the feature propagation to a *complete* or fully-connected (latent) graph (Wu et al., 2023), in contrast with common GNNs that allow message passing only between neighboring nodes. Formally speaking, we can define the gradient and divergence operators on a complete graph: $(\nabla\mathbf{X})_{uv} = \mathbf{x}_v - \mathbf{x}_u$ $(u, v \in \mathcal{V})$ and $(\nabla^*\mathbf{E})_u = \sum_{v \in \mathcal{V}} \mathbf{e}_{uv}$ $(u \in \mathcal{V})$. The corresponding diffusion equation still exhibits the form of Eqn. 3. Nevertheless, unlike the models studied in Sec. 3.2 assuming that $\mathbf{C}(t)$ only has non-zeros entries $c_{uv}(t) \neq 0$ for neighboring node pairs $(u, v) \in \mathcal{E}$, the non-local diffusion model allows non-zero $c_{uv}(t)$ for arbitrary $(u, v)$'s to accommodate the all-pair information flows. For example, the coupling matrix can be instantiated as the global attention $\mathbf{C}(\mathbf{Z}(t)) = [c_{uv}(t)]_{u,v \in \mathcal{V}}$ with

---

[1]The influence of topology variation is inherently associated with $h$. For example, if one considers $h$ as the stochastic block model (Snijders & Nowicki, 1997), then the change of $E$ may lead to generated graph data with different edge probabilities. In the case of real-world data with intricate topological patterns, the functional forms of $h$ can be more complex, consequently inducing different types of topological shifts.

$c_{uv}(t) = \frac{\eta(\mathbf{z}_u(t), \mathbf{z}_v(t))}{\sum_{w \in \mathcal{V}} \eta(\mathbf{z}_u(t), \mathbf{z}_w(t))}$, in which case the finite-difference iteration of the non-local diffusion equation corresponds to a Transformer layer (Vaswani et al., 2017) (see details in Appendix A).

The non-local diffusion model essentially learns latent interaction graphs among nodes from input data and is agnostic to observed graph. For the predictive function $\Gamma_\theta$ built by the diffusion equation along with the encoder $\phi_{enc}$ and decoder $\phi_{dec}$, we can theoretically guarantee topological generalization when $Y$ is conditionally independent from $A$ within the data-generating process in Sec. 3.1.

**Proposition 3.** *Suppose the label $Y$ is conditionally independent from $A$ with given $\{U_u\}_{u \in \mathcal{V}}$ in the data generation hypothesis of Sec. 3.1, then for non-local diffusion model $\Gamma_\theta$ minimizing the empirical risk $\mathcal{R}_{emp}(\Gamma_\theta; E_{tr}) = \frac{1}{N_{tr}} \sum_i^{N_{tr}} l(\Gamma_\theta(\mathbf{X}^{(i)}, \mathbf{A}^{(i)}), \mathbf{Y}^{(i)})$ over training data $\{(\mathbf{X}^{(i)}, \mathbf{A}^{(i)}, \mathbf{Y}^{(i)})\}$ generated from $p(X, A, Y | E = E_{tr})$, it holds with confidence $1 - \delta$ for the bounded generalization error on unseen data $(\mathbf{X}', \mathbf{A}', \mathbf{Y}')$ from a new environment $E_{te} \neq E_{tr}: \mathcal{R}(\Gamma_\theta; E_{te}) \triangleq$*

$$\mathbb{E}_{(\mathbf{X}', \mathbf{A}', \mathbf{Y}') \sim p(X, A, Y | E = E_{te})}[l(\Gamma_\theta(\mathbf{X}', \mathbf{A}'), \mathbf{Y}')] \leq \mathcal{R}_{emp}(\Gamma_\theta; E_{tr}) + \mathcal{D}_1(\Gamma, N_{tr}), \quad (5)$$

*where $\mathcal{D}_1(\Gamma, N_{tr}) = 2\mathcal{H}(\Gamma) + \mathcal{O}\left(\sqrt{(1/N_{tr})\log(1/\delta)}\right)$, $\mathcal{H}(\Gamma)$ denotes the Rademacher complexity of the function class of $\Gamma$, $N_{tr}$ is the size of the training set, and $l$ denotes any bounded loss function.*

The conditional independence between $Y$ and $A$, however, can be violated in many situations where labels strongly correlate with observed graph structures. In such cases, the non-local diffusion alone, discarding any observed structural information, could be insufficient for generalization.

## 4 GRAPH ADVECTIVE DIFFUSION FOR TOPOLOGICAL GENERALIZATION

The preceding analysis reveals that the obstacles for graph diffusion models to achieving generalization arise from the non-fulfillment of two critical criteria: i) the diffusion process is capable of learning useful topological patterns; ii) the node representations are insensitive to variation of graph structures. While balancing these two objectives can be challenging due to the inherent trade-off, we present a novel graph diffusion model in this section that offers a provable level of generalization. The new model is inspired by a different class of diffusion equations, *advective diffusion*.

### 4.1 MODEL FORMULATION: GRAPH ADVECTIVE DIFFUSION

**Advective Diffusion Equations.** We first introduce the classic advective diffusion commonly used for characterizing physical systems with convoluted quantity transfers, where the term *advection* (or *convection*) refers to the evolution caused by the movement of the diffused quantity (Chandrasekhar, 1943). Consider the abstract domain $\Omega$ of our interest defined in Sec. 2, and assume $V(u, t) \in T_u \Omega$ (a vector field in $\Omega$) to denote the velocity of the particle at location $u$ and time $t$. The advective diffusion of the physical quantity $q$ on $\Omega$ is governed by the PDE as (Leveque, 1992)

$$\frac{\partial q(u, t)}{\partial t} = \underbrace{\nabla^* (S(u, t) \odot \nabla q(u, t))}_{\text{diffusion}} + \beta \underbrace{\nabla^* (V(u, t) \cdot q(u, t))}_{\text{advection}}, \quad t \geq 0, u \in \Omega; \quad q(u, 0) = q_0(u),$$

$$(6)$$

where $\beta \geq 0$ is a weight. For example, if we consider $q(u, t)$ as the water salinity in a river, then Eqn. 6 describes the temporal evolution of salinity at each location that equals to the spatial transfers of both diffusion process (caused by the concentration difference of salt and $S$ reflects the molecular diffusivity in the water) and advection process (caused by the movement of the water and $V$ characterizes the flowing directions).

Similarly, on a graph $\mathcal{G} = (\mathcal{V}, \mathcal{E})$, we can define the velocity for each node $u$ as a $|\mathcal{V}|$-dimensional vector-valued function $\mathbf{V}(t) = [\mathbf{v}_u(t)]$. Then, we have $(\nabla^*(\mathbf{V}(t) \cdot \mathbf{Z}(t)))_u = \sum_{v \in \mathcal{V}} v_{uv}(t)\mathbf{z}_v(t)$, giving rise to the graph advective diffusion equation:

$$\frac{\partial \mathbf{Z}(t)}{\partial t} = [\mathbf{C}(\mathbf{Z}(t), t) + \beta\mathbf{V}(t) - \mathbf{I}]\mathbf{Z}(t), \quad 0 \leq t \leq T. \quad (7)$$

**Graph Advective Diffusion.** We proceed to discuss how to properly define the coupling matrix $\mathbf{C}$ and the velocity $\mathbf{V}$ to ensure that advective diffusion equations are stable under topological shifts. Our inspiration stems from the recent research line in the pursuit of invariance in data generation (Rojas-Carulla et al., 2018; Arjovsky et al., 2019; Schölkopf et al., 2021), where the

principle of (out-of-distribution) generalization lies in enforcing proper inductive bias that guides the model to capture the invariant underlying mechanism shared across environments. Different from natural data in Euclidean space (e.g., images), the invariant topological patterns in graphs can be much more difficult to capture given their abstract and versatile characteristics. We next generalize the invariance principle as an important inductive bias integrated into the advective diffusion for generalization purpose (with illustration in Fig. 2).

*Non-local diffusion as global attention.* The diffusion process led by the concentration gradient acts as an internal driving force, where the diffusivity keeps invariant across environments (e.g., the molecular diffusivity stays constant in different rivers). This resonates with the environment-invariant latent interactions among nodes, determined by the underlying data manifold, that induce all-pair information flows over a complete graph. We thus follow Sec. 3.3 and instantiate $\mathbf{C}$ as a global attention that computes the similarities between arbitrary node pairs.

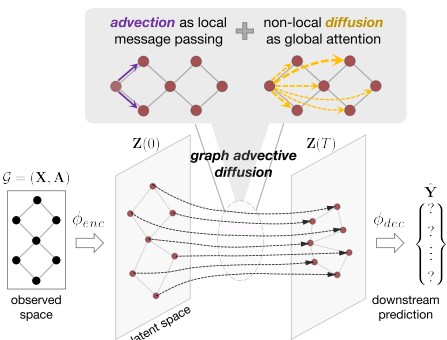

Figure 2: Illustration of the proposed model.

*Advection as local message passing.* The advection process driven by the directional movement belongs to an external force, with the velocity depending on contexts (e.g., different rivers). This is analogous to the environment-sensitive graph topology that is informative for prediction in specific environments. We instantiate the velocity as the normalized adjacency $\mathbf{V} = \tilde{\mathbf{A}}$ that reflects graph structures. With the above definitions, our graph advective diffusion model can be formulated as:

$$\frac{\partial \mathbf{Z}(t)}{\partial t} = \left[\mathbf{C} + \beta\tilde{\mathbf{A}} - \mathbf{I}\right]\mathbf{Z}(t), \quad 0 \leq t \leq T \quad \text{with initial conditions} \quad \mathbf{Z}(0) = \phi_{enc}(\mathbf{X}),$$

$$\text{where} \quad \mathbf{C} = [c_{uv}]_{u,v\in\mathcal{V}}, \quad c_{uv} = \frac{\eta(\mathbf{z}_u(0), \mathbf{z}_v(0))}{\sum_{w\in\mathcal{V}}\eta(\mathbf{z}_u(0), \mathbf{z}_w(0))}. \tag{8}$$

Here $\beta \in [0, 1]$ is a weight hyper-parameter and $\eta$ is a learnable pairwise similarity function. The two mechanisms of non-local diffusion (implemented through attention akin to Transformers) and advection (implemented like message passing neural networks) give rise to a new architecture, which we call the *Advective Diffusion Transformer*, or ADiT for short.

***Remark.*** Eqn. 8 has a closed-form solution $\mathbf{Z}(t) = e^{-(\mathbf{I}-\mathbf{C}-\beta\tilde{\mathbf{A}})t}\mathbf{Z}(0)$, and as we will show in the next subsection, it allows generalization guarantees with topological distribution shifts. A special case of $\beta = 0$ (no advection) can be used in situations where the graph structure is not useful. Moreover, one can extend Eqn. 8 to a non-linear equation with time-dependent $\mathbf{C}(\mathbf{Z}(t), t)$, in which situation the equation will have no closed-form solution and need numerical schemes for solving. Similarly to Di Giovanni et al. (2022), we found in our experiments a simple linear diffusion to be sufficient to yield promising performance. We therefore leave the study of the non-linear variant for the future.

### 4.2 HOW GRAPH ADVECTIVE DIFFUSION HANDLES TOPOLOGICAL SHIFTS

We proceed to analyze the behavior of our proposed model w.r.t. topological shifts to demonstrate its capability of generalizing to out-of-distribution (OOD) data. Our first main result is derived based on the universal approximation power of neural networks and the data generation hypothesis in Sec. 3.1.

**Theorem 1.** *For the model Eqn. 7 with $\mathbf{C}$ pre-computed by global attention over $\mathbf{Z}(0)$ and fixed velocity $\mathbf{V} = \tilde{\mathbf{A}}$, the change rate of node representations $\mathbf{Z}(T; \tilde{\mathbf{A}})$ w.r.t. a small perturbation $\Delta\tilde{\mathbf{A}}$ can be reduced to $\mathcal{O}(\psi(\|\Delta\tilde{\mathbf{A}}\|_2))$ where $\psi$ denotes an arbitrary polynomial function.*

Theorem 1 suggests that the advective diffusion model with observed structural information incorporated is capable of controlling the impact of topology variation on node representations to arbitrary rates. We can further derive the generalization error that is decomposed into the in-distribution generalization (ID) error $\mathcal{D}_1(\Gamma, N_{tr})$ and the topological distribution gap between ID and OOD data.

**Theorem 2.** *Assume $l$ and $\phi_{dec}$ are Lipschitz continous. Then for data generated with the data generation hypothesis of Sec. 3.1 from arbitrary $E_{tr}$ and $E_{te}$, we have the generalization error bound of the model $\Gamma_\theta$ with confidence $1 - \delta$:*

$$\mathcal{R}(\Gamma_\theta; E_{te}) \leq \mathcal{R}_{emp}(\Gamma_\theta; E_{tr}) + \mathcal{D}_1(\Gamma, N_{tr}) + \mathcal{D}_2(E_{tr}, E_{te}, W), \tag{9}$$

where $\mathcal{D}_2(E_{tr}, E_{te}, W) = \mathcal{O}(\mathbb{E}_{\mathbf{A} \sim p(A|E_{tr}), \mathbf{A}' \sim p(A|E_{te})}[\psi(\|\Delta\tilde{\mathbf{A}}\|_2)])$.

Theorem 2 implies that the generalization error can be controlled with the adaptive change rate yielded by the model. The model possesses provable potential for achieving a desired level of generalization with topological shifts. Furthermore, our model only requires trainable parameters for two shallow MLPs $\phi_{enc}$ and $\phi_{dec}$ and the attention network $\eta$, which is highly parameter-efficient. This helps to reduce the model complexity measured by $\mathcal{H}(\Gamma)$ that impacts $\mathcal{D}_1$ and is beneficial for generalization.

### 4.3 NUMERICAL SOLVERS FOR GRAPH ADVECTIVE DIFFUSION

We next delve into the model implementation, with a key question how to compute the closed-form solution $e^{-(\mathbf{I}-\mathbf{C}-\beta\tilde{\mathbf{A}})t}$. Direct computation of the matrix exponential through eigendecomposition is computationally intractable for large matrices. As an alternative, we explore several numerical approximation techniques based on series expansion.

**ADiT-INVERSE** uses a numerical method based on the extension of Padé-Chebyshev theory to rational fractions (Golub & Van Loan, 1989; Gallopoulos & Saad, 1992), which has shown empirical success in 3D shape analysis (Patané, 2014). The matrix exponential is approximated by solving multiple linear systems (see more details and derivations in Appendix D) and we generalize it as a flexible multi-head network where each head propagates in parallel:

$$\mathbf{Z}(T) \approx \sum_{h=1}^{H} \phi_{FC}^{(h)}(\mathbf{Z}_h), \quad \mathbf{Z}_h = \text{linsolver}(\mathbf{L}_h, \mathbf{Z}(0)), \quad \mathbf{L}_h = (1+\theta)\mathbf{I} - \mathbf{C}_h - \beta\tilde{\mathbf{A}}, \quad (10)$$

where the *linsolver* computes the matrix inverse $\mathbf{Z}_h = (\mathbf{L}_h)^{-1}\mathbf{Z}(0)$ and can be efficiently implemented via `torch.linalg.solve()` that supports automated differentiation. Each head contributes to propagation with the pre-computed attention $\mathbf{C}_h$ and node-wise transformation $\phi_{FC}^{(h)}$.

**ADiT-SERIES** approximates the matrix inverse via finite geometric series (see Appendix D for detailed derivations)

$$\mathbf{Z}(T) \approx \sum_{h=1}^{H} \phi_{FC}^{(h)}(\mathbf{Z}_h), \quad \mathbf{Z}_h = [\mathbf{Z}(0), \mathbf{P}_h\mathbf{Z}(0), \cdots, (\mathbf{P}_h)^K\mathbf{Z}(0)], \quad \mathbf{P}_h = \mathbf{C}_h + \beta\tilde{\mathbf{A}}, \quad (11)$$

for better scalability. This model resorts to aggregation of $K$-order propagation with the propagation matrix $\mathbf{P}_h$ in each head. The feed-forward of the model can be efficiently computed within linear complexity w.r.t. the number of nodes (see how we achieve this acceleration in Appendix E.1.2).

The node representations obtained by approximate solution of the diffusion equation $\mathbf{Z}(T)$ are then fed into $\phi_{dec}$ for prediction and loss computation (e.g., cross-entropy for classification or mean square loss for regression). Due to space limit, we defer details of model architectures to Appendix E.1. Moreover, in Appendix E.2 we discuss how to extend our model to accommodate edge attributes.

## 5 EXPERIMENTS

We apply our model to synthetic and real-world datasets that involve various topological distribution shifts. We consider a wide variety of graph-based downstream tasks of disparate scales and granularities. More detailed dataset information is provided in Appendix F.1. In each case, we compare with different sets of competitors that are suitable for the tasks. Details on baselines and implementation are deferred to Appendix F.2 and F.3, respectively.

### 5.1 SYNTHETIC DATASETS

We create synthetic datasets that simulate the data generation in Sec. 3.1 to validate our model. We instantiate $h$ as a stochastic block model which generates edges $A_{uv}$ according to block numbers ($b$), intra-block edge probability ($p_1$) and inter-block edge probability ($p_2$). Then we study three types of topological distribution shifts: **homophily shift** (changing $p_2$ with fixed $p_1$); **density shift** (changing $p_1$ and $p_2$); and **block shift** (varying $b$). The predictive task is node regression and we use RMSE to measure the performance. Details for dataset generation is presented in Appendix F.1.1.

Fig. 3 plots RMSE on training/validation/testing graphs in three cases. We compare our model (ADiT-INVERSE and ADiT-SERIES) with diffusion-based models analyzed in Sec. 3. The latter includes *Diff-Linear* (graph diffusion with constant $\mathbf{C}$), *Diff-MultiLayer* (the extension of *Diff-Linear* with intermediate feature transformations), *Diff-Time* (graph diffusion with time-dependent $\mathbf{C}(\mathbf{Z}(t))$)

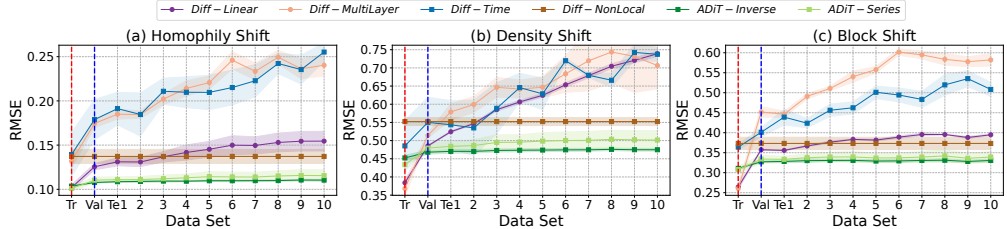

Figure 3: Results of RMSE ($\downarrow$) on synthetic datasets that simulate the topological shifts caused by the environment $E$ in Fig. 1. We consider three types of shifts w.r.t. homophily levels, edge densities, and block numbers, respectively. In each case, the validation and #1∼#10 testing sets are generated with different configurations introducing increasing distribution gaps from the training set.

and *Diff-NonLocal* (non-local diffusion with global attentive diffusivity $\mathbf{C}(\mathbf{Z}(t))$). Three local graph diffusion models exhibit clear performance degradation w.r.t. topological shifts exacerbated from #1 to #10 testing graphs, while our two models yield consistently low RMSE across environments. In contrast, the non-local diffusion model produce comparably stable performance yet inferior to our models due to its failure of utilizing the observed topological information.

Table 1: Results on `Arxiv` and `Twitch`, where we use time and spatial contexts for data splits, respectively. We report the Accuracy ($\uparrow$) for three testing sets of `Arxiv` and average ROC-AUC ($\uparrow$) for all testing graphs of `Twitch` (results for each case are reported in Appendix G.1). Top performing methods are marked as first/second/third. OOM indicates out-of-memory error.

| | Arxiv (2018) | Arxiv (2019) | Arxiv (2020) | Twitch (avg) |
|---|---|---|---|---|
| **MLP** (Rumelhart et al., 1986) | 49.91 ± 0.59 | 47.30 ± 0.63 | 46.78 ± 0.98 | 61.12 ± 0.16 |
| **GCN** (Kipf & Welling, 2017) | 50.14 ± 0.46 | 48.06 ± 1.13 | 46.46 ± 0.85 | 59.76 ± 0.34 |
| **GAT** (Velickovic et al., 2018) | 51.60 ± 0.43 | 48.60 ± 0.28 | 46.50 ± 0.21 | 59.14 ± 0.72 |
| **SGC** (Wu et al., 2019) | 51.40 ± 0.10 | 49.15 ± 0.16 | 46.94 ± 0.29 | 60.86 ± 0.13 |
| **GDC** (Klicpera et al., 2019) | 51.53 ± 0.42 | 49.02 ± 0.51 | 47.33 ± 0.60 | 61.36 ± 0.10 |
| **GRAND** (Chamberlain et al., 2021a) | 52.45 ± 0.27 | 50.18 ± 0.18 | 48.01 ± 0.24 | 61.65 ± 0.23 |
| **GraphTrans** (Wu et al., 2021) | OOM | OOM | OOM | 61.65 ± 0.23 |
| **GraphGPS** (Rampásek et al., 2022) | 51.11 ± 0.19 | 48.91 ± 0.34 | 46.46 ± 0.95 | 62.13 ± 0.34 |
| **DIFFormer** (Wu et al., 2023) | 50.45 ± 0.94 | 47.37 ± 1.58 | 44.30 ± 2.02 | 62.11 ± 0.11 |
| **ADIT-SERIES** | 53.41 ± 0.48 | 51.53 ± 0.60 | 49.64 ± 0.54 | 62.51 ± 0.07 |

## 5.2 REAL-WORLD DATASETS

We proceed to evaluate ADIT beyond the synthetic cases and experiment on real-world datasets with more complex shifts in graph topologies encountered in diverse and broad applications.

**Information Networks**. We first consider node classification on citation networks `Arxiv` (Hu et al., 2020) and social networks `Twitch` (Rozemberczki et al., 2021) with graph sizes ranging from 2K to 0.2M, where we use the scalable version ADIT-SERIES. To introduce topological shifts, we partition the data according to publication years and geographic information for `Arxiv` and `Twitch`, respectively. The predictive task is node classification, and we follow the common practice comparing Accuracy (resp. ROC-AUC) for `Arxiv` (resp. `Twitch`). We compare with three types of state-of-the-art baselines: (i) **classical GNNs** (*GCN* (Kipf & Welling, 2017), *GAT* (Velickovic et al., 2018) and *SGC* (Wu et al., 2019)); (ii) **diffusion-based GNNs** (*GDC* (Klicpera et al., 2019) and *GRAND* (Chamberlain et al., 2021a)), and (iii) **graph Transformers** (*GraphTrans* (Wu et al., 2021), *GraphGPS* (Rampásek et al., 2022), and the diffusion-based *DIFFormer* (Wu et al., 2023)). Appendix F.2 presents detailed descriptions for these models. Table 1 reports the results, showing that our model offers significantly superior generalization for node classification.

**Molecular Property Prediction**. We next study graph classification for predicting molecular properties on `OGB-BACE` and `OGB-SIDER`. We follow the scaffold-based splits by Hu et al. (2020), which guarantee structural diversity across training and test sets and provide a realistic estimate of model generalization in prospective experimental settings (Yang et al., 2019). The performance is measured by ROC-AUC. Table 2 reports the results, showing that our model outperforms classical GNNs and powerful graph Transformers[2] that use the same input data and training loss.

**Protein Interactions**. We then test on protein-protein interactions of yeast cells (Fu & He, 2022). Each node denotes a protein with a time-aware gene expression value and the edges indicate co-expressed protein pairs at each time. The dataset consists of 12 dynamic networks each of which is

---

[2]Note that our comparison focuses on generic GNN architectures, rather than specialized methods that are tailored for chemical problems and additionally leverage domain knowledge such as structural motifs.

Table 2: ROC-AUC ($\uparrow$) on two molecule datasets `OGB-BACE` and `OGB-SIDER` with scaffold splits for training/validation/testing, where the task is to predict molecular graph properties.

| | OGB-BACE | | | OGB-SIDER | | |
|---|---|---|---|---|---|---|
| | Train | Valid | Test | Train | Valid | Test |
| **MLP** | $67.78 \pm 0.01$ | $65.31 \pm 0.00$ | $66.80 \pm 0.01$ | $71.83 \pm 2.07$ | $57.72 \pm 0.16$ | $57.98 \pm 0.23$ |
| **GCN** | $93.58 \pm 0.43$ | $67.83 \pm 0.39$ | $\textbf{80.93} \pm \textbf{0.59}$ | $76.21 \pm 0.10$ | $61.84 \pm 0.18$ | $59.87 \pm 0.14$ |
| **GAT** | $91.67 \pm 1.85$ | $79.31 \pm 1.27$ | $78.18 \pm 1.43$ | $80.26 \pm 0.03$ | $61.88 \pm 0.10$ | $58.99 \pm 0.06$ |
| **GraphTrans** | $96.96 \pm 0.59$ | $71.76 \pm 1.53$ | $80.12 \pm 0.58$ | $97.67 \pm 1.22$ | $62.46 \pm 0.85$ | $60.73 \pm 1.97$ |
| **GraphGPS** | $68.24 \pm 2.18$ | $66.54 \pm 2.44$ | $73.46 \pm 0.30$ | $74.97 \pm 1.06$ | $60.87 \pm 0.07$ | $\textbf{61.71} \pm \textbf{0.07}$ |
| **DIFFormer** | $95.97 \pm 0.97$ | $74.48 \pm 1.31$ | $79.67 \pm 0.87$ | $89.94 \pm 3.57$ | $64.13 \pm 0.58$ | $60.94 \pm 2.17$ |
| **ADIT-INVERSE** | $97.39 \pm 1.67$ | $73.82 \pm 1.45$ | $\textbf{80.38} \pm \textbf{1.40}$ | $83.67 \pm 0.09$ | $60.85 \pm 0.22$ | $\textbf{65.29} \pm \textbf{0.16}$ |
| **ADIT-SERIES** | $93.58 \pm 0.46$ | $67.03 \pm 0.53$ | $\textbf{82.03} \pm \textbf{0.42}$ | $80.24 \pm 0.23$ | $59.70 \pm 0.35$ | $\textbf{62.28} \pm \textbf{0.36}$ |

Table 3: Results on dynamic protein interaction networks `DDPIN` with splits by different protein identification methods. The predictive tasks span node regression, edge regression and link prediction.

| | Node Regression (RMSE) ($\downarrow$) | | Edge Regression (RMSE) ($\downarrow$) | | Link Prediction (ROC-AUC) ($\uparrow$) | |
|---|---|---|---|---|---|---|
| | Valid | Test | Valid | Test | Valid | Test |
| **MLP** | $2.44 \pm 0.02$ | $2.34 \pm 0.03$ | $0.163 \pm 0.004$ | $0.185 \pm 0.003$ | $0.658 \pm 0.014$ | $0.616 \pm 0.117$ |
| **GCN** | $3.74 \pm 0.01$ | $3.40 \pm 0.01$ | $0.170 \pm 0.004$ | $0.184 \pm 0.004$ | $0.673 \pm 0.088$ | $0.683 \pm 0.062$ |
| **GAT** | $3.10 \pm 0.09$ | $2.86 \pm 0.06$ | $0.164 \pm 0.001$ | $0.176 \pm 0.001$ | $0.765 \pm 0.023$ | $0.687 \pm 0.031$ |
| **SGC** | $3.66 \pm 0.00$ | $3.40 \pm 0.02$ | $0.177 \pm 0.016$ | $0.190 \pm 0.004$ | $0.658 \pm 0.044$ | $0.775 \pm 0.042$ |
| **GraphTrans** | OOM | OOM | OOM | OOM | OOM | OOM |
| **GraphGPS** | $1.80 \pm 0.01$ | $\textbf{1.65} \pm \textbf{0.02}$ | $0.165 \pm 0.016$ | $0.159 \pm 0.007$ | $0.604 \pm 0.029$ | $0.673 \pm 0.068$ |
| **DIFFormer** | $2.06 \pm 0.04$ | $2.04 \pm 0.02$ | $0.173 \pm 0.012$ | $\textbf{0.155} \pm \textbf{0.002}$ | $0.935 \pm 0.030$ | $\textbf{0.902} \pm \textbf{0.054}$ |
| **ADIT-INVERSE** | $1.83 \pm 0.02$ | $\textbf{1.75} \pm \textbf{0.02}$ | $0.146 \pm 0.002$ | $\textbf{0.147} \pm \textbf{0.002}$ | $0.946 \pm 0.027$ | $\textbf{0.957} \pm \textbf{0.018}$ |
| **ADIT-SERIES** | $1.56 \pm 0.02$ | $\textbf{1.49} \pm \textbf{0.03}$ | $0.146 \pm 0.002$ | $\textbf{0.144} \pm \textbf{0.001}$ | $0.828 \pm 0.026$ | $\textbf{0.866} \pm \textbf{0.036}$ |

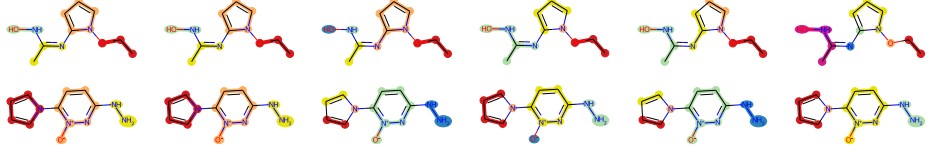

Ground Truth    ADiT (0.697)    GCN (0.685)    GAT (0.664)    GraphGPS (0.694)   Difformer (0.674)

Figure 4: Testing cases for molecular mapping operators generated by different models with averaged testing Accuracy ($\uparrow$) reported. The task is to generate subgraph-level partitions resembling expert annotations (ground-truth) for each molecule instance. See more results in Appendix G.1.

obtained by one protein identification method and records the metabolic cycles of yeast cells. The networks have distinct topological features (e.g., distribution of cliques) as observed by (Fu & He, 2022), and we use 6/1/5 networks for train/valid/test. To test the generalization of the model across different tasks, we consider: i) node regreesion for gene expression values (measured by RMSE); 2) edge regression for predicting the co-expression correlation coefficients (measured by RMSE); 3) link prediction for identifying co-expressed protein pairs (measured by ROC-AUC). Table 3 shows that our models yield the first-ranking results in three tasks. In contrast, ADIT-SERIES performs better in node/edge regression tasks, while ADIT-INVERSE exhibits better competitiveness for link prediction. The possible reason might be that ADIT-INVERSE can better exploit high-order structural information as the matrix inverse can be treated as ADIT-SERIES with $K \to \infty$.

**Molecular Mapping Operator Generation**. Finally we investigate on the generation of molecular coarse-grained mapping operators, an important step for molecular dynamics simulation, aiming to find a representation of how atoms are grouped in a molecule (Li et al., 2020). The task is a graph segmentation problem which can be modeled as predicting edges that indicate where to partition the graph. We use the relative molecular mass to split the data and test the model's extrapolation ability for larger molecules. Fig. 4 compares the testing cases (with more cases in Appendix G.1) generated by different models, which shows the more accurate estimation of our model (we use ADIT-SERIES for experiments) that demonstrates desired generalization.

**Additional Experimental Results.** Due to space limit, we defer more results such as ablation studies and hyper-parameter analysis (for $\beta$, $\theta$ and $K$) along with more discussions to Appendix G.2.

## 6 CONCLUSIONS AND DISCUSSIONS

This paper has systematically studied the generalization capabilities of graph diffusion equations under topological shifts, and shed lights on building generalizable GNNs in the open-world regime. The latter remains a largely under-explored question in graph ML community. Our new model, inspired by advective diffusion equations, has provable topological generalization capability and is implemented as a Transformer-like architecture. It shows superior performance in various graph learning tasks. Our analysis and proposed methodology open new possibilities of leveraging established PDE techniques for building generalizable GNNs.

**Reproducibility Statement.** We supplement the complete proofs for all the theoretical results and detailed information for model implementations and experiments, with references below:

- The proofs for technical results in Sec. 3 are presented in Appendix B.
- The proofs for technical results in Sec. 4 are presented in Appendix C.
- The detailed derivations for our proposed models in Sec. 4.3 are shown in Appendix D.
- The architectures of our models along with pseudo codes are illustrated in Appendix E.
- The detailed information for all experimental datasets is presented in Appendix F.1.
- The details for competitors are provided in Appendix F.2.
- The implementation details for experiments are provided in Appendix F.3.

The source codes will be made publicly available.

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

## CONTENTS

# A    CONNECTION BETWEEN DIFFUSION EQUATIONS AND MESSAGE PASSING

In this section, we provide a systematically introduction on the fundamental connections between graph diffusion equations and neural message passing, as supplementary technical background for our analysis and methodology presented in the main text. Consider graph diffusion equations of the generic form

$$\frac{\partial \mathbf{Z}(t)}{\partial t} = (\mathbf{C}(\mathbf{Z}(t), t; \mathbf{A}) - \mathbf{I})\mathbf{Z}(t), \ \ 0 \le t \le T, \ \ \text{with initial conditions } \mathbf{Z}(0) = \phi_{enc}(\mathbf{X}). \quad (12)$$

As demonstrated by existing works, e.g., Chamberlain et al. (2021a), using finite-difference numerical schemes for solving Eqn. 12 would induce the message passing neural networks of various forms. The latter is recognized as the common paradigm in modern graph neural networks and Transformers whose layer-wise updating aggregates the embeddings of other nodes to compute the embeddings for the next layer.

## A.1    GRAPH NEURAL NETWORKS AS LOCAL DIFFUSION

Consider the explicit Euler's scheme as the commonly used finite-difference method for approximately solving the differential equations, and Eqn. 12 will induce the discrete iterations with step size $\tau$:

$$\frac{\mathbf{Z}^{(k+1)} - \mathbf{Z}^{(k)}}{\tau} \approx (\mathbf{C}(\mathbf{Z}^{(k)}, k; \mathbf{A}) - \mathbf{I})\mathbf{Z}^{(k)}. \quad (13)$$

With some re-arranging we have

$$\mathbf{Z}^{(k+1)} = (1 - \tau)\mathbf{Z}^{(k)} + \tau\mathbf{C}(\mathbf{Z}^{(k)}, k; \mathbf{A})\mathbf{Z}^{(k)}, \quad (14)$$

with the initial states $\mathbf{Z}^{(0)} = \phi_{enc}(\mathbf{X})$. The above updating equation gives one-layer update through residual connection and propagation with $\mathbf{C}(\mathbf{Z}^{(k)}, k; \mathbf{A})$. There are some well-known graph neural network architectures that can be derived with different instantiations of the coupling matrix.

**Simplifying Graph Convolution (SGC).** If one considers $\mathbf{C}(\mathbf{Z}^{(k)}; \mathbf{A}) = \tilde{\mathbf{A}} = \mathbf{D}^{-1/2}\mathbf{A}\mathbf{D}^{-1/2}$, then we will get the one-layer updating rule:

$$\mathbf{Z}^{(k+1)} = (1 - \tau)\mathbf{Z}^{(k)} + \tau\mathbf{D}^{-1/2}\mathbf{A}\mathbf{D}^{-1/2}\mathbf{Z}^{(k)}. \quad (15)$$

This can be seen as one-layer propagation of SGC (Wu et al., 2019) with residual connection, and when $\tau = 1$ it becomes exactly the SGC layer. Since SGC model does not involve feature transformation layers and non-linearity throughout the message passing, one often uses a pre-computed propagation matrix for one-step convolution that is much faster than the multi-layer convolution:

$$\mathbf{Z}^{(K)} = \mathbf{P}^K\mathbf{Z}^{(0)}, \quad \mathbf{P} = (1 - \tau)\mathbf{I} + \tau\mathbf{D}^{-1/2}\mathbf{A}\mathbf{D}^{-1/2}. \quad (16)$$

**Graph Convolution Networks (GCN).** The GCN network inserts feature transformation layers in-between the propagation layers. This can be achieved by considering $K$ stacked piece-wise diffusion equations, where the $k$-th dynamics is given by the differential equation with time boundaries:

$$\frac{\partial \mathbf{Z}(t; k)}{\partial t} = (\mathbf{C} - \mathbf{I})\mathbf{Z}(t; k), \ \ t \in [t_{k-1}, t_k], \ \ \text{with initial conditions } \mathbf{Z}(t_{k-1}; k) = \phi_{int}^{(k)}(\mathbf{Z}(t_{k-1}; k-1)), \quad (17)$$

where $\phi_{int}^{(k)}$ denotes the node-wise feature transformation of the $k$-th layer. Assume $\mathbf{C} = \mathbf{D}^{-1/2}\mathbf{A}\mathbf{D}^{-1/2}$. Then consider one-step feed-forward of the explicit Euler scheme for Eqn. 17, and one can obtain the updating rule at the $k$-th layer:

$$\mathbf{Z}^{(k+1)} = \phi_{int}^{(k+1)}\left((1 - \tau)\mathbf{Z}^{(k)} + \tau\mathbf{D}^{-1/2}\mathbf{A}\mathbf{D}^{-1/2}\mathbf{Z}^{(k)}\right). \quad (18)$$

This corresponds to one GCN layer (Kipf & Welling, 2017) if one considers $\phi_{int}^{(k+1)}$ as a fully-connected neural layer with ReLU activation and simply sets $\tau = 1$.

**Graph Attention Networks (GAT).** Apart from the system with constant diffusivity, one can assume $\mathbf{C}$ to be time-dependent through a function of $\mathbf{Z}(t)$. In this way, the coupling function plays a similar

role as the attention function that computes the pairwise similarity between neighboring nodes. In specific, consider the coupling matrix of the form involving a row-normalized similarity function:

$$\mathbf{C}(\mathbf{Z}(t); \mathbf{A}) = [c_{uv}(t)]_{u,v \in \mathcal{V}}, \quad c_{uv}(t) = \mathbb{I}[(u, v) \in \mathcal{E}] \cdot \frac{\eta(\mathbf{z}_u(t), \mathbf{z}_v(t))}{\sum_{w,(u,w) \in \mathcal{E}} \eta(\mathbf{z}_u(t), \mathbf{z}_w(t))}. \tag{19}$$

Similar to the preceding cases, we use the explicit Euler's scheme and obtain the one-layer updating rule of GAT (Velickovic et al., 2018) with residual connection:

$$\mathbf{Z}^{(k+1)} = (1 - \tau)\mathbf{Z}^{(k)} + \tau\mathbf{C}^{(k)}\mathbf{Z}^{(k)}, \quad c_{uv}^{(k)} = \mathbb{I}[(u, v) \in \mathcal{E}] \cdot \frac{\eta(\mathbf{z}_u^{(k)}, \mathbf{z}_v^{(k)})}{\sum_{w,(u,w) \in \mathcal{E}} \eta(\mathbf{z}_u^{(k)}, \mathbf{z}_w^{(k)})}. \tag{20}$$

In GAT instantiation, the similarity function $\eta$ is further assumed to be

$$\eta(\mathbf{z}_u^{(k)}, \mathbf{z}_v^{(k)}) = \exp\left(\text{LeakyReLU}(\mathbf{a}^\top[\mathbf{\Theta}\mathbf{z}_u^{(k)} \| \mathbf{\Theta}\mathbf{z}_v^{(k)}])\right), \tag{21}$$

where $\mathbf{a} \in \mathbb{R}^d$ and $\mathbf{\Theta} \in \mathbb{R}^{d \times d}$ are trainable parameters specific to the $k$-th layer (we omit the superscript $k$ for simplicity).

**High-Order Propagation.** Besides the explicit numerical scheme, one can also utilize the implicit scheme and multi-step schemes (e.g., Runge-Kutta) for solving the diffusion equation, and the induced updating form will involve high-order information (Chamberlain et al., 2021a).

## A.2 TRANSFORMERS AS NON-LOCAL DIFFUSION

The original architectures of Transformers (Vaswani et al., 2017) involve self-attention layers as the key module, where the attention measures the pairwise influence between arbitrary token pairs in the input. There are recent works, e.g., Dwivedi & Bresson (2020); Ying et al. (2021); Wu et al. (2021); Rampásek et al. (2022); Wu et al. (2022b) transferring the Transformer architectures originally designed for sequence inputs into graph-structured data, and the attention is computed for node pairs in the graph. Different from GAT that only attends on neighboring nodes in each propagation, the attention of Transformers targets arbitrary node pairs in the graph, which can be seen as a counterpart of non-local diffusion. In specific, the coupling matrix allows non-zero entries for arbitrary location pairs and can be instantiated as a global attention. Then using the explicit Euler's scheme as Eqn. 14 we can obtain the self-attention propagation layer of common Transformers:

$$\mathbf{Z}^{(k+1)} = (1 - \tau)\mathbf{Z}^{(k)} + \tau\mathbf{C}^{(k)}\mathbf{Z}^{(k)}, \quad c_{uv}^{(k)} = \frac{\eta(\mathbf{z}_u^{(k)}, \mathbf{z}_v^{(k)})}{\sum_{w \in \mathcal{V}} \eta(\mathbf{z}_u^{(k)}, \mathbf{z}_w^{(k)})}. \tag{22}$$

For obtaining the fully-connected layers and non-linear activations adopted in Transformers, one can inherit the spirit of GCN and extend the diffusion model to $K$ piece-wise equations as Eqn. 17. Then the layer-wise updating rule will involve $\phi_{int}^{(k)}$ in-between two attention layers. In original Transformers, $\eta$ is considered as a dot-then-scale exponential function

$$\eta(\mathbf{z}_u^{(k)}, \mathbf{z}_v^{(k)}) = \exp\left(\frac{(\mathbf{W}_Q\mathbf{z}_u^{(k)})^\top(\mathbf{W}_K\mathbf{z}_v^{(k)})}{\sqrt{d}}\right), \tag{23}$$

and there also exist more scalable choices, such as the attention function inspired by diffusion (Wu et al., 2023) that can compute all-pair interactions with linear complexity.

## B ANALYSIS AND PROOFS IN SECTION 3

Before we go into the proofs for our main results, we first introduce some basic concepts and technical lemmas. For any matrix $\mathbf{A} = [a_{ij}]_{N \times N}$ we define

- $\sigma_{max}(\mathbf{A}) = \max_i\{|\lambda_i|\}$ denotes its singular value with the largest magnitude.
- $n_{max}(\mathbf{A}) = \max_{ij}\{|a_{ij}|\}$ denotes its absolute maximum entry.

**Lemma 1.** *For any two $N \times N$ matrices $\mathbf{A}$ and $\mathbf{B}$, we have the upper bound for the L2-norm of their Hadamard product*

$$\|\mathbf{A} \odot \mathbf{B}\|_2 \leq \sqrt{N} \cdot n_{max}(\mathbf{A})\|\mathbf{B}\|_2. \tag{24}$$

*Proof.* If $\mathbf{A}$ is positive definite, it is known that

$$\|\mathbf{A} \odot \mathbf{B}\|_2 \le n_{max}(\mathbf{A})\|\mathbf{B}\|_2. \tag{25}$$

We next prove the conclusion for $\mathbf{A}$ that is not positive definite. Let $\{\mathbf{s}_i\}_{i=1}^N$ be the standard basis vectors in the space $\mathbb{R}^N$, and then any vector $\mathbf{c}$ can be denoted as $\mathbf{c} = \sum_{i=1}^N c_i \mathbf{s}_i$. For $\forall i \in \{1, \cdots, N\}$, we have

$$\|(\mathbf{A} \odot \mathbf{B})\mathbf{s}_i\|_2 \le n_{max}(\mathbf{A}) \cdot \|\mathbf{B}\mathbf{s}_i\|_2 \le n_{max}(\mathbf{A}) \cdot \|\mathbf{B}\|_2. \tag{26}$$

Therefore we have the result

$$\|(\mathbf{A} \odot \mathbf{B})\mathbf{c}\|_2 \le \sum_{i=1}^N |c_i| \cdot \|(\mathbf{A} \odot \mathbf{B})\mathbf{s}_i\|_2 \le n_{max}(\mathbf{A}) \cdot \|\mathbf{B}\|_2 \cdot \sum_{i=1}^N |c_i|. \tag{27}$$

According to Cauchy-Schwarz inequality, we have $\sum_{i=1}^N |c_i| \le \sqrt{N}\|\mathbf{c}\|_2$. Then the result of the lemma can be obtained by noting that $\|(\mathbf{A} \odot \mathbf{B})\mathbf{c}\|_2 \le \sqrt{N} \cdot n_{max}(\mathbf{A}) \cdot \|\mathbf{B}\|_2 \cdot \|\mathbf{c}\|_2$. $\square$

### B.1 ANALYSIS FOR LINEAR DIFFUSION EQUATIONS

We first consider the diffusion equations with fixed diffusivity and coupling matrix $\mathbf{C}(\mathbf{Z}(t), t; \mathbf{A}) = \mathbf{C}$. The model can be described as the following linear differential equation:

$$\frac{\partial \mathbf{Z}(t)}{\partial t} = (\mathbf{C} - \mathbf{I})\mathbf{Z}(t), \ \ 0 \le t \le T, \ \ \text{with initial conditions } \mathbf{Z}(0) = \phi_{enc}(\mathbf{X}). \tag{28}$$

The proposition below shows the change of $\mathbf{Z}(T)$ given by the dynamics of the above model w.r.t. the variation of graph topology $\mathbf{A}$.

**Proposition 1.** *If the coupling matrix $\mathbf{C}$ is set as the normalized adjacency $\tilde{\mathbf{A}} = \mathbf{D}^{-1}\mathbf{A}$ or $\tilde{\mathbf{A}} = \mathbf{D}^{-1/2}\mathbf{A}\mathbf{D}^{-1/2}$, where $\mathbf{D}$ denotes the diagonal degree matrix of $\mathbf{A}$, then the change of $\mathbf{Z}(T; \tilde{\mathbf{A}})$ given by Eqn. 3 w.r.t. a small perturbation $\Delta\tilde{\mathbf{A}}$ is $\left\|\mathbf{Z}(T; \tilde{\mathbf{A}} + \Delta\tilde{\mathbf{A}}) - \mathbf{Z}(T; \tilde{\mathbf{A}})\right\|_2 = \mathcal{O}\left(\exp\left(\|\Delta\tilde{\mathbf{A}}\|_2 T\right)\right).$*

*Proof.* When the coupling matrix is independent of time $t$, i.e., set as a fixed matrix $\mathbf{C}$, the differential equation Eqn. 3 becomes the linear system Eqn. 28 and has a closed-form solution

$$\mathbf{Z}(T) = e^{-(\mathbf{I}-\mathbf{C})T}\mathbf{Z}(0), \ \ T \ge 0. \tag{29}$$

For an arbitrary given finite time $T$, the final state $\mathbf{Z}(T)$ is determined by the initial value $\mathbf{Z}(0)$ and the coupling matrix $\mathbf{C}$. According to the definition in the proposition $\mathbf{C} = \tilde{\mathbf{A}}$, for given $T$ and initial states $\mathbf{Z}(0)$, the node representations $\mathbf{Z}(T)$ can be considered as a function of $\tilde{\mathbf{A}}$. Let $\tilde{\mathbf{A}}' = \tilde{\mathbf{A}} + \Delta\tilde{\mathbf{A}}$. Using basic calculus of matrix exponentials (Horn & Johnson, 2012), we have

$$\begin{aligned}
\mathbf{Z}(T; \tilde{\mathbf{A}}') - \mathbf{Z}(T; \tilde{\mathbf{A}}) &= e^{-(\mathbf{I}-\tilde{\mathbf{A}}')T}\mathbf{Z}(0) - e^{-(\mathbf{I}-\tilde{\mathbf{A}})T}\mathbf{Z}(0) \\
&= \left[e^{-(\mathbf{I}-\tilde{\mathbf{A}}')T} - e^{-(\mathbf{I}-\tilde{\mathbf{A}})T}\right]\mathbf{Z}(0) \\
&= \left[e^{-(\mathbf{I}-\tilde{\mathbf{A}}')T} - e^{-(\mathbf{I}-\tilde{\mathbf{A}})T}\right]e^{(\mathbf{I}-\tilde{\mathbf{A}})T}e^{-(\mathbf{I}-\tilde{\mathbf{A}})T}\mathbf{Z}(0) \\
&= \left[e^{(\tilde{\mathbf{A}}'-\tilde{\mathbf{A}})T} - \mathbf{I}\right]e^{-(\mathbf{I}-\tilde{\mathbf{A}})T}\mathbf{Z}(0) \\
&= \left[e^{(\tilde{\mathbf{A}}'-\tilde{\mathbf{A}})T} - \mathbf{I}\right]\mathbf{Z}(T; \tilde{\mathbf{A}}).
\end{aligned} \tag{30}$$

The second-to-last step follows $e^{\tilde{\mathbf{A}}'+\tilde{\mathbf{A}}} = e^{\tilde{\mathbf{A}}'}e^{\tilde{\mathbf{A}}} = e^{\tilde{\mathbf{A}}}e^{\tilde{\mathbf{A}}'}$ which holds for arbitrary $\tilde{\mathbf{A}}'$ and $\tilde{\mathbf{A}}$ that are commutative. This property is satisfied due to that given the small perturbation $\Delta\tilde{\mathbf{A}}$, $\tilde{\mathbf{A}}'$ and $\tilde{\mathbf{A}}$ share the same eigenspace, and $\tilde{\mathbf{A}}'\tilde{\mathbf{A}} = \mathbf{U}^{-1}\Lambda'\mathbf{U}\mathbf{U}^{-1}\Lambda\mathbf{U} = \mathbf{U}^{-1}\Lambda'\Lambda\mathbf{U} = \mathbf{U}^{-1}\Lambda\Lambda'\mathbf{U} = \tilde{\mathbf{A}}\tilde{\mathbf{A}}'$. From Eqn. 30, we have

$$\left\|\mathbf{Z}(T; \tilde{\mathbf{A}}') - \mathbf{Z}(T; \tilde{\mathbf{A}})\right\|_2 = \left\|\left(e^{\Delta\tilde{\mathbf{A}}T} - \mathbf{I}\right)\mathbf{Z}(T; \tilde{\mathbf{A}})\right\|_2 \le \left\|e^{\Delta\tilde{\mathbf{A}}T} - \mathbf{I}\right\|_2 \sigma_{max}\left(\mathbf{Z}(T; \tilde{\mathbf{A}})\right), \tag{31}$$

where $\sigma_{max}$ denotes the singular value with the largest magnitude. Moreover, the series expansion of the matrix exponential induces that

$$e^{\Delta\tilde{\mathbf{A}}T} = \mathbf{I} + \Delta\tilde{\mathbf{A}}T + \frac{1}{2!}\Delta\tilde{\mathbf{A}}^2T^2 + \cdots + \frac{1}{n!}\Delta\tilde{\mathbf{A}}^nT^n + \cdots \tag{32}$$

Combing Eqn. 31 and 32, we have

$$
\begin{aligned}
\left\|\mathbf{Z}(T;\tilde{\mathbf{A}}') - \mathbf{Z}(T;\tilde{\mathbf{A}})\right\|_2 &\leq \left\|\sum_{k=1}^{\infty} \frac{1}{k!}(\Delta\tilde{\mathbf{A}})^k T^k\right\|_2 \sigma_{max}\left(\mathbf{Z}(T;\tilde{\mathbf{A}})\right) \\
&\leq \sum_{k=1}^{\infty} \frac{1}{k!}\|\Delta\tilde{\mathbf{A}}\|_2^k T^k \sigma_{max}\left(\mathbf{Z}(T;\tilde{\mathbf{A}})\right) \\
&= \left(e^{\|\Delta\tilde{\mathbf{A}}\|_2 T} - 1\right)\sigma_{max}\left(\mathbf{Z}(T;\tilde{\mathbf{A}})\right).
\end{aligned}
\tag{33}
$$

We thus arrive at the result in the proposition and conclude the proof. □

## B.2 Analysis with Feature Transformations

The diffusion model in Eqn. 28 assumes no non-linearity throughout the continuous dynamics and the neural networks are only applied to $\phi_{enc}$ and $\phi_{dec}$. Similar to modern graph neural networks (Kipf & Welling, 2017), the common practice is to insert feature transformations in-between two propagation layers. In this spirit, the diffusion model can be modified as a cascade of $K$ piecewise continuous dynamics, where the $k$-th layer determines the diffusion trajectory from $t_{k-1}$ to $t_k$ and can be formulated as a differential equation with time boundaries:

$$\frac{\partial\mathbf{Z}(t;k)}{\partial t} = (\mathbf{C}-\mathbf{I})\mathbf{Z}(t;k), \ t \in [t_{k-1}, t_k], \ \text{with initial conditions } \mathbf{Z}(t_{k-1};k) = \phi_{int}^{(k)}(\mathbf{Z}(t_{k-1};k-1)). \tag{34}$$

The feature transformation $\phi_{int}^{(k)}$ can be e.g., a fully-connected neural layer with trainable weights and non-linear activation function. With $K$ layers as a cascade, the model induces a diffusion process from $t_0 = 0$ to $t_K$ and gives rise to the embedding trajectory: $\mathbf{Z}(0) = \mathbf{Z}^{(0)} \to \mathbf{Z}^{(1)} \to \cdots \to \mathbf{Z}^{(K)}$, where $\mathbf{Z}^{(k)} = \mathbf{Z}(t_k;k)$. Despite the potentially better capacity, we can show that the final representations $\mathbf{Z}^{(K)} = \mathbf{Z}(t_K;K)$ in this system are also (exponentially) sensitive to the change of graph structures, which may hinder decent generalization.

**Proposition 4.** *For the graph diffusion model Eqn. 34 with constant coupling matrix* $\mathbf{C} = \tilde{\mathbf{A}}$, *the change of output states* $\mathbf{Z}(t_K;K,\tilde{\mathbf{A}})$ *w.r.t. the perturbation* $\Delta\tilde{\mathbf{A}}$ *is* $\mathcal{O}\left(\exp\left(\|\Delta\tilde{\mathbf{A}}\|_2 t_K\right)\right)$.

*Proof.* The differential equation Eqn. 34 has the closed-form solution for the diffusion dynamics from $t_{k-1}$ to $t_k$:

$$\mathbf{Z}(t_{k-1}+\tau;k) = e^{-(\mathbf{I}-\mathbf{C})\tau}\mathbf{Z}(t_{k-1};k), \ \tau \in [0,\tau_k], \ \text{where } \tau_k = t_k - t_{k-1}. \tag{35}$$

Therefore we have $\mathbf{Z}(t_k;k) = e^{-(\mathbf{I}-\mathbf{C})\tau_k}\mathbf{Z}(t_{k-1};k)$. We can treat the representation $\mathbf{Z}^{(k)} = \mathbf{Z}(t_k;k)$ returned by the $k$-th layer as a function of $\mathbf{C} = \tilde{\mathbf{A}}$, and denote the intermediate embeddings of the $k$-th layer updates with the following notations associated to $\tilde{\mathbf{A}}$:

$$
\begin{aligned}
i) & \ \text{feature transformation: } \mathbf{Z}(t_{k-1};k,\tilde{\mathbf{A}}) = \phi_{int}^{(k)}\left(\mathbf{Z}(t_{k-1};k-1,\tilde{\mathbf{A}})\right), \\
ii) & \ \text{diffusion propagation: } \mathbf{Z}(t_k;k,\tilde{\mathbf{A}}) = e^{-(\mathbf{I}-\tilde{\mathbf{A}})\tau_k}\mathbf{Z}(t_{k-1};k,\tilde{\mathbf{A}}).
\end{aligned}
\tag{36}
$$

We first prove the case of $k = 1$, i.e., the diffusion from $t_0 = 0$ to $t_1$. By utilizing the result of Proposition 1 and the fact $\mathbf{Z}(t_0; 0, \tilde{\mathbf{A}}) = \mathbf{Z}(t_0; 0, \tilde{\mathbf{A}}') = \mathbf{Z}(0)$, we have

$$
\begin{aligned}
&\left\| \mathbf{Z}(t_1; 1, \tilde{\mathbf{A}}') - \mathbf{Z}(t_1; 1, \tilde{\mathbf{A}}) \right\|_2 \\
&= \left\| e^{-(\mathbf{I}-\tilde{\mathbf{A}}')\tau_1} \phi_{int}^{(1)} \left( \mathbf{Z}(t_0; 0, \tilde{\mathbf{A}}') \right) - e^{-(\mathbf{I}-\tilde{\mathbf{A}})\tau_1} \phi_{int}^{(1)} \left( \mathbf{Z}(t_0; 0, \tilde{\mathbf{A}}) \right) \right\|_2 \\
&= \left\| e^{-(\mathbf{I}-\tilde{\mathbf{A}}')\tau_1} \phi_{int}^{(1)} \left( \mathbf{Z}(t_0; 0, \tilde{\mathbf{A}}) \right) - e^{-(\mathbf{I}-\tilde{\mathbf{A}})\tau_1} \phi_{int}^{(1)} \left( \mathbf{Z}(t_0; 0, \tilde{\mathbf{A}}) \right) \right\|_2 \\
&= \left\| e^{\Delta\tilde{\mathbf{A}}\tau_1} e^{-(\mathbf{I}-\tilde{\mathbf{A}})\tau_1} \phi_{int}^{(1)} \left( \mathbf{Z}(t_0; 0, \tilde{\mathbf{A}}) \right) - e^{-(\mathbf{I}-\tilde{\mathbf{A}})\tau_1} \phi_{int}^{(1)} \left( \mathbf{Z}(t_0; 0, \tilde{\mathbf{A}}) \right) \right\|_2 \\
&\leq \left[ e^{\|\Delta\tilde{\mathbf{A}}\|_2 \tau_1} - 1 \right] \cdot \sigma_{max} \left( \mathbf{Z}(t_1; 1, \tilde{\mathbf{A}}) \right) \quad (*) \\
&= \left[ e^{\|\Delta\tilde{\mathbf{A}}\|_2 t_1} - 1 \right] \cdot \sigma_{max} \left( \mathbf{Z}(t_1; 1, \tilde{\mathbf{A}}) \right),
\end{aligned}
\tag{37}
$$

where the step with $(*)$ follows Eqn. 30 and 31. We thus have the conclusion held.

We next prove the case for arbitrary $k \geq 2$. To this end, we resort to an important relationship formulated via the lemma below.

**Lemma 2.** *The graph diffusion model with layer-wise updating rule of Eqn. 36 induces a relationship for arbitrary $\tilde{\mathbf{A}}$ and $\tilde{\mathbf{A}}'$:*

$$
\mathbf{Z}(t_k; k, \tilde{\mathbf{A}}') = e^{\Delta\tilde{\mathbf{A}}t_k} \mathbf{Z}(t_k; k, \tilde{\mathbf{A}}).
\tag{38}
$$

*Proof.* We prove this lemma by induction. First, for the case $k = 1$, we have the result holds:

$$
\begin{aligned}
\mathbf{Z}(t_1; 1, \tilde{\mathbf{A}}') &= e^{-(\mathbf{I}-\tilde{\mathbf{A}}')\tau_1} \phi_{int}^{(1)} \left( \mathbf{Z}(t_0; 0, \tilde{\mathbf{A}}') \right) \\
&= e^{(\tilde{\mathbf{A}}'-\tilde{\mathbf{A}})\tau_1} e^{-(\mathbf{I}-\tilde{\mathbf{A}})\tau_1} \phi_{int}^{(1)} \left( \mathbf{Z}(t_0; 0, \tilde{\mathbf{A}}') \right) \\
&= e^{(\tilde{\mathbf{A}}'-\tilde{\mathbf{A}})\tau_1} e^{-(\mathbf{I}-\tilde{\mathbf{A}})\tau_1} \phi_{int}^{(1)} \left( \mathbf{Z}(t_0; 0, \tilde{\mathbf{A}}) \right) \\
&= e^{\Delta\tilde{\mathbf{A}}t_1} \overline{\mathbf{Z}}(t_1; 1, \tilde{\mathbf{A}}).
\end{aligned}
\tag{39}
$$

Next, we prove the result for arbitrary $k > 1$ via induction. Assuming that we have the conclusion held for the case of $k = l - 1$, we consider the case of $k = l$.

$$
\begin{aligned}
\mathbf{Z}(t_k; k, \tilde{\mathbf{A}}') &= e^{-(\mathbf{I}-\tilde{\mathbf{A}}')\tau_k} \phi_{int}^{(k)} \left( \mathbf{Z}(t_{k-1}; k-1, \tilde{\mathbf{A}}') \right) \\
&= e^{(\tilde{\mathbf{A}}'-\tilde{\mathbf{A}})\tau_k} e^{-(\mathbf{I}-\tilde{\mathbf{A}})\tau_k} \phi_{int}^{(k)} \left( \mathbf{Z}(t_{k-1}; k-1, \tilde{\mathbf{A}}') \right) \\
&= e^{(\tilde{\mathbf{A}}'-\tilde{\mathbf{A}})\tau_k} e^{-(\mathbf{I}-\tilde{\mathbf{A}})\tau_k} \phi_{int}^{(k)} \left( e^{\Delta\tilde{\mathbf{A}}t_{k-1}} \mathbf{Z}(t_{k-1}; k-1, \tilde{\mathbf{A}}) \right) \quad (*) \\
&= e^{\Delta\tilde{\mathbf{A}}\tau_k} e^{\Delta\tilde{\mathbf{A}}t_{k-1}} \phi_{int}^{(k)} \left( e^{-(\mathbf{I}-\tilde{\mathbf{A}})\tau_k} \mathbf{Z}(t_{k-1}; k-1, \tilde{\mathbf{A}}) \right) \quad (**) \\
&= e^{\Delta\tilde{\mathbf{A}}t_k} \mathbf{Z}(t_k; k, \tilde{\mathbf{A}}),
\end{aligned}
\tag{40}
$$

where the step with $(*)$ follows the condition held for $k - 1$, and the step with $(**)$ stems from the fact that $\phi_{int}^{(k)}$ is a node-wise feature map, i.e., commutative with arbitrary propagation matrices. $\quad\square$

We then go back to the proof for the proposition w.r.t. arbitrary $k \geq 2$.

$$
\begin{aligned}
&\left\| \mathbf{Z}(t_k; k, \tilde{\mathbf{A}}') - \mathbf{Z}(t_k; k, \tilde{\mathbf{A}}) \right\|_2 \\
&= \left\| e^{-(\mathbf{I}-\tilde{\mathbf{A}}')\tau_k} \phi_{int}^{(k)} \left( \mathbf{Z}(t_{k-1}; k-1, \tilde{\mathbf{A}}') \right) - e^{-(\mathbf{I}-\tilde{\mathbf{A}})\tau_k} \phi_{int}^{(k)} \left( \mathbf{Z}(t_{k-1}; k-1, \tilde{\mathbf{A}}) \right) \right\|_2 \\
&= \left\| e^{-(\mathbf{I}-\tilde{\mathbf{A}}')\tau_k} \phi_{int}^{(k)} \left( e^{\Delta\tilde{\mathbf{A}} t_{k-1}} \mathbf{Z}(t_{k-1}; k-1, \tilde{\mathbf{A}}) \right) - e^{-(\mathbf{I}-\tilde{\mathbf{A}})\tau_k} \phi_{int}^{(k)} \left( \mathbf{Z}(t_{k-1}; k-1, \tilde{\mathbf{A}}) \right) \right\|_2 \quad (*) \\
&= \left\| e^{-(\mathbf{I}-\tilde{\mathbf{A}}')\tau_k} e^{\Delta\tilde{\mathbf{A}} t_{k-1}} \phi_{int}^{(k)} \left( \mathbf{Z}(t_{k-1}; k-1, \tilde{\mathbf{A}}) \right) - e^{-(\mathbf{I}-\tilde{\mathbf{A}})\tau_k} \phi_{int}^{(k)} \left( \mathbf{Z}(t_{k-1}; k-1, \tilde{\mathbf{A}}) \right) \right\|_2 \\
&= \left\| e^{\Delta\tilde{\mathbf{A}} t_k} e^{-(\mathbf{I}-\tilde{\mathbf{A}})\tau_k} \phi_{int}^{(k)} \left( \mathbf{Z}(t_{k-1}; k-1, \tilde{\mathbf{A}}) \right) - e^{-(\mathbf{I}-\tilde{\mathbf{A}})\tau_k} \phi_{int}^{(k)} \left( \mathbf{Z}(t_{k-1}; k-1, \tilde{\mathbf{A}}) \right) \right\|_2 \\
&\leq \left\| e^{\Delta\tilde{\mathbf{A}} t_k} - \mathbf{I} \right\|_2 \cdot \left\| e^{-(\mathbf{I}-\tilde{\mathbf{A}})\tau_k} \phi_{int}^{(k)} \left( \mathbf{Z}(t_{k-1}; k-1, \tilde{\mathbf{A}}) \right) \right\|_2 \\
&\leq \left[ e^{\|\Delta\tilde{\mathbf{A}}\|_2 t_k} - 1 \right] \cdot \sigma_{max} \left( \mathbf{Z}(t_k; k, \tilde{\mathbf{A}}) \right),
\end{aligned}
\tag{41}
$$

where the step with $(*)$ uses Lemma 2 and the last step follows the similar reasoning of Eqn 31 and 33. We thus conclude the proof for arbitrary $k$. $\qquad\square$

### B.3 ANALYSIS FOR NON-LINEAR DIFFUSION

We next extend the analysis to the case with time-dependent diffusivity that induces $\mathbf{C}(\mathbf{Z}(t), t; \mathbf{A})$ evolving with time in the equation. We assume the dependence between the coupling matrix and time is fully given by the impact of node states $\mathbf{Z}(t)$ at each time, so we use $\mathbf{C}(\mathbf{Z}(t); \mathbf{A})$ to simplify the notation. Then the diffusion equation can be written as:

$$
\frac{\partial \mathbf{Z}(t)}{\partial t} = (\mathbf{C}(\mathbf{Z}(t); \mathbf{A}) - \mathbf{I})\mathbf{Z}(t), \ \ 0 \leq t \leq T, \ \ \text{with initial conditions } \mathbf{Z}(0) = \phi_{enc}(\mathbf{X}). \tag{42}
$$

We further instantiate the coupling matrix as a parametric attention function over graph topology that is adopted by (Chamberlain et al., 2021a):

$$
\mathbf{C}(\mathbf{Z}(t); \mathbf{A}) = [c_{uv}(t)]_{u,v\in\mathcal{V}}, \quad c_{uv}(t) = \mathbb{I}[(u,v) \in \mathcal{E}] \cdot \frac{\eta(\mathbf{z}_u(t), \mathbf{z}_v(t))}{\sum_{w,(u,w)\in\mathcal{E}} \eta(\mathbf{z}_u(t), \mathbf{z}_w(t))}. \tag{43}
$$

The differential equation becomes non-linear and has no closed-form solution, yet we can extend our analysis in previous subsections to such a case for the change rate of $\mathbf{Z}(T; \mathbf{A})$ in this system.

**Proposition 2.** *For arbitrary time limit $T$ and bounded function $\eta$, the change of $\mathbf{Z}(T)$ by the diffusion model Eqn. 3 with $\mathbf{C}(\mathbf{Z}(t); \mathbf{A})$ by Eqn. 4 w.r.t. a perturbation $\Delta\mathbf{A}$ is $\mathcal{O}\left(\exp\left(\|\Delta\mathbf{A}\|_2 T\right)\right)$.*

*Proof.* We are to compare the difference of the node representations $\mathbf{Z}(T; \mathbf{A})$ and $\mathbf{Z}(T; \mathbf{A}')$ given by the diffusion model with different input graphs. The diffusion dynamics give the relationship

$$
\mathbf{Z}(T; \mathbf{A}) = \int_0^T \frac{\partial \mathbf{Z}(t; \mathbf{A})}{\partial t} dt = \int_0^T \left[ \mathbf{C}(\mathbf{Z}(t; \mathbf{A}); \mathbf{A}) - \mathbf{I} \right] \mathbf{Z}(t; \mathbf{A}) dt, \tag{44}
$$

$$
\mathbf{Z}(T; \mathbf{A}') = \int_0^T \frac{\partial \mathbf{Z}(t; \mathbf{A}')}{\partial t} dt = \int_0^T \left[ \mathbf{C}(\mathbf{Z}(t; \mathbf{A}'); \mathbf{A}') - \mathbf{I} \right] \mathbf{Z}(t; \mathbf{A}') dt, \tag{45}
$$

where the coupling matrices $\mathbf{C}(\mathbf{Z}(t; \mathbf{A}); \mathbf{A})$ and $\mathbf{C}(\mathbf{Z}(t; \mathbf{A}'); \mathbf{A}')$ are given by Eqn. 4. We define the difference between two coupling matrices at arbitrary time $t$ as

$$
\Delta\mathbf{C}(t) = \mathbf{C}(\mathbf{Z}(t; \mathbf{A}'); \mathbf{A}') - \mathbf{C}(\mathbf{Z}(t; \mathbf{A}); \mathbf{A}). \tag{46}
$$

The lemma below suggests the magnitude of the gap is bounded by the difference between two adjacency matrices.

**Lemma 3.** *For any bounded attention function $\eta$, the difference between coupling matrices satisfies (where $c$ is a constant associated with $\eta$)*

$$
\|\Delta\mathbf{C}(t)\|_2 \leq c(\eta) \cdot \|\Delta\mathbf{A}\|_2. \tag{47}
$$

*Proof.* We define $\mathbf{C}(\mathbf{Z}(t); \mathbf{A}) = \mathbf{N}(\mathbf{Z}(t); \mathbf{A})\mathbf{U}(\mathbf{Z}(t); \mathbf{A})$ where $\mathbf{N}(\mathbf{Z}(t); \mathbf{A})$ and $\mathbf{U}(\mathbf{Z}(t); \mathbf{A})$ denote the un-normalized attention matrix and (diagonal) normalization matrix, respectively:

$$\mathbf{U}(\mathbf{Z}(t); \mathbf{A}) = [u_{uv}(t)]_{u,v \in \mathcal{V}}, \quad u_{uv}(t) = \mathbb{I}[(u, v) \in \mathcal{E}]\eta(\mathbf{z}_u(t), \mathbf{z}_v(t)), \tag{48}$$

$$\mathbf{N}(\mathbf{Z}(t); \mathbf{A}) = \text{diag}^{-1}(n_u(t))_{u \in \mathcal{V}}, \quad n_u(t) = \sum_{v,(u,v) \in \mathcal{E}} \eta(\mathbf{z}_u(t), \mathbf{z}_v(t)). \tag{49}$$

Furthermore, we define a global un-normalized attention matrix and its (diagonal) normalization matrix:

$$\overline{\mathbf{U}}(\mathbf{Z}(t)) = [\overline{u}_{uv}(t)]_{u,v \in \mathcal{V}}, \quad \overline{u}_{uv}(t) = \eta(\mathbf{z}_u(t), \mathbf{z}_v(t)), \tag{50}$$

$$\overline{\mathbf{N}}(\mathbf{Z}(t)) = \text{diag}^{-1}(\overline{n}_u(t))_{u \in \mathcal{V}}, \quad \overline{n}_u(t) = \sum_{v \in \mathcal{V}} \eta(\mathbf{z}_u(t), \mathbf{z}_v(t)). \tag{51}$$

We thus have the results for attention matrices defined with any two different graphs $\mathbf{A}$, $\mathbf{A}'$:

$$\mathbf{C}(\mathbf{Z}(t); \mathbf{A}) = \mathbf{N}(\mathbf{Z}(t); \mathbf{A})\mathbf{U}(\mathbf{Z}(t); \mathbf{A}) = \mathbf{N}(\mathbf{Z}(t); \mathbf{A})(\overline{\mathbf{U}}(\mathbf{Z}(t)) \odot \mathbf{A}), \tag{52}$$

$$\mathbf{C}(\mathbf{Z}(t); \mathbf{A}') = \mathbf{N}(\mathbf{Z}(t); \mathbf{A}')\mathbf{U}(\mathbf{Z}(t); \mathbf{A}') = \mathbf{N}(\mathbf{Z}(t); \mathbf{A}')(\overline{\mathbf{U}}(\mathbf{Z}(t)) \odot \mathbf{A}'). \tag{53}$$

By definition we have

$$\begin{aligned}
&\|\Delta\mathbf{C}(t)\|_2 \\
&= \|\mathbf{C}(\mathbf{Z}(t); \mathbf{A}') - \mathbf{C}(\mathbf{Z}(t); \mathbf{A})\|_2 \\
&= \|\mathbf{N}(\mathbf{Z}(t); \mathbf{A}')(\overline{\mathbf{U}}(\mathbf{Z}(t)) \odot \mathbf{A}') - \mathbf{N}(\mathbf{Z}(t); \mathbf{A})(\overline{\mathbf{U}}(\mathbf{Z}(t)) \odot \mathbf{A})\|_2 \\
&= \|(\mathbf{N}(\mathbf{Z}(t); \mathbf{A}') - \mathbf{N}(\mathbf{Z}(t); \mathbf{A})) \cdot (\overline{\mathbf{U}}(\mathbf{Z}(t)) \odot \mathbf{A}') + \mathbf{N}(\mathbf{Z}(t); \mathbf{A})(\overline{\mathbf{U}}(\mathbf{Z}(t)) \odot (\mathbf{A}' - \mathbf{A}))\|_2.
\end{aligned} \tag{54}$$

Since $\mathbf{N}(\mathbf{Z}(t); \mathbf{A})$ and $\mathbf{N}(\mathbf{Z}(t); \mathbf{A}')$ are diagonal matrices and according to Lemma 1, we have

$$\begin{aligned}
&\|\mathbf{N}(\mathbf{Z}(t); \mathbf{A})(\overline{\mathbf{U}}(\mathbf{Z}(t)) \odot (\mathbf{A}' - \mathbf{A}))\|_2 \\
&\leq n_{max}(\mathbf{N}(\mathbf{Z}(t); \mathbf{A}')) \cdot \|\overline{\mathbf{U}}(\mathbf{Z}(t)) \odot \Delta\mathbf{A}\|_2 \\
&\leq n_{max}(\mathbf{N}(\mathbf{Z}(t); \mathbf{A}')) \cdot \sqrt{|\mathcal{V}|} \cdot n_{max}(\overline{\mathbf{U}}(\mathbf{Z}(t))) \cdot \|\Delta\mathbf{A}\|_2.
\end{aligned} \tag{55}$$

On the other side, we have

$$\begin{aligned}
&\|(\mathbf{N}(\mathbf{Z}(t); \mathbf{A}') - \mathbf{N}(\mathbf{Z}(t); \mathbf{A})) \cdot (\overline{\mathbf{U}}(\mathbf{Z}(t)) \odot \mathbf{A}')\|_2 \\
&\leq \|\mathbf{N}(\mathbf{Z}(t); \mathbf{A}') - \mathbf{N}(\mathbf{Z}(t); \mathbf{A})\|_2 \cdot \|\overline{\mathbf{U}}(\mathbf{Z}(t)) \odot \mathbf{A}'\|_2 \\
&\leq \|\mathbf{N}(\mathbf{Z}(t); \mathbf{A}')\|_2\|\mathbf{N}(\mathbf{Z}(t); \mathbf{A})\|_2\|\overline{\mathbf{U}}(\mathbf{Z}(t)) \odot \mathbf{A}'\mathbf{1} - \overline{\mathbf{U}}(\mathbf{Z}(t)) \odot \mathbf{A}\mathbf{1}\|_2 \cdot \|\overline{\mathbf{U}}(\mathbf{Z}(t)) \odot \mathbf{A}'\|_2 \\
&= \|\mathbf{N}(\mathbf{Z}(t); \mathbf{A}')\|_2\|\mathbf{N}(\mathbf{Z}(t); \mathbf{A})\|_2\|\overline{\mathbf{U}}(\mathbf{Z}(t)) \odot \Delta\mathbf{A}\mathbf{1}\|_2 \cdot \|\overline{\mathbf{U}}(\mathbf{Z}(t)) \odot \mathbf{A}'\|_2 \\
&\leq n_{max}(\mathbf{N}(\mathbf{Z}(t); \mathbf{A}'))n_{max}(\mathbf{N}(\mathbf{Z}(t); \mathbf{A})) \cdot \sqrt{|\mathcal{V}|} \cdot \|\overline{\mathbf{U}}(\mathbf{Z}(t)) \odot \Delta\mathbf{A}\|_2 \cdot \|\overline{\mathbf{U}}(\mathbf{Z}(t)) \odot \mathbf{A}'\|_2 \\
&\leq n_{max}(\mathbf{N}(\mathbf{Z}(t); \mathbf{A}'))n_{max}(\mathbf{N}(\mathbf{Z}(t); \mathbf{A})) \cdot |\mathcal{V}|^{3/2} \cdot n_{max}(\overline{\mathbf{U}}(\mathbf{Z}(t))) \cdot \sigma_{max}(\overline{\mathbf{U}}(\mathbf{Z}(t))) \cdot \|\Delta\mathbf{A}\|_2,
\end{aligned} \tag{56}$$

where the last step uses Lemma 1 and the fact $n_{max}(\mathbf{A}') \leq 1$. Inserting the results of Eqn. 55 and Eqn. 56 into Eqn. 54 and according to the triangle inequality, we can arrive at the lemma. $\quad\square$

We next continue the proof for the proposition. Our proof follows the similar reasoning line of the proof for Proposition 4 by induction from $t = 0$ to $T$, and in particular, we consider the time interval $\tau = t_k - t_{k-1}$ to be infinitesimal.

We first move from $t = 0$ to $t = \tau$ (i.e., an infinitesimal forward from the initial state $\mathbf{Z}(0)$). Since $\mathbf{Z}(0; \mathbf{A}) = \mathbf{Z}(0; \mathbf{A}') = \mathbf{Z}(0)$, one can easily verify through Eqn. 44 and 45 that

$$\mathbf{Z}(\tau; \mathbf{A}) = e^{-(\mathbf{I} - \mathbf{C}(\mathbf{Z}(0); \mathbf{A}))\tau}\mathbf{Z}(0), \quad \mathbf{Z}(\tau; \mathbf{A}') = e^{-(\mathbf{I} - \mathbf{C}(\mathbf{Z}(0); \mathbf{A}'))\tau}\mathbf{Z}(0). \tag{57}$$

Then similar to the derivation of Eqn. 37, replacing $\tilde{\mathbf{A}}$ by $\mathbf{C}(\mathbf{Z}(0); \mathbf{A})$ and $\Delta\tilde{\mathbf{A}}$ by $\Delta\mathbf{C}(t_0) = \mathbf{C}(t_0; 1, \mathbf{A}') - \mathbf{C}(t_0; 1, \mathbf{A})$, we can easily have

$$
\begin{aligned}
& \|\mathbf{Z}(\tau; \mathbf{A}') - \mathbf{Z}(\tau; \mathbf{A})\|_2 \\
&= \left\| e^{-(\mathbf{I} - \mathbf{C}(\mathbf{Z}(0); \mathbf{A}'))\tau} \mathbf{Z}(0) - e^{-(\mathbf{I} - \mathbf{C}(\mathbf{Z}(0); \mathbf{A}))\tau} \mathbf{Z}(0) \right\|_2 \\
&= \left\| e^{\Delta\mathbf{C}(t_0)} e^{-(\mathbf{I} - \mathbf{C}(\mathbf{Z}(0); \mathbf{A}))\tau} \mathbf{Z}(0) - e^{-(\mathbf{I} - \mathbf{C}(\mathbf{Z}(0); \mathbf{A}))\tau} \mathbf{Z}(0) \right\|_2 \\
&= \left\| \left( e^{\Delta\mathbf{C}(t_0)} - 1 \right) \cdot e^{-(\mathbf{I} - \mathbf{C}(\mathbf{Z}(0); \mathbf{A}))\tau} \mathbf{Z}(0) \right\|_2 \\
&\leq \left[ e^{\|\Delta\mathbf{C}(t_0)\|_2 \tau} - 1 \right] \cdot \sigma_{max}\left(\mathbf{Z}(\tau; \mathbf{A})\right).
\end{aligned}
\tag{58}
$$

We thus have the conclusion held for $t = \tau$.

We next prove the case for arbitrary $t > 0$. And, before we delve into such a case, we prove an important property that acts as the continuous version of Lemma 2.

**Lemma 4.** *The graph diffusion model Eqn. 3 with time-depdendent coupling matrix $\mathbf{C}(\mathbf{Z}(t); \mathbf{A})$ induces a relationship for the representations yielded by arbitrary $\mathbf{A}$ and $\mathbf{A}'$:*

$$
\mathbf{Z}(T; \mathbf{A}') = e^{\int_0^T \Delta\mathbf{C}(t)dt} \mathbf{Z}(T; \mathbf{A}).
\tag{59}
$$

*Proof.* We prove this lemma by induction. First, for the case $T = \tau$, we have the result:

$$
\begin{aligned}
\mathbf{Z}(\tau; \mathbf{A}') &= e^{-(\mathbf{I} - \mathbf{C}(\mathbf{Z}(0); \mathbf{A}'))\tau} \mathbf{Z}(0) \\
&= e^{(\mathbf{C}(\mathbf{Z}(0); \mathbf{A}') - \mathbf{C}(\mathbf{Z}(0); \mathbf{A}))\tau} e^{-(\mathbf{I} - \mathbf{C}(\mathbf{Z}(0); \mathbf{A}))\tau} \mathbf{Z}(0) \\
&= e^{\Delta\mathbf{C}(0)\tau} \mathbf{Z}(\tau; \mathbf{A}).
\end{aligned}
\tag{60}
$$

Next, we prove the result for arbitrary $T > 0$ via induction. Assuming that we have the conclusion held for $T \leq t_s$, we consider the case for an infinitesimal forward step $t_s + \tau$. According to Eqn. 44 and 45 we can easily have

$$
\mathbf{Z}(t_s + \tau; \mathbf{A}) = e^{-(\mathbf{I} - \mathbf{C}(\mathbf{Z}(t_s; \mathbf{A}); \mathbf{A}))\tau} \mathbf{Z}(t_s; \mathbf{A}),
\tag{61}
$$

$$
\mathbf{Z}(t_s + \tau; \mathbf{A}') = e^{-(\mathbf{I} - \mathbf{C}(\mathbf{Z}(t_s; \mathbf{A}'); \mathbf{A}'))\tau} \mathbf{Z}(t_s; \mathbf{A}').
\tag{62}
$$

Therefore we can derive the relationship between $\mathbf{Z}(t_s + \tau; \mathbf{A}')$ and $\mathbf{Z}(t_s + \tau; \mathbf{A})$:

$$
\begin{aligned}
\mathbf{Z}(t_s + \tau; \mathbf{A}') &= e^{-(\mathbf{I} - \mathbf{C}(\mathbf{Z}(t_s; \mathbf{A}'); \mathbf{A}'))\tau} \mathbf{Z}(t_s; \mathbf{A}') \\
&= e^{(\mathbf{C}(\mathbf{Z}(t_s; \mathbf{A}'); \mathbf{A}') - \mathbf{C}(\mathbf{Z}(t_s; \mathbf{A}); \mathbf{A}))\tau} e^{-(\mathbf{I} - \mathbf{C}(\mathbf{Z}(t_s; \mathbf{A}); \mathbf{A}))\tau} \mathbf{Z}(t_s; \mathbf{A}') \\
&= e^{\Delta\mathbf{C}(t_s)\tau} e^{-(\mathbf{I} - \mathbf{C}(\mathbf{Z}(t_s; \mathbf{A}); \mathbf{A}))\tau} e^{\int_0^{t_s} \Delta\mathbf{C}(t)dt} \mathbf{Z}(t_s; \mathbf{A}) \quad (*) \\
&= e^{\Delta\mathbf{C}(t_s)\tau} e^{\int_0^{t_s} \Delta\mathbf{C}(t)dt} e^{-(\mathbf{I} - \mathbf{C}(\mathbf{Z}(t_s; \mathbf{A}); \mathbf{A}))\tau} \mathbf{Z}(t_s; \mathbf{A}) \\
&= e^{\int_0^{t_s + \tau} \Delta\mathbf{C}(t)dt} \mathbf{Z}(t_s + \tau; \mathbf{A}),
\end{aligned}
\tag{63}
$$

where the step with $(*)$ follows the condition held for $t \leq t_s$. $\qquad\square$

We then proceed to prove the proposition for arbitrary $T > 0$.

$$
\begin{aligned}
&\left\| \mathbf{Z}(T; \mathbf{A}') - \mathbf{Z}(T; \mathbf{A}) \right\|_2 \\
&= \left\| e^{-(\mathbf{I} - \mathbf{C}(\mathbf{Z}(T-\tau; \mathbf{A}'); \mathbf{A}'))\tau} \mathbf{Z}(T - \tau; \mathbf{A}') - e^{-(\mathbf{I} - \mathbf{C}(\mathbf{Z}(T-\tau; \mathbf{A}); \mathbf{A}))\tau} \mathbf{Z}(T - \tau; \mathbf{A}) \right\|_2 \\
&= \left\| e^{-(\mathbf{I} - \mathbf{C}(\mathbf{Z}(T-\tau; \mathbf{A}'); \mathbf{A}'))\tau} e^{\int_0^{T-\tau} \Delta \mathbf{C}(t) dt} \mathbf{Z}(T - \tau; \mathbf{A}) - e^{-(\mathbf{I} - \mathbf{C}(\mathbf{Z}(T-\tau; \mathbf{A}); \mathbf{A}))\tau} \mathbf{Z}(T - \tau; \mathbf{A}) \right\|_2 \quad (*) \\
&= \left\| e^{-(\mathbf{I} - \mathbf{C}(\mathbf{Z}(T-\tau; \mathbf{A}); \mathbf{A}))\tau} e^{\Delta \mathbf{C}(T-\tau)\tau} e^{\int_0^{T-\tau} \Delta \mathbf{C}(t) dt} \mathbf{Z}(T - \tau; \mathbf{A}) - e^{-(\mathbf{I} - \mathbf{C}(\mathbf{Z}(T-\tau; \mathbf{A}); \mathbf{A}))\tau} \mathbf{Z}(T - \tau; \mathbf{A}) \right\|_2 \\
&= \left\| e^{\int_0^T \Delta \mathbf{C}(t) dt} e^{-(\mathbf{I} - \mathbf{C}(\mathbf{Z}(T-\tau; \mathbf{A}); \mathbf{A}))\tau} \mathbf{Z}(T - \tau; \mathbf{A}) - e^{-(\mathbf{I} - \mathbf{C}(\mathbf{Z}(T-\tau; \mathbf{A}); \mathbf{A}))\tau} \mathbf{Z}(T - \tau; \mathbf{A}) \right\|_2 \\
&\leq \left\| e^{\int_0^T \Delta \mathbf{C}(t) dt} - \mathbf{I} \right\|_2 \cdot \left\| e^{-(\mathbf{I} - \mathbf{C}(\mathbf{Z}(T-\tau; \mathbf{A}); \mathbf{A}))\tau} \mathbf{Z}(T - \tau; \mathbf{A}) \right\|_2 \\
&= \left\| e^{\int_0^T \Delta \mathbf{C}(t) dt} - \mathbf{I} \right\|_2 \cdot \sigma_{max}\left( \mathbf{Z}(T - \tau; \mathbf{A}) \right) \\
&\leq \left[ e^{c \|\Delta \mathbf{A}\|_2 T} - 1 \right] \cdot \sigma_{max}\left( \mathbf{Z}(T - \tau; \mathbf{A}) \right),
\end{aligned}
\tag{64}
$$

where the step with $(*)$ is due to Lemma 4 and the last step can be obtained by the series expansion of matrix exponential along with Lemma 3. We thus conclude the proof for arbitrary $T$.

$\square$

## B.4 ANALYSIS FOR NON-LOCAL DIFFUSION WITH GLOBAL DIFFUSIVITY

Non-local diffusion is distinguished from the local diffusion discussed in previous subsections, stemming from the different diffusion process enabling quantity transfers between any location pairs in the system. In the context of learning on graphs, the generic non-local diffusion equations can also be described via the following equation:

$$
\frac{\partial \mathbf{Z}(t)}{\partial t} = (\mathbf{C}(\mathbf{Z}(t)) - \mathbf{I})\mathbf{Z}(t), \ \ 0 \leq t \leq T, \ \ \text{with initial conditions} \ \ \mathbf{Z}(0) = \phi_{enc}(\mathbf{X}),
\tag{65}
$$

where the difference from local diffusion lies in the coupling function $\mathbf{C}(\mathbf{Z}(t))$ that allows non-zero entries $c_{uv}$'s for arbitrary node pair $(u, v) \in \mathcal{V} \times \mathcal{V}$. Specifically, as an extension of the attention defined over graph structures, we can consider the global attention that computes similarities between arbitrary node pairs:

$$
\mathbf{C}(\mathbf{Z}(t)) = [c_{uv}(t)]_{u,v \in \mathcal{V}}, \quad c_{uv}(t) = \frac{\eta(\mathbf{z}_u(t), \mathbf{z}_v(t))}{\sum_{w, (u,w) \in \mathcal{E}} \eta(\mathbf{z}_u(t), \mathbf{z}_w(t))}.
\tag{66}
$$

In this way, the feed-forward of the diffusion model becomes independent of $\mathbf{A}$ and we have the following result that gives the generalization bound under a specific condition of data-generating assumption.

**Proposition 3.** *Suppose the label $Y$ is conditionally independent from $A$ with given $\{U_u\}_{u \in \mathcal{V}}$ in the data generation hypothesis of Sec. 3.1, then for non-local diffusion model $\Gamma_\theta$ minimizing the empirical risk $\mathcal{R}_{emp}(\Gamma_\theta; E_{tr}) = \frac{1}{N_{tr}} \sum_i^{N_{tr}} l(\Gamma_\theta(\mathbf{X}^{(i)}, \mathbf{A}^{(i)}), \mathbf{Y}^{(i)})$ over training data $\{(\mathbf{X}^{(i)}, \mathbf{A}^{(i)}, \mathbf{Y}^{(i)})\}$ generated from $p(X, A, Y | E = E_{tr})$, it holds with confidence $1 - \delta$ for the bounded generalization error on unseen data $(\mathbf{X}', \mathbf{A}', \mathbf{Y}')$ from a new environment $E_{te} \neq E_{tr} : \mathcal{R}(\Gamma_\theta; E_{te}) \triangleq$*

$$
\mathbb{E}_{(\mathbf{X}', \mathbf{A}', \mathbf{Y}') \sim p(X, A, Y | E = E_{te})}[l(\Gamma_\theta(\mathbf{X}', \mathbf{A}'), \mathbf{Y}')] \leq \mathcal{R}_{emp}(\Gamma_\theta; E_{tr}) + \mathcal{D}_1(\Gamma, N_{tr}),
\tag{67}
$$

*where $\mathcal{D}_1(\Gamma, N_{tr}) = 2\mathcal{H}(\Gamma) + \mathcal{O}\left( \sqrt{(1/N_{tr}) \log(1/\delta)} \right)$, $\mathcal{H}(\Gamma)$ denotes the Rademacher complexity of the function class of $\Gamma$, $N_{tr}$ is the size of the training set, and $l$ denotes any bounded loss function.*

*Proof.* Since the generation of the label $Y$ is assumed to be independent from $A$ (i.e., the dependence path from $A$ to $Y$ is cut off in Fig. 1), we therefore have the following two properties:

$$
p(X, Y | E = E_{tr}) = p(X, Y | E = E_{te}),
\tag{68}
$$

$$p(X, A, Y|E) = p(X, Y|E)p(A|E). \tag{69}$$

The node features $X$ and labels $Y$ can be treated as generated from an identical distribution shared by training and testing sets.

Moreover, since the non-local diffusion model as defined in Sec. 3.3 does not leverage any information of input graphs $\mathbf{A}$, we have the following result

$$l(\Gamma_\theta(\mathbf{X}, \mathbf{A}), \mathbf{Y}) = l(\Gamma_\theta(\mathbf{X}, \mathbf{A}'), \mathbf{Y}), \ \ \forall \mathbf{A}, \mathbf{A}'. \tag{70}$$

Then consider the expectation of the error on testing data

$$\mathcal{R}(\Gamma_\theta; E) = \mathbb{E}_{(\mathbf{X}, \mathbf{A}, \mathbf{Y}) \sim p(X, A, Y|E)} [l(\Gamma_\theta(\mathbf{X}, \mathbf{A}), \mathbf{Y})]. \tag{71}$$

For any graph adjacency matrix $\mathbf{A}^* \in \text{supp}(p(A))$ from the support of $p(A)$, we have the relationship

$$
\begin{aligned}
&\mathbb{E}_{(\mathbf{X}', \mathbf{A}', \mathbf{Y}') \sim p(X, A, Y|E=E_{te})} [l(\Gamma_\theta(\mathbf{X}', \mathbf{A}'), \mathbf{Y}')] \\
&= \mathbb{E}_{(\mathbf{X}', \mathbf{Y}') \sim p(X, Y|E=E_{te}), \mathbf{A}' \sim p(A|E_{te})} [l(\Gamma_\theta(\mathbf{X}', \mathbf{A}'), \mathbf{Y}')] \\
&= \mathbb{E}_{(\mathbf{X}', \mathbf{Y}') \sim p(X, Y|E=E_{te})} [l(\Gamma_\theta(\mathbf{X}', \mathbf{A}^*), \mathbf{Y}')] \\
&= \mathbb{E}_{(\mathbf{X}', \mathbf{Y}') \sim p(X, Y|E=E_{tr})} [l(\Gamma_\theta(\mathbf{X}', \mathbf{A}^*), \mathbf{Y}')] \\
&= \mathbb{E}_{(\mathbf{X}, \mathbf{Y}) \sim p(X, Y|E=E_{tr}), \mathbf{A} \sim p(A|E_{tr})} [l(\Gamma_\theta(\mathbf{X}, \mathbf{A}), \mathbf{Y})] \\
&= \mathbb{E}_{(\mathbf{X}, \mathbf{A}, \mathbf{Y}) \sim p(X, AY|E=E_{tr})} [l(\Gamma_\theta(\mathbf{X}, \mathbf{A}), \mathbf{Y})].
\end{aligned} \tag{72}
$$

The above result indicates $\mathcal{R}(\Gamma_\theta; E_{tr}) = \mathcal{R}(\Gamma_\theta; E_{te})$. Then for any bounded loss function $l$, the existing result based on the Rademacher complexity (Shalev-Shwartz & Ben-David, 2014) gives the generalization error bound

$$\mathcal{R}(\Gamma_\theta; E_{te}) = \mathcal{R}(\Gamma_\theta; E_{tr}) \leq \mathcal{R}_{emb}(\Gamma_\theta; E_{tr}) + 2\mathcal{H}(\Gamma) + \mathcal{O}\left(\sqrt{\frac{\log(1/\delta)}{N_{tr}}}\right), \tag{73}$$

where $\mathcal{H}(\Gamma)$ denotes the Rademacher complexity of the function class induced by $\Gamma$. We thus conclude the proof for the proposition. $\square$

## C   ANALYSIS AND PROOFS IN SECTION 4

**Theorem 1.** *For the advective diffusion model Eqn. 7 with $\mathbf{C}$ pre-computed by global attention over $\mathbf{Z}(0)$ and fixed velocity $\mathbf{V} = \tilde{\mathbf{A}}$, the change rate of node representations $\mathbf{Z}(T; \tilde{\mathbf{A}})$ w.r.t. $\Delta\tilde{\mathbf{A}}$ can be reduced to $\mathcal{O}(\psi(\|\Delta\tilde{\mathbf{A}}\|_2))$ where $\psi$ denotes an arbitrary polynomial function.*

*Proof.* The advective diffusion equation with coupling matrix $\mathbf{C}$ pre-computed by attention network $\eta(\mathbf{z}_u(0), \mathbf{z}_v(0))$ and fixed velocity $\mathbf{V} = \tilde{\mathbf{A}}$, we have its closed-form solution

$$\mathbf{Z}(T) = e^{-(\mathbf{I} - \mathbf{C} - \beta\tilde{\mathbf{A}})T}\mathbf{Z}(0), \ \ T \geq 0. \tag{74}$$

We next conclude the proof by construction. Notice that the initial states are given by the encoder MLP $\mathbf{Z}(0) = \phi_{enc}(\mathbf{X})$. According to our data generation hypothesis in Fig. 1, we know that node embeddings are generated from the latents of each node (we use $u_u$ to denote the realization of $U_u$): $\mathbf{x}_u = g(\mathbf{U}_u; W)$ and graph adjacency is generated through a pair-wise function $a_{uv} = h(\mathbf{U}_u, \mathbf{U}_v; W, E)$. We can construct a mapping to obtain a propagation matrix in the form of $\mathbf{C} = \overline{\mathbf{C}} + m\log(\mathbf{I} + \tilde{\mathbf{A}}) - \beta\tilde{\mathbf{A}}$, where $\overline{\mathbf{C}}$ is independent from $\mathbf{A}$ and $m$ is an arbitrary non-negative number, from the composition of the MLP encoder $\phi_{enc}$ and attention network $\eta$ due to the universal approximation results that hold for MLPs on the compact set (Hornik et al., 1989).

Then we consider the difference between node representations generated by two graph adjacency matrices:

$$
\begin{aligned}
\mathbf{Z}(T; \tilde{\mathbf{A}}') - \mathbf{Z}(T; \tilde{\mathbf{A}}) &= e^{-(\mathbf{I} - \mathbf{C}' - \beta\tilde{\mathbf{A}}')T}\mathbf{Z}(0) - e^{-(\mathbf{I} - \mathbf{C} - \beta\tilde{\mathbf{A}}')T}\mathbf{Z}(0) \\
&= \left[e^{-(\mathbf{I} - \overline{\mathbf{C}} - m\log(\mathbf{I} + \tilde{\mathbf{A}}') + \beta\tilde{\mathbf{A}}' - \beta\tilde{\mathbf{A}}')T} - e^{-(\mathbf{I} - \overline{\mathbf{C}} - m\log(\mathbf{I} + \tilde{\mathbf{A}}) + \beta\tilde{\mathbf{A}} - \beta\tilde{\mathbf{A}})T}\right]\mathbf{Z}(0) \\
&= \left[e^{(m\log(\mathbf{I} + \tilde{\mathbf{A}}') - m\log(\mathbf{I} + \tilde{\mathbf{A}}))T} - \mathbf{I}\right]e^{-(\mathbf{I} - \overline{\mathbf{C}} - m\log(\mathbf{I} + \tilde{\mathbf{A}}))T}\mathbf{Z}(0) \\
&= \left[(\mathbf{I} + \tilde{\mathbf{A}}')^{mT}(\mathbf{I} + \tilde{\mathbf{A}})^{-mT} - \mathbf{I}\right]\mathbf{Z}(T; \tilde{\mathbf{A}}),
\end{aligned}
$$

$$\tag{75}$$

where the penultimate step is based on the commutative property for $m \log(\mathbf{I} + \tilde{\mathbf{A}})$ and $m \log(\mathbf{I} + \tilde{\mathbf{A}}')$ given the small perturbation $\Delta\tilde{\mathbf{A}}$. We proceed to consider the L2 norm of the difference:

$$
\begin{aligned}
\left\| \mathbf{Z}(T; \tilde{\mathbf{A}}') - \mathbf{Z}(T; \tilde{\mathbf{A}}) \right\|_2 &\leq \left\| (\mathbf{I} + \tilde{\mathbf{A}}')^{mT}(\mathbf{I} + \tilde{\mathbf{A}})^{-mT} - \mathbf{I} \right\|_2 \sigma_{max}\left( \mathbf{Z}(T; \tilde{\mathbf{A}}) \right) \\
&= \left\| (\mathbf{I} + \tilde{\mathbf{A}} + \Delta\tilde{\mathbf{A}})^{mT}(\mathbf{I} + \tilde{\mathbf{A}})^{-mT} - \mathbf{I} \right\|_2 \sigma_{max}\left( \mathbf{Z}(T; \tilde{\mathbf{A}}) \right) \quad (76) \\
&= \mathcal{O}(\|\Delta\tilde{\mathbf{A}}\|_2^{mT}).
\end{aligned}
$$

Therefore we conclude the proof for the theorem. $\qquad\square$

**Theorem 2.** *Assume $l$ and $\phi_{dec}$ are Lipschitz continous. Then for any data generated with the data generation hypothesis of Sec. 3.1, if $\Gamma_\theta$ minimizing the empirical risk $\mathcal{R}_{emp}(\Gamma_\theta; E_{tr}) = \frac{1}{N_{tr}} \sum_{i=1}^{N_{tr}} l(\Gamma_\theta(\mathbf{X}^{(i)}, \mathbf{A}^{(i)}), \mathbf{Y}^{(i)})$ over training data $\{(\mathbf{X}^{(i)}, \mathbf{A}^{(i)}, \mathbf{Y}^{(i)})\}$ generated from $p(X, A, Y|E = E_{tr})$, the generalization error on unseen data $(\mathbf{X}', \mathbf{A}', \mathbf{Y}')$ from a new environment $E_{te} \neq E_{tr}$ would have an upper bound that holds with the confidence $1 - \delta$:*

$$
\mathcal{R}(\Gamma_\theta; E_{te}) \leq \mathcal{R}_{emp}(\Gamma_\theta; E_{tr}) + \mathcal{D}_1(\Gamma, N_{tr}) + \mathcal{D}_2(E_{tr}, E_{te}, W), \quad (77)
$$

*where $\mathcal{D}_2(E_{tr}, E_{te}, W) = \mathcal{O}(\mathbb{E}_{\mathbf{A} \sim p(A|E_{tr}), \mathbf{A}' \sim p(A|E_{te})}[\psi(\|\Delta\mathbf{A}\|_2)]).$*

*Proof.* According to the data generation in Fig. 1, for given node latents $U_u$'s, we can decompose the joint distribution into

$$
p(X, A, Y|E) = p(X|E)p(A|E)p(Y|A, E). \quad (78)
$$

Also, by definition in Sec. 3.1 we have

$$
p(X|E = E_{tr}) = p(X|E = E_{te}), \quad (79)
$$

$$
p(Y|A, E = E_{tr}) = p(Y|A, E = E_{te}). \quad (80)
$$

We next consider the gap between $\mathcal{R}(\Gamma_\theta; E_{tr})$ and $\mathcal{R}(\Gamma_\theta; E_{te})$:

$$
\begin{aligned}
&|\mathcal{R}(\Gamma_\theta; E_{te}) - \mathcal{R}(\Gamma_\theta; E_{tr})| \\
=& \left| \mathbb{E}_{(\mathbf{X}', \mathbf{A}', \mathbf{Y}') \sim p(X, A, Y|E=E_{te})}[l(\Gamma_\theta(\mathbf{X}', \mathbf{A}'), \mathbf{Y}')] - \mathbb{E}_{(\mathbf{X}, \mathbf{A}, \mathbf{Y}) \sim p(X, A, Y|E=E_{tr})}[l(\Gamma_\theta(\mathbf{X}, \mathbf{A}), \mathbf{Y})] \right| \\
=& \left| \mathbb{E}_{\mathbf{X}' \sim p(X|E_{te}), \mathbf{A}' \sim p(A|E_{te}), \mathbf{Y}' \sim p(Y|A=\mathbf{A}', E_{te}))}[l(\Gamma_\theta(\mathbf{X}', \mathbf{A}'), \mathbf{Y}')] \right. \\
& \left. - \mathbb{E}_{\mathbf{X} \sim p(X|E_{tr}), \mathbf{A} \sim p(A|E_{tr}), \mathbf{Y} \sim p(Y|A=\mathbf{A}, E_{tr}))}[l(\Gamma_\theta(\mathbf{X}, \mathbf{A}), \mathbf{Y})] \right| \\
\leq& \left| \mathbb{E}_{\mathbf{X}' \sim p(X|E_{te}), \mathbf{A}' \sim p(A|E_{te}), \mathbf{Y}' \sim p(Y|A=\mathbf{A}', E_{te}))}[l(\Gamma_\theta(\mathbf{X}', \mathbf{A}'), \mathbf{Y}')] \right. \\
& \left. - \mathbb{E}_{\mathbf{X} \sim p(X|E_{tr}), \mathbf{A} \sim p(A|E_{tr}), \mathbf{Y}' \sim p(Y|A=\mathbf{A}', E_{tr}))}[l(\Gamma_\theta(\mathbf{X}, \mathbf{A}), \mathbf{Y}')] \right| \quad (*) \\
=& \left| \mathbb{E}_{\mathbf{X}' \sim p(X|E_{te}), \mathbf{A}' \sim p(A|E_{te}), \mathbf{Y}' \sim p(Y|A=\mathbf{A}', E_{te}))}[l(\Gamma_\theta(\mathbf{X}', \mathbf{A}'), \mathbf{Y}')] \right. \\
& \left. - \mathbb{E}_{\mathbf{X}' \sim p(X|E_{te}), \mathbf{A} \sim p(A|E_{tr}), \mathbf{Y}' \sim p(Y|A=\mathbf{A}', E_{tr}))}[l(\Gamma_\theta(\mathbf{X}, \mathbf{A}), \mathbf{Y}')] \right| \\
=& \left| \mathbb{E}_{\mathbf{X}' \sim p(X|E_{te}), \mathbf{A}' \sim p(A|E_{te}), \mathbf{Y}' \sim p(Y|A=\mathbf{A}', E_{te}))}[l(\Gamma_\theta(\mathbf{X}', \mathbf{A}'), \mathbf{Y})] \right. \\
& \left. - \mathbb{E}_{\mathbf{X}' \sim p(X|E_{te}), \mathbf{A} \sim p(A|E_{tr}), \mathbf{Y}' \sim p(Y|A=\mathbf{A}', E_{te}))}[l(\Gamma_\theta(\mathbf{X}', \mathbf{A}), \mathbf{Y}')] \right| \\
=& \left| \mathbb{E}_{\mathbf{X}' \sim p(X|E_{te}), \mathbf{A} \sim p(A|E_{tr}), \mathbf{A}' \sim p(A|E_{te}), \mathbf{Y}' \sim p(Y|A=\mathbf{A}', E_{te}))}[l(\Gamma_\theta(\mathbf{X}', \mathbf{A}'), \mathbf{Y}') - l(\Gamma_\theta(\mathbf{X}', \mathbf{A}), \mathbf{Y}')] \right| \\
\leq& \, \mathbb{E}_{\mathbf{X}' \sim p(X|E_{te}), \mathbf{A} \sim p(A|E_{tr}), \mathbf{A}' \sim p(A|E_{te}), \mathbf{Y}' \sim p(Y|A=\mathbf{A}', E_{te}))} \left[ |l(\Gamma_\theta(\mathbf{X}', \mathbf{A}'), \mathbf{Y}') - l(\Gamma_\theta(\mathbf{X}', \mathbf{A}), \mathbf{Y}')| \right],
\end{aligned}
$$
$$(81)$$

where the step with $(*)$ follows the observation that for a well-trained model $\Gamma_\theta$, $(\mathbf{X}, \mathbf{A}, \mathbf{Y}) \sim p(X, A, Y|E = E_{tr})$ and $\mathbf{Y}' \neq \mathbf{Y}$, it almost surely holds that $l(\Gamma_\theta(\mathbf{X}, \mathbf{A}), \mathbf{Y}) \leq l(\Gamma_\theta(\mathbf{X}, \mathbf{A}), \mathbf{Y}')$.

Moreover, due to the Lipschitz continuity of $l$ and $\phi_{dec}$, we have

$$
|l(\Gamma_\theta(\mathbf{X}', \mathbf{A}'), \mathbf{Y}') - l(\Gamma_\theta(\mathbf{X}', \mathbf{A}), \mathbf{Y}')| \leq L \cdot \|\mathbf{Z}(T; \mathbf{A}') - \mathbf{Z}(T; \mathbf{A})\|_2. \quad (82)
$$

Combing Eqn. 81, Eqn. 82 and the result of Theorem 1, we have

$$
|\mathcal{R}(\Gamma_\theta; E_{te}) - \mathcal{R}(\Gamma_\theta; E_{tr})| \leq LC \cdot \mathbb{E}_{\mathbf{A} \sim p(A|E_{tr}), \mathbf{A}' \sim p(A|E_{te})} \left[ \|\Delta\tilde{\mathbf{A}}\|_2^{mT} \right]. \quad (83)
$$

where $C$ is a constant. Moreover, we have the relationship between the empirical risk $\mathcal{R}_{emb}(\Gamma_\theta; E_{tr})$ and the in-distribution generalization error $\mathcal{R}(\Gamma_\theta; E_{tr})$

$$|\mathcal{R}_{emb}(\Gamma_\theta; E_{tr}) - \mathcal{R}(\Gamma_\theta; E_{tr})| \leq 2\mathcal{H}(\Gamma) + \mathcal{O}\left(\sqrt{\frac{\log(1/\delta)}{N_{tr}}}\right). \tag{84}$$

The conclusion for the main theorem can be obtained via combining Eqn. 83 and 84 using the triangle inequality. □

## D    APPROXIMATION STRATEGIES FOR DIFFUSION PDE SOLUTIONS

The closed-form solutions of linear diffusion equations often involve the form of matrix exponential $e^{-\mathbf{L}t}$, which is intractable for computing its exact value. There are many established techniques based on numerical approximations, e.g., series expansion, in this fundamental challenge. In our presented model in Sec. 4.3, we propose two implementation versions based on two approximation ways for handling the closed-form solution of the advective diffusion equations on graphs.

**Approximation with Linear Systems.** One scalable scheme proposed by Gallopoulos & Saad (1992) is via the extension of the minimax Padé-Chebyshev theory to rational fractions (Golub & Van Loan, 1989). This approximation technique has been utilized by Patané (2014) as an effective and efficient method for spectrum-free computation of the diffusion distances in 3D shape analysis. In specific, the matrix exponential of the form $e^{-\mathbf{L}t}$ is approximated by the combination of multiple matrix inverses:

$$\exp(-\mathbf{L}t) \approx -\sum_{i=1}^{r} \alpha_i (\mathbf{L} + \theta_i \mathbf{I})^{-1}, \tag{85}$$

where $\alpha_i$ and $\theta_i$ can be pre-defined parameters Gallopoulos & Saad (1992). To unleash the capacity of neural networks, in Sec. 4.3, our model implementation (ADIT-INVERSE) extends this scheme to a multi-head network where each head contributes to propagation with independently parameterized attention networks. The matrix inverse is computed with the linear system solver that is available in common deep learning tools (e.g., PyTorch) and supports automatic differentiation.

**Approximation with Geometric Series.** When the graph sizes become large, the matrix inverse can be computationally expensive. For better scalability, we can use the geometric series for approximation:

$$(\mathbf{L} + \theta_i \mathbf{I})^{-1} = \sum_{k=0}^{\infty} (-1)^k \theta_i^{-(k+1)} \mathbf{L}^k \approx \sum_{k=0}^{K} (-1)^k \theta_i^{-(k+1)} \mathbf{L}^k. \tag{86}$$

In this way, the matrix exponential can be approximately computed via a combination of finite series:

$$\exp(-\mathbf{L}t) \approx -\sum_{i=1}^{r} \alpha_i \sum_{k=0}^{K} (-1)^k \theta_i^{-(k+1)} \mathbf{L}^k. \tag{87}$$

In our model, the closed-form solution for the PDE induces $\mathbf{L} = (\mathbf{I} - \mathbf{C} - \beta\tilde{\mathbf{A}})$, and the summation in Eqn. 87 can be expressed as a weighted sum of $\mathbf{P}^k = (\mathbf{C} + \beta\tilde{\mathbf{A}})^k$ for $k = 0, \cdots, K$. Our model implementation (ADIT-SERIES) proposed in Sec. 4.3 generalizes the weighted sum to a one-layer neural network.

## E    MODEL IMPLEMENTATIONS AND ALGORITHMS

In this section, we provide detailed and self-contained descriptions about our model architectures in Appendix E.1. Then in Appendix E.2, we discuss how to apply our model to various graph-structured data with additional input information. To make the presentation clear and focused on the model implementation side, we will re-define some notations that are originally defined in Sec. 4, where we formulate the model with the terminology of the PDE domain.

### E.1 MODEL ARCHITECTURES

The model takes a graph $\mathcal{G} = (\mathcal{V}, \mathcal{E}, \mathbf{X}, \mathbf{A})$ as input, and output prediction in the downstream tasks. We assume the number of nodes in the graph $|\mathcal{V}| = N$, node feature matrix $\mathbf{X} \in \mathbb{R}^{N \times D}$ and graph adjacency matrix $\mathbf{A} \in \{0, 1\}^{N \times N}$. We use $\mathbf{D}$ to denote the diagonal degree matrix of $\mathbf{A}$. The normalized adjacency is denoted by $\tilde{\mathbf{A}}$, and $\mathbf{1}$ is an all-one $N$-dimensional column vector. In this subsection, we assume $\mathcal{G}$ has no edge weight or edge feature for presentation, and with loss of generality, we will discuss how to incorporate these additional attributes in Appendix E.2.

### E.1.1 INSTANTIATIONS AND PARAMETERIZATIONS

Our model is comprised of three modules: the encoder $\phi_{enc}$, the decoder $\phi_{dec}$, and the propagation network in-between the first two.

**Encoder:** The node features $\mathbf{X} = [\mathbf{x}_u]_{u \in \mathcal{V}} \in \mathbb{R}^{N \times D}$ are first mapped to embeddings in the latent space $\mathbf{Z}^{(0)} = [\mathbf{z}_u^{(0)}]_{u \in \mathcal{V}} \in \mathbb{R}^{N \times d}$ via the encoder: $\mathbf{Z}^{(0)} = \phi_{enc}(\mathbf{X})$. The encoder $\phi_{enc}(\cdot)$ is instantiated as a shallow MLP with non-linear activation (e.g., ReLU).

**Propagation:** The propagation network converts the initial node embeddings $\mathbf{Z}^{(0)}$ to the node representations $\mathbf{Z} = [\mathbf{z}_u]_{u \in \mathcal{V}} \in \mathbb{R}^{N \times d}$ (where $\mathbf{Z}^{(0)}$ and $\mathbf{Z}$ are the re-defined counterparts of $\mathbf{Z}(0)$ and $\mathbf{Z}(T)$, respectively, presented in Sec. 4). The propagation network is implemented via a multi-head network with $H$ heads involving the attention network $\eta^{(h)}(\cdot, \cdot)$ and feature transformation network $\phi_{FC}^{(h)}(\cdot)$. The latter is instantiated as a fully-connected layer $\mathbf{W}_{O,h}$, and the attention network is instantiated as a normalized dot-product positive similarity function:

$$\eta^{(h)}(\mathbf{z}_u^{(0)}, \mathbf{z}_v^{(0)}) = 1 + \left( \frac{\mathbf{W}_{Q,h} \mathbf{z}_u^{(0)}}{\|\mathbf{W}_{Q,h} \mathbf{z}_u^{(0)}\|_2} \right)^\top \left( \frac{\mathbf{W}_{K,h} \mathbf{z}_v^{(0)}}{\|\mathbf{W}_{K,h} \mathbf{z}_v^{(0)}\|_2} \right),$$
$$\mathbf{C}_h = \{c_{uv}^{(h)}\}, \quad c_{uv}^{(h)} = \frac{\eta^{(h)}(\mathbf{z}_u^{(0)}, \mathbf{z}_v^{(0)})}{\sum_{w \in \mathcal{V}} \eta^{(h)}(\mathbf{z}_u^{(0)}, \mathbf{z}_w^{(0)})}, \tag{88}$$

where $\mathbf{W}_{Q,h} \in \mathbb{R}^{d \times d}$ and $\mathbf{W}_{K,h} \in \mathbb{R}^{d \times d}$ are trainable weights for query and key, respectively, of the $h$-th head. Then the node representations will be computed in different ways by two models.

- For ADIT-INVERSE, the node representations are calculated via

$$\mathbf{L}_h = (1 + \theta)\mathbf{I} - \mathbf{C}_h - \beta \tilde{\mathbf{A}},$$
$$\mathbf{Z}_h = \text{linsolver}(\mathbf{L}_h, \mathbf{Z}^{(0)}),$$
$$\mathbf{Z} = \sum_{h=1}^{H} \mathbf{Z}_h \mathbf{W}_{O,h}, \tag{89}$$

  where $\mathbf{W}_{O,h} \in \mathbb{R}^{d \times d}$. Alg. 1 summarizes the feed-forward computation of ADIT-INVERSE.

- For ADIT-SERIES, the node representations are computed by

$$\mathbf{P}_h = \mathbf{C}_h + \beta \tilde{\mathbf{A}},$$
$$\mathbf{Z}^{(k)} = \mathbf{P}_h \mathbf{Z}^{(k-1)}, \text{ for } k = 1, \cdots K,$$
$$\mathbf{Z} = \sum_{h=1}^{H} [\mathbf{Z}^{(0)}, \mathbf{Z}^{(1)}, \cdots, \mathbf{Z}^{(K)}] \mathbf{W}_{O,h}, \tag{90}$$

  where $\mathbf{W}_{O,h} \in \mathbb{R}^{(K+1)d \times d}$. To accelerate the computation of Eqn. 90, we can inherit the strategy used in Wu et al. (2023) and alter the order of matrix products, which reduces the time and space complexity to $\mathcal{O}(N)$ (see Appendix E.1.2 for detailed illustration). Alg. 2 presents the feed-forward computation of ADIT-SERIES that only requires $\mathcal{O}(N)$ algorithmic complexity.

---

**Algorithm 1** Feed-Forward of the Model ADiT-INVERSE.

---

**INPUT:** Node feature matrix $\mathbf{X}$ and normalized adjacency matrix $\tilde{\mathbf{A}}$.
$\mathbf{Z}^{(0)} = \phi_{enc}(\mathbf{X})$
**for** $h = 1, \cdots, H$ **do**

$\quad \mathbf{Z}_{Q,h} = \left[ \frac{\mathbf{W}_{Q,h}\mathbf{z}_u^{(0)}}{\|\mathbf{W}_{Q,h}\mathbf{z}_u^{(0)}\|_2} \right]_{u \in \mathcal{V}}, \quad \mathbf{Z}_{K,h} = \left[ \frac{\mathbf{W}_{K,h}\mathbf{z}_u^{(0)}}{\|\mathbf{W}_{K,h}\mathbf{z}_u^{(0)}\|_2} \right]_{u \in \mathcal{V}}$

$\quad \mathbf{U}_h = \mathbf{1}\mathbf{1}^\top + \mathbf{Z}_{Q,h}(\mathbf{Z}_{K,h})^\top$

$\quad \mathbf{C}_h = \text{diag}^{-1}(\mathbf{U}_h\mathbf{1})\,\mathbf{U}_h$

$\quad \mathbf{L}_h = (1+\theta)\mathbf{I} - \mathbf{S}_h - \beta\tilde{\mathbf{A}}$

$\quad \mathbf{Z}_h = \text{linsolver}(\mathbf{L}_h, \mathbf{Z})$

$\mathbf{Z} = \sum_{h=1}^H \mathbf{Z}_h\mathbf{W}_{O,h}$
**OUTPUT:** Node representations $\mathbf{Z}$ and predicted labels with $\phi_{dec}(\mathbf{Z})$.

---

**Decoder:** The decoder $\phi_{dec}(\cdot)$ transforms the node representations into prediction. Depending on the specific downstream tasks, the decoder can be implemented in different ways:

$$\begin{aligned}
&\text{(node-level prediction):} \quad \hat{y}_u = \text{MLP}(\mathbf{z}_u) \\
&\text{(graph-level prediction):} \quad \hat{y} = \text{MLP}(\text{SumPooling}(\{\mathbf{z}_u\}_{u \in \mathcal{V}})) \\
&\text{(edge-level prediction):} \quad \hat{y}_{uv} = \text{MLP}([\mathbf{z}_u, \mathbf{z}_v]).
\end{aligned} \tag{91}$$

In particular, the softmax activation is used for output in classification tasks. For training, we adopt standard loss functions, i.e., cross-entropy for classification and mean square loss for regression.

### E.1.2 ACCELERATION OF ADiT-SERIES WITH LINEAR COMPLEXITY

We illustrate how to achieve the propagation of ADiT-SERIES in Eqn. 90 with $\mathcal{O}(N)$ complexity. With the query and key matrices defined by $\mathbf{Z}_{Q,h} = \left[ \frac{\mathbf{W}_{Q,h}\mathbf{z}_u^{(0)}}{\|\mathbf{W}_{Q,h}\mathbf{z}_u^{(0)}\|_2} \right]_{u \in \mathcal{V}}$ and $\mathbf{Z}_{K,h} = \left[ \frac{\mathbf{W}_{K,h}\mathbf{z}_u^{(0)}}{\|\mathbf{W}_{K,h}\mathbf{z}_u^{(0)}\|_2} \right]_{u \in \mathcal{V}}$, the attention matrix $\mathbf{C}_h$ in Eqn. 88 is computed by (in the matrix form used for implementation)

$$\mathbf{C}_h = \text{diag}^{-1}\left(N + \mathbf{Z}_{Q,h}(\mathbf{Z}_{K,h})^\top\mathbf{1}\right)\left(\mathbf{1}\mathbf{1}^\top + \mathbf{Z}_{Q,h}(\mathbf{Z}_{K,h})^\top\right). \tag{92}$$

Computing the above result requires $\mathcal{O}(N^2)$ time and space complexity. Still, if we consider the feature propagation with $\mathbf{C}_h$, we have

$$\begin{aligned}
\mathbf{C}_h\mathbf{Z}_h^{(k)} &= \text{diag}^{-1}\left(N + \mathbf{Z}_{Q,h}(\mathbf{Z}_{K,h})^\top\mathbf{1}\right) \cdot \left(\mathbf{1}\mathbf{1}^\top + \mathbf{Z}_{Q,h}(\mathbf{Z}_{K,h})^\top\right) \cdot \mathbf{Z}_h^{(k)} \\
&= \text{diag}^{-1}\left(N + \mathbf{Z}_{Q,h}((\mathbf{Z}_{K,h})^\top\mathbf{1})\right) \cdot \left[\mathbf{1}\left(\mathbf{1}^\top\mathbf{Z}_h^{(k)}\right) + \mathbf{Z}_{Q,h}\left((\mathbf{Z}_{K,h})^\top\mathbf{Z}_h^{(k)}\right)\right],
\end{aligned} \tag{93}$$

where the equality is achieved by altering the order of matrix products. The above computation only requires $\mathcal{O}(N)$ time and space complexity. The feed-forward computation of ADiT-SERIES with $\mathcal{O}(N)$ acceleration is summarized in Alg. 2.

### E.2 APPLICABILITY OF OUR MODEL

In the main paper, we assume unweighted graphs without edge attribute features for model formulation. Without loss of generality, we next discuss how to extend our model to handle the edge weights and edge features.

**Edge Weights.** For weighted graphs, the adjacency matrix $\mathbf{A}$ would become a real matrix where the entry $a_{uv}$ denotes the weight on the edge $(u,v) \in \mathcal{E}$. In this situation, we still have the corresponding normalized adjacency $\tilde{\mathbf{A}} = \mathbf{D}^{-1}\mathbf{A}$ or $\tilde{\mathbf{A}} = \mathbf{D}^{-1/2}\mathbf{A}\mathbf{D}^{-1/2}$, where $\mathbf{D} = \text{diag}([d_u]_{u \in \mathcal{V}})$ and $d_u = \sum_{v,(u,v) \in \mathcal{E}} a_{uv}$. Our model implementations can be trivially generalized to this case by using $\tilde{\mathbf{A}}$ as the propagation matrix for local message passing.

**Edge Features.** If the graph contains edge features, denoted by $\mathbf{E} = [\mathbf{e}_{uv}]_{(u,v) \in \mathcal{E}} \in \mathbb{R}^{|\mathcal{E}| \times D'}$, we introduce an encoding layer $\mathbf{W}_E \in \mathbb{R}^{D' \times d}$ for mapping the edge features into embeddings in the

---

**Algorithm 2** Feed-Forward of the Model ADIT-SERIES (with $\mathcal{O}(N)$ complexity).

---

**INPUT:** Node feature matrix $\mathbf{X}$ and normalized adjacency matrix $\tilde{\mathbf{A}}$.
$\mathbf{Z}^{(0)} = \phi_{enc}(\mathbf{X})$
**for** $h = 1, \cdots, H$ **do**

$\quad \mathbf{Z}_{Q,h} = \left[ \dfrac{\mathbf{W}_{Q,h}\mathbf{z}_u^{(0)}}{\|\mathbf{W}_{Q,h}\mathbf{z}_u^{(0)}\|_2} \right]_{u \in \mathcal{V}}, \quad \mathbf{Z}_{K,h} = \left[ \dfrac{\mathbf{W}_{K,h}\mathbf{z}_u^{(0)}}{\|\mathbf{W}_{K,h}\mathbf{z}_u^{(0)}\|_2} \right]_{u \in \mathcal{V}}$

$\quad \mathbf{N}_h = \text{diag}^{-1}\left( N + \mathbf{Z}_{Q,h}\left((\mathbf{Z}_{K,h})^\top \mathbf{1}\right)\right)$

$\quad \mathbf{Z}_h^{(0)} = \mathbf{Z}^{(0)}$

$\quad$ **for** $k = 1, \cdots, K$ **do**

$\quad\quad \mathbf{Z}_h^{(k)} = \mathbf{N}_h \cdot \left[ \mathbf{1}\left(\mathbf{1}^\top \mathbf{Z}_h^{(k-1)}\right) + \mathbf{Z}_{Q,h}\left((\mathbf{Z}_{K,h})^\top \mathbf{Z}_h^{(k-1)}\right)\right] + \beta\tilde{\mathbf{A}}\mathbf{Z}_h^{(k-1)}$

$\quad \mathbf{Z}_h = [\mathbf{Z}_h^{0)}, \mathbf{Z}_h^{(1)}, \cdots, \mathbf{Z}_h^{(K)}]$

$\mathbf{Z} = \sum_{h=1}^H \mathbf{Z}_h \mathbf{W}_{O,h}$
**OUTPUT:** Node representations $\mathbf{Z}$ and predicted labels with $\phi_{dec}(\mathbf{Z})$.

---

latent space and then incorporate them with node embeddings. In specific, we first compute the edge-to-node signals:

$$\mathbf{M} = [\mathbf{m}_u]_{u \in \mathcal{V}}, \quad \mathbf{m}_u = \sum_{v,(u,v) \in \mathcal{E}} \tilde{\mathbf{A}}_{u,v} \mathbf{W}_E \mathbf{e}_{uv}. \tag{94}$$

- For ADIT-INVERSE, we can modify Eqn. 89 as

$$
\begin{aligned}
\mathbf{L}_h &= (1+\theta)\mathbf{I} - \mathbf{C}_h - \beta\tilde{\mathbf{A}}, \\
\mathbf{Z}_h &= \text{linsolver}\left(\mathbf{L}_h, (\mathbf{Z}^{(0)} + \mathbf{M})\right), \\
\mathbf{Z} &= \sum_{h=1}^H \mathbf{Z}_h \mathbf{W}_{O,h}.
\end{aligned}
\tag{95}
$$

- For ADIT-SERIES, we can modify Eqn. 90 to be

$$
\begin{aligned}
\mathbf{P}_h &= \mathbf{C}_h + \beta\tilde{\mathbf{A}}, \\
\mathbf{Z}^{(k)} &= \mathbf{P}_h(\mathbf{Z}^{(k-1)} + \mathbf{M}), \quad k = 1, \cdots K, \\
\mathbf{Z} &= \sum_{h=1}^H [\mathbf{Z}^{(0)}, \mathbf{Z}^{(1)}, \cdots, \mathbf{Z}^{(K)}]\mathbf{W}_{O,h},
\end{aligned}
\tag{96}
$$

## F EXPERIMENT DETAILS

We supplement details for our experiments, regarding datasets, competitors, and implementations, for facilitating the reproducibility.

### F.1 DATASETS

The datasets we use for the experiments in Sec. 5 span diverse domains and learning tasks. We summarize the statistics and brief descriptions for each dataset in Table 4, with the detailed information presented in the following subsections.

#### F.1.1 SYNTHETIC DATASETS

The synthetic datasets used in Sec. 5.1 simulate the graph data generation in Sec. 3.1, where the topological distribution shifts are caused by the difference of environments across training and testing data. In specific, we generate graphs of $|\mathcal{V}| = 1000$ nodes, with the node features $\mathbf{X}$, graph adjacency matrix $\mathbf{A}$ and labels $\mathbf{Y}$ generated by the following process.

Table 4: Statistics and descriptions for experimental datasets.

| Dataset | #Nodes | #Edges | #Graphs | Train/Val/Test Split | Task | Metric |
|---|---|---|---|---|---|---|
| Synthetic-h | 1,000 | 14,064 - 32,066 | 12 | SBM (Homophily) | Node Regression | RMSE |
| Synthetic-d | 1,000 | 7,785 - 13,912 | 12 | SBM (Density) | Node Regression | RMSE |
| Synthetic-b | 1,000 | 14,073 - 59,936 | 12 | SBM (Block Number) | Node Regression | RMSE |
| Twitch | 1,912 - 9,498 | 31,299 - 153,138 | 7 | Geographic Domain | Node Classification | ROC-AUC |
| Arxiv | 169,343 | 1,166,243 | 1 | Publication Time | Node Classification | Accuracy |
| OGB-BACE | 10 - 97 | 10 - 101 | 1,513 | Molecular Scaffold | Graph Classification | ROC-AUC |
| OGB-SIDER | 1 - 492 | 0 - 505 | 1,427 | Molecular Scaffold | Graph Classification | ROC-AUC |
| DDPIN-nr | 143 - 5,003 | 22 - 25,924 | 12 | Protein Identification Method | Node Regression | RMSE |
| DDPIN-er | 143 - 5,003 | 22 - 25,924 | 12 | Protein Identification Method | Edge Regression | RMSE |
| DDPIN-lp | 143 - 5,003 | 22 - 25,924 | 12 | Protein Identification Method | Link Prediction | ROC-AUC |
| HAM | 8 - 25 | 7 - 29 | 1,987 | Relative Molecular Mass | Edge Classification | Accuracy |

- Each node $u \in \mathcal{V}$ is assigned with a scalar $u_u$ randomly sampled from the uniform distribution $U[0, 1]$.

- For the generation of node features $\mathbf{X} = [\mathbf{x}_u]_{u \in \mathcal{V}}$, we instantiate the node-wise function $g$ as a 2-layer MLP with ReLU activation and 4-dimensional output. Then the node feature $\mathbf{x}_u$ is generated through $\mathbf{x}_u = \text{MLP}(u_u)$.

- For the generation of graph adjacency $\mathbf{A} = [a_{uv}]_{u,v \in \mathcal{V}}$, we instantiate the pairwise function $h$ as the stochastic block model (Snijders & Nowicki, 1997) which generates edges according to the intra-block edge probability ($p_1$) and the inter-block edge probability ($p_2$). We map the nodes into $b$ blocks by the following rule: for node $u \in \mathcal{V}$, we assign it to the $k$-th block if $v_u \in [\frac{k-1}{b}, \frac{k}{b})$ (where $1 \leq k \leq b$). Then the edge $a_{uv}$ is randomly generated from a bernoulli distribution with $p_1$ if $u$ and $v$ are in the same block, and $p_2$ otherwise.

- For the generation of labels $\mathbf{Y}$, we consider the regression tasks and each node has a label $y_u$ generated through an ensemble model of a 2-layer GCN and a 1-layer DIFFormer (without using the graph-based propagation) with random initializations: $\mathbf{Y} = \text{gcn}(\mathbf{U}, \mathbf{A}) + \text{difformer}(\mathbf{U}, \mathbf{A})$, where $\mathbf{U} = [u_u]_{u \in \mathcal{V}}$.

Using the above data generation, we create 12 graphs with the indices #1∼ #12, and use the graph #1 for training, the graph #2 for validation, and the graphs #3∼ #12 for testing. The topological distribution shifts are introduced in three different ways as described in Sec. 5.1, where in each case, the detailed configurations for $p_1$, $p_2$ and $b$ are illustrated below.

- *Homophily Shift*: $p_1 = 0.1$, $b = 5$ and $p_2 = 0.01 + 0.05 * \frac{1}{12} * (i - 1)$ for the graph #$i$.

- *Density Shift*: $b = 5$, $p_1 = 0.1 + 0.1 * \frac{1}{12} * (i - 1)$ and $p_2 = 0.01 + 0.1 * \frac{1}{12} * (i - 1)$ for the graph #$i$.

- *Block Shift*: $p_1 = 0.1$, $p_2 = 0.01$ and $b = 5 + (i - 1)$ for the graph #$i$.

### F.1.2 INFORMATION NETWORKS

The citation network Arxiv provided by Hu et al. (2020) consists of a single graph with 0.16M nodes, where each node represents a paper with the publication year (ranging from 1960 to 2020) and a subarea id (from 40 different subareas in total). The node attribute features are 128-dimensional obtained by averaging the word embeddings of the paper's title and abstract. The edges are given by the citation relationship between papers. The predictive task is to estimate the paper's subarea. We use the publication years to split the data: papers published before 2014 for training, within the range from 2014 to 2017 for validation, and on 2018/2019/2020 for testing. Since there is a single graph, to increase the difficulty of generalization, we consider the inductive setting: the testing nodes are not contained in the training graph. Table 6 demonstrates the dissimilar statistics for training/validation/testing graphs, manifesting the existence of topological shifts. Following the common practice, we use Accuracy as the evaluation metric.

Twitch (Rozemberczki et al., 2021) is comprised of seven dis-connected graphs, where each node represents a Twitch user and edges indicate the friendship. Each graph is collected from the social newtork in a particular region, including DE, ENGB, ES, FR, PTBR, RU and TW. The node features

Table 5: Statistics for training/validation/testing graphs on `Arxiv`. There is a single citation network that augments with time evolving, and with the data splits in the inductive setting, the previous graph is contained by the subsequent one.

| | Train (1960-2014) | Valid (2015-2017) | Test 1 (2018) | Test 2 (2019) | Test 3 (2020) |
|---|---|---|---|---|---|
| # Target Nodes | 41,125 | 49,816 | 29,799 | 39,711 | 8,892 |
| # All Nodes | 41,125 | 90,941 | 120,740 | 160,451 | 169,343 |
| # All Edges | 102,316 | 374,839 | 622,466 | 1,061,197 | 1,166,243 |
| Max Degrees | 275 | 3,036 | 6,251 | 12,006 | 13,161 |
| Avg Degrees | 4.98 | 8.24 | 10.31 | 13.23 | 13.77 |

are multi-hot with 2,545 dimensions indicating the user's profile. The predictive task is to classify the gender of the user. The seven networks with sizes ranging from 2K to 9K have distinct structural characteristics (such as densities and maximum degrees) as observed by Wu et al. (2022a). We therefore split the data according to the geographic information: use the network DE for training, ENGB for validation, and the remaining networks for testing. The evaluation metric is ROC-AUC for binary classification.

### F.1.3 MOLECULAR PROPERTY PREDICTION

The molecule datasets `OGB-BACE` and `OGB-SIDER` are released by Hu et al. (2020) and the task is to identify the target molecular properties. Each dataset contains a collection of molecules, where each molecule is a graph with dozens of atom nodes and bond edges. The node features are 9-dimensional, composed of the atom's characteristics (e.g., atomic number, chirality, normal charge, etc.). Additionally, each bond edge has a 3-dimensional feature indicating the bond type, bond stereochemistry and whether the bond is conjugated.

We follow the pre-processed steps used by Hu et al. (2020) and adopt its released public splits, which partition the molecules of each dataset into training/validation/testing ones with the ratio 80%/10%/10% based on molecular scaffolds. The latter guarantees that the molecules in the testing set are maximally different from the training set w.r.t. structural characteristics, allowing us to evaluate the model's performance on the challenging topological generalization. We follow the common practice using ROC-AUC for performance evaluation.

### F.1.4 BIOLOGICAL PROTEIN INTERACTIONS

`DDPIN` (Fu & He, 2022) contains 12 individual dynamic network datasets at different scales, and each dataset is a dynamic protein-protein interaction network that describes the protein-level interactions of yeast cells. Each graph dataset is obtained by one protein identification method and consists of 36 graph snapshots, wherein each node denotes a protein that has a sequence of 1-dimensional continuous features with 36 time stamps. This records the evolution of gene expression values within metabolic cycles of yeast cells. The edges in the graph are determined by co-expressed protein pairs at one time, and each edge is associated with a co-expression correlation coefficient.

We consider the predictive tasks within each graph snapshot and ignore the temporal evolution between different snapshots. In specific, we use the graph topology of each snapshot as the observed graph adjacency $\mathbf{A}$ and use the gene expression values at the previous 10 time steps as node features $\mathbf{X}$. On top of this, we consider three different predictive tasks: 1) node regression for gene expression value at the current time (measured by RMSE); 2) edge regression for predicting the co-expression correlation coefficient (measured by RMSE); 3) link prediction for identifying co-expressed protein pairs (measured by ROC-AUC). Given the fact that each graph dataset has distinct sizes (ranging from 143 to 5,003 nodes) and distributions of 3-cliques and 4-cliques (ranging from 0 to hundreds) (Fu & He, 2022), we consider the dataset-level data splitting and use 6/1/5 graph datasets for training/validation/testing, which introduces topological distribution shifts.

### F.1.5 MOLECULAR MAPPING OPERATOR GENERATION

The *Human Annotated Mappings* (`HAM`) dataset (Li et al., 2020) consists of 1,206 molecules with expert annotated mapping operators, i.e., a representation of how atoms are grouped in a molecule. The latter segments the atoms of a molecule into groups of varying sizes. As an important step in

molecular dynamics simulation, generating coarse-grained mapping operators aims to reproduce the mapping operators produced by experts. This task can be modeled as a graph segmentation problem (Li et al., 2020) which takes a molecule graph as input and outputs the labels for each edge that indicates if there is cut needed to partition the source and end atoms into different groups.

For data splits, we calculate the relative molecular mass of each molecule using the RDKit package[3], and rank the molecules with increasing mass. Then we use the first 70% molecules for training, the following 15% for validation, and the remaining for testing. This splitting protocol partitions molecules with different weights, and requires generalization from small molecules in the training set to larger molecules in the testing set.

Table 6: The range of relative molecular mass for training/validation/testing molecules in `HAM`.

|  | Train | Valid | Test |
|---|---|---|---|
| Relative Molecular Mass | $108.18 \sim 273.34$ | $273.34 \sim 311.14$ | $311.14 \sim 762.94$ |

## F.2 COMPETITORS

In our experiments, we compare with peer encoder backbones for graph learning tasks. The competitors span three aspects: 1) classical GNNs, 2) diffusion-based GNNs, and 3) graph Transformers. We briefly introduce the competitors and illuminate their connections with our model.

- **GCN** (Kipf & Welling, 2017) is a popular model that propagates node embeddings over observed graphs for computing node representations, which can be seen as the discretized version of graph diffusion equations with feature transformations. While the model architecture is simple, GCN is still a strong competitor in graph learning tasks, as shown by quite a few recent empirical studies.

- **GAT** (Velickovic et al., 2018) introduces attention networks for computing pairwise weights for neighboring nodes in the graph and propagates node signals with adaptive strengths given by the attention weights. GAT can be seen as the discretized version of the non-linear graph diffusion equation.

- **SGC** (Wu et al., 2019) proposes to simplify the GCN architecture by removing the feature transformations in-between propagation layers, reducing multi-layer propagation to one-layer. This brings up significant acceleration for training and inference. SGC can be seen as the discretization of the linear diffusion equation on graphs.

- **GDC** (Klicpera et al., 2019) extends the graph convolution operator to graph diffusion convolution derived from the linear diffusion equation on graphs. We use its implementation version based on the heat kernel for diffusion coefficients.

- **GRAND** (Chamberlain et al., 2021a) proposes graph neural diffusion, a continuous PDE model, that generalizes manifold diffusion to graphs and then uses numerical schemes to solve the PDE. We compare with its linear version that implements the linear graph diffusion equation.

- **GraphTrans** (Wu et al., 2021) is a recently proposed Transformer for graph-structured data that satisfies the permutation-invariant property. The model architecture sequentially combines GNNs and Transformers in order, where the GNN can learn local, short-range structures and the Transformer can capture global, long-range relationships.

- **GraphGPS** (Rampásek et al., 2022) introduces a scalable and powerful Transformer model class for graph data and achieves state-of-the-art results on molecular property prediction benchmarks. We use its scalable implementation version with the Performer attentions (Choromanski et al., 2021).

- **DIFFormer** (Wu et al., 2023) is a scalable Transformer inspired by diffusion on graphs. The model is comprised of principled attention layers, which implements the diffusion iterations minimizing a global energy. The architecture integrates graph-based feature propagation and global attention in each layer. We use its version with simple diffusivity that only requires linear complexity and yields state-of-the-art results on some large-graph benchmarks.

---

[3]https://github.com/rdkit/rdkit

### F.3 IMPLEMENTATION DETAILS

**Computation Systems.** All the experiments are run on NVIDIA 3090 with 24GB memory. The environment is based on Ubuntu 18.04.6, Cuda 11.6, Pytorch 1.13.0 and Pytorch Geometric 2.1.0.

**Evaluation Protocol.** For all the experiments, we run the training and evaluation of each model with five independent trials, and report the mean and standard deviation results in our tables and figures. In each run, we train the model with a fixed budget of epochs and record the testing performance produced by the epoch where the model yields the best performance on validation data.

**Hyper-Parameters.** We use the grid search for hyper-parameter tuning on the validation dataset with the searching space described below.

- For information networks, hidden size $d \in \{32, 64, 128\}$, learning rate $\in \{0.0001, 0.001\}$, head number $H \in \{1, 2, 4\}$, the weight for local message passing $\beta \in \{0.2, 0.5, 0.8, 1.0\}$, and the order of propagation (only used for ADIT-SERIES) $K \in \{1, 2, 4\}$.

- For molecular datasets, hidden size $d = 256$, learning rate $\in \{0.01, 0.005, 0.001, 0.0005, 0.0001, 0.00005\}$, dropout $\in \{0.0, 0.1, 0.3, 0.5\}$, head number $H \in \{1, 2, 4\}$, the weight for local message passing $\beta \in \{0.5, 0.75, 1.0\}$, the coefficient for identity matrix (only used for ADIT-INVERSE) $\theta \in \{0.5, 1.0\}$, and the order of propagation (only used for ADIT-SERIES) $K \in \{1, 2, 3, 4\}$.

- For protein interaction networks, hidden size $d \in \{32, 64\}$, learning rate $\in \{0.01, 0.001, 0.0001\}$, head number $H \in \{1, 2, 4\}$, the weight for local message passing $\beta \in \{0.3, 0.5, 0.8, 1.0\}$, the coefficient for identity matrix (only used for ADIT-INVERSE) $\theta \in \{0.5, 1.0\}$, and the order of propagation (only used for ADIT-SERIES) $K \in \{1, 2, 3, 4\}$.

## G ADDITIONAL EXPERIMENTAL RESULTS

In this section, we supplement more experimental results including additional results for main experiments, ablation studies and hyper-parameter analysis.

### G.1 SUPPLEMENTARY RESULTS FOR MAIN EXPERIMENTS

In Table 7, we report the Accuracy for training/validation/testing sets on `Arxiv`. In Table 8, we present the ROC-AUC for each graph of `Twitch`. In Fig. 7 and 8, we show the generated results for more testing cases of molecular mapping operators in `HAM`.

Table 7: Result of Accuracy for training/validation/testing sets on `Arxiv` where we consider inductive setting and use published papers in different years for data splitting.

| | Train (1960-2014) | Valid (2015-2017) | Test 1 (2018) | Test 2 (2019) | Test 3 (2020) |
|---|---|---|---|---|---|
| **MLP** | $58.15 \pm 0.67$ | $51.43 \pm 0.35$ | $49.91 \pm 0.59$ | $47.30 \pm 0.63$ | $46.78 \pm 0.98$ |
| **GCN** | $60.50 \pm 1.21$ | $53.03 \pm 0.26$ | $50.14 \pm 0.46$ | $48.06 \pm 1.13$ | $46.46 \pm 0.85$ |
| **GAT** | $63.01 \pm 1.05$ | $54.44 \pm 0.20$ | $51.60 \pm 0.43$ | $48.60 \pm 0.28$ | $46.50 \pm 0.21$ |
| **SGC** | $60.30 \pm 0.21$ | $54.39 \pm 0.06$ | $51.40 \pm 0.10$ | $49.07 \pm 0.16$ | $46.94 \pm 0.29$ |
| **GDC** | $66.76 \pm 2.51$ | $54.67 \pm 0.34$ | $51.88 \pm 0.41$ | $49.53 \pm 0.50$ | $47.33 \pm 0.60$ |
| **GRAND** | $64.65 \pm 1.10$ | $55.00 \pm 0.12$ | $52.45 \pm 0.27$ | $50.18 \pm 0.18$ | $48.01 \pm 0.24$ |
| **GraphTrans** | $63.39 \pm 1.85$ | $54.45 \pm 0.43$ | $51.70 \pm 0.55$ | $49.38 \pm 1.08$ | $47.01 \pm 0.98$ |
| **GraphGPS** | $67.02 \pm 3.18$ | $53.55 \pm 0.41$ | $51.11 \pm 0.19$ | $48.91 \pm 0.34$ | $46.46 \pm 0.95$ |
| **DIFFormer** | $60.29 \pm 0.88$ | $53.71 \pm 0.42$ | $50.45 \pm 0.94$ | $47.37 \pm 1.58$ | $44.30 \pm 2.02$ |
| **ADIT-SERIES** | $63.79 \pm 0.66$ | $55.25 \pm 0.14$ | $53.41 \pm 0.48$ | $51.53 \pm 0.60$ | $49.64 \pm 0.54$ |

### G.2 ABLATION STUDIES AND HYPER-PARAMETER ANALAYSIS

We next conduct more analysis on our proposed model by ablation studies on some key components and investigating the impact of hyper-parameters.

**Diffusion and Advection.** We conduct ablation studies on the advection term (i.e., the local message passing with the propagation by $\tilde{\mathbf{A}}$) and the diffusion term (i.e., the global attention). In Table 9

Table 8: Result of ROC-AUC for each graph on `Twitch` where we use nodes in different networks to split the training, validation and testing data.

| | Train (DE) | Valid (ENGB) | Test 1 (ES) | Test 2 (FR) | Test 3 (PTBR) | Test 4 (RU) | Test 5 (TW) |
|---|---|---|---|---|---|---|---|
| **MLP** | $75.26 \pm 1.49$ | $63.48 \pm 0.15$ | $65.19 \pm 0.37$ | $62.25 \pm 0.28$ | $65.01 \pm 0.19$ | $54.92 \pm 0.33$ | $58.23 \pm 0.13$ |
| **GCN** | $69.55 \pm 0.34$ | $60.76 \pm 0.21$ | $63.75 \pm 0.44$ | $61.56 \pm 0.56$ | $63.26 \pm 0.42$ | $54.51 \pm 0.21$ | $55.72 \pm 0.28$ |
| **GAT** | $69.28 \pm 1.14$ | $59.80 \pm 0.42$ | $62.81 \pm 1.16$ | $60.65 \pm 0.92$ | $63.13 \pm 1.25$ | $53.80 \pm 0.27$ | $55.31 \pm 0.94$ |
| **SGC** | $71.68 \pm 0.33$ | $61.98 \pm 0.07$ | $65.12 \pm 0.15$ | $63.06 \pm 0.12$ | $64.14 \pm 0.19$ | $55.17 \pm 0.06$ | $56.83 \pm 0.20$ |
| **GDC** | $80.73 \pm 1.69$ | $62.14 \pm 0.30$ | $66.33 \pm 0.25$ | $60.70 \pm 0.51$ | $64.21 \pm 0.23$ | $56.60 \pm 0.24$ | $58.97 \pm 0.37$ |
| **GRAND** | $79.17 \pm 0.74$ | $62.48 \pm 0.39$ | $66.52 \pm 0.23$ | $61.62 \pm 0.62$ | $64.44 \pm 0.73$ | $56.42 \pm 0.38$ | $59.27 \pm 0.57$ |
| **GraphTrans** | $79.17 \pm 0.74$ | $62.48 \pm 0.39$ | $66.52 \pm 0.23$ | $61.62 \pm 0.62$ | $64.44 \pm 0.73$ | $56.42 \pm 0.38$ | $59.27 \pm 0.57$ |
| **GraphGPS** | $74.49 \pm 1.35$ | $63.40 \pm 0.31$ | $66.85 \pm 0.32$ | $63.74 \pm 0.37$ | $65.03 \pm 0.58$ | $56.39 \pm 0.39$ | $58.63 \pm 0.83$ |
| **DIFFormer** | $73.12 \pm 0.52$ | $63.06 \pm 0.09$ | $66.68 \pm 0.15$ | $64.44 \pm 0.13$ | $65.23 \pm 0.20$ | $55.75 \pm 0.12$ | $58.91 \pm 0.37$ |
| **ADIT-SERIES** | $75.46 \pm 0.28$ | $63.53 \pm 0.14$ | $66.78 \pm 0.14$ | $63.35 \pm 0.10$ | $65.68 \pm 0.06$ | $56.27 \pm 0.06$ | $60.48 \pm 0.21$ |

Table 9: Ablation studies for ADIT-SERIES on `Arxiv`.

| | Train (1960-2014) | Valid (2015-2017) | Test 1 (2018) | Test 2 (2019) | Test 3 (2020) |
|---|---|---|---|---|---|
| **ADIT** | $63.79 \pm 0.66$ | $55.25 \pm 0.14$ | $53.41 \pm 0.48$ | $51.53 \pm 0.60$ | $49.64 \pm 0.54$ |
| **ADIT** w/o diffusion | $64.65 \pm 1.10$ | $55.00 \pm 0.12$ | $52.45 \pm 0.27$ | $50.18 \pm 0.18$ | $48.01 \pm 0.24$ |
| **ADIT** w/o advection | $57.71 \pm 0.86$ | $51.27 \pm 0.21$ | $49.52 \pm 1.02$ | $45.74 \pm 1.74$ | $44.71 \pm 1.85$ |

we report the results for ADIT-SERIES on `Arxiv`, which shows that the two modules are indeed effective for producing superior generalization on node classification tasks.

**Edge Features for Molecular Tasks.** For molecular property prediction datasets, we use edge features (that contain attribution information of chemical bonds) for our model as well as other competitors. In Table 10 we compare with the model variant not using edge features. The results show that edge features, which provide additional information useful for predicting the target molecular properties, can indeed contribute to improving the generalization performance.

Table 10: Ablation studies for ADIT-INVERSE on `OGB-BACE` and `OGB-SIDER`.

| | OGB-BACE | | | OGB-SIDER | | |
|---|---|---|---|---|---|---|
| | Train | Valid | Test | Train | Valid | Test |
| **ADIT** | $97.39 \pm 1.67$ | $73.82 \pm 1.45$ | $80.38 \pm 1.40$ | $83.67 \pm 0.09$ | $60.85 \pm 0.22$ | $65.29 \pm 0.16$ |
| **ADIT** w/o edge features | $95.76 \pm 4.27$ | $75.90 \pm 2.68$ | $79.14 \pm 1.96$ | $81.26 \pm 1.18$ | $57.44 \pm 0.26$ | $63.77 \pm 0.40$ |

**Impact of $\beta$.** The hyper-parameter $\beta$ controls the importance weight for the advection term (implemented as the local message passing). Fig. 5 shows the model performance of ADIT-SERIES on `Arxiv` and `DDPIN` with different $\beta$'s. We found that the optimal settings for $\beta$ can be different across datasets and tasks. In specific, for node classification on `Arxiv`, the model gives the best performance with $\beta$ in the range between 0.7 and 1.0. The performance degrades when $\beta$ is too small ($<0.5$) or too large ($>2.0$). The reason could be that the graph structural information is useful for the predictive task on `Arxiv` yet too much emphasis on the graph structure can lead to undesired generalization of the model. Differently, for the protein interaction dataset `DDPIN`, we found that using smaller $\beta$ can bring up more satisfactory performance across node regression and edge regression tasks. In particular, setting $\beta = 0$, in which case the advection term is completely dropped, can yield optimal performance for the node regression task. This is possibly due to that the graph structure is not that informative and pure global attention can learn generalizable topological patterns from latent interactions. To sum up, in practice, the model enables much flexibility for adjusting the weight on the advection effect (the importance of observed structural information) to accommodate the diversity of graph-structured data.

**Impact of $K$.** The hyper-parameter $K$ (used for ADIT-SERIES) controls the number of propagation orders in the model. In fact, the value of $K$ would impact how the structural information is utilized by the model. If $K$ is small, the model only utilizes the low-order structure, and large $K$ enables the usage of high-order structural information. Fig. 6 presents the model performance on `Arxiv` and `DDPIN` with $K$ ranging from 1 to 6. We observe that the optimal settings for $K$ are different across cases, and using larger $K$ can not necessarily yield better performance. This is because in these cases, the low-order structural information is informative enough for desired generalization.

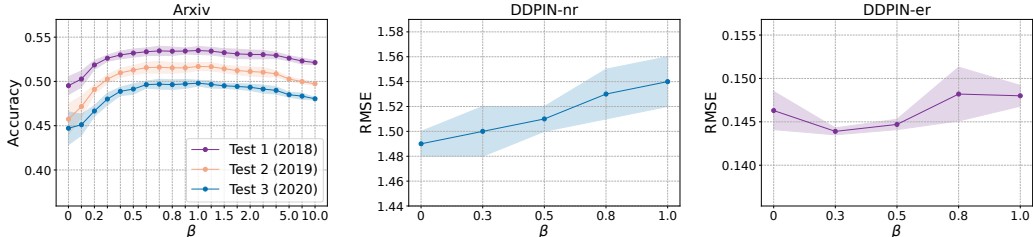

Figure 5: Model performance on `Arxiv` and `DDPIN` with different settings of $\beta$. The latter involves node regression (nr) and edge regression (er) tasks.

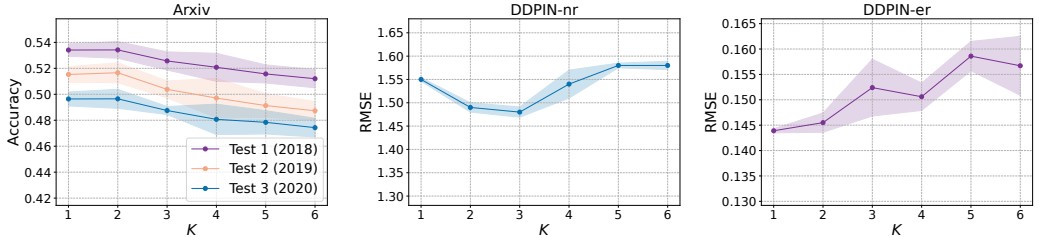

Figure 6: Model performance on `Arxiv` and `DDPIN` with different settings of $K$. The latter involves node regression (nr) and edge regression (er) tasks.

**Impact of $\theta$.** Finally, we study the impact of $\theta$ used for computing $\mathbf{L}_h$ in ADIT-INVERSE. Table 11 shows the performance of ADIT-INVERSE on `DDPIN` with different $\theta$'s. We found that using $\theta$ close to 1 can bring up stably good performance, which is consistently manifested by experiments on other cases. Still, if $\theta$ is too small, e.g., close to 0, it would sometimes lead to numerical instability. This is due to that, in such a case, the matrix $\mathbf{L}_h$ could become a singular matrix.

Table 11: Testing accuracy of ADIT-INVERSE with different $\theta$'s in the edge regression task on `DDPIN`.

| $\theta$ | 0 | 0.5 | 1.0 | 2.0 |
|---|---|---|---|---|
| Accuracy | 0.241 | 0.154 | 0.147 | 0.149 |

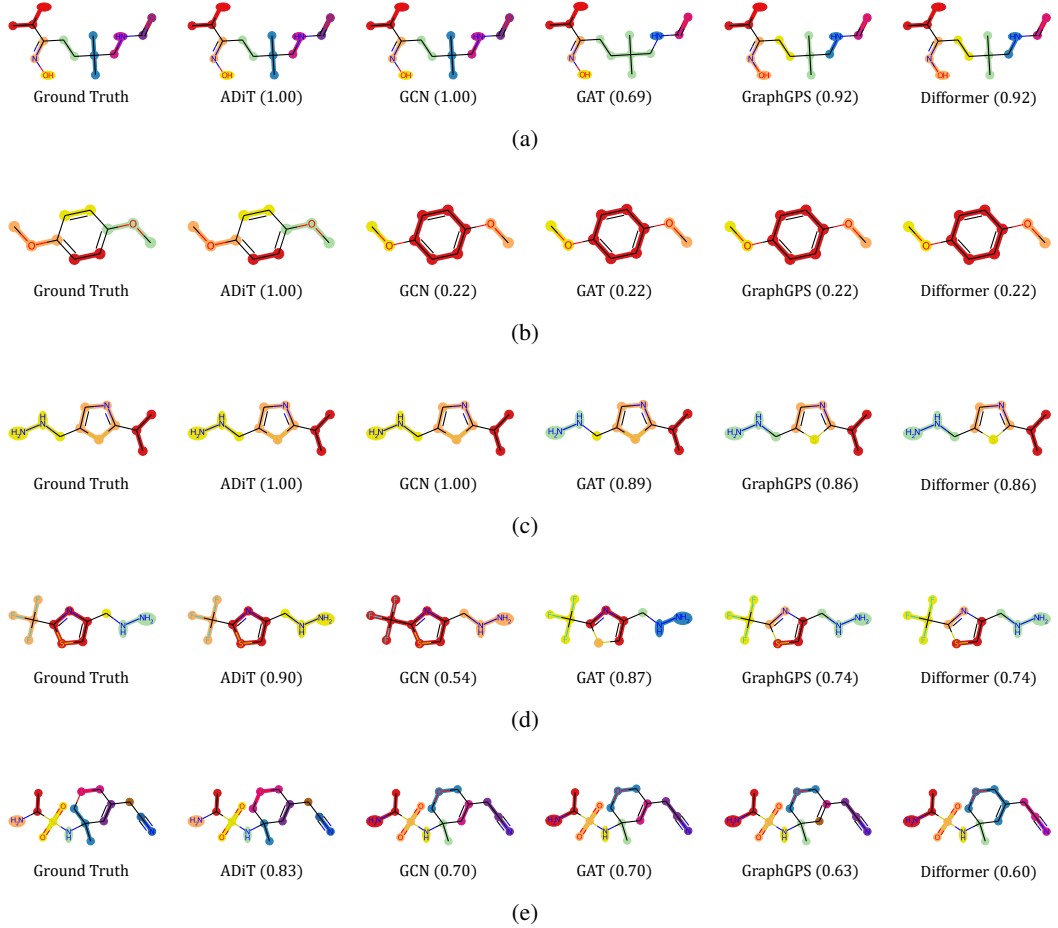

Figure 7: Additional testing cases for molecular mapping operators generated by different models and the expert annotations (ground-truth). For each case, we report the score (the higher is better) that measures the closeness between the generated results and the expert annotations.

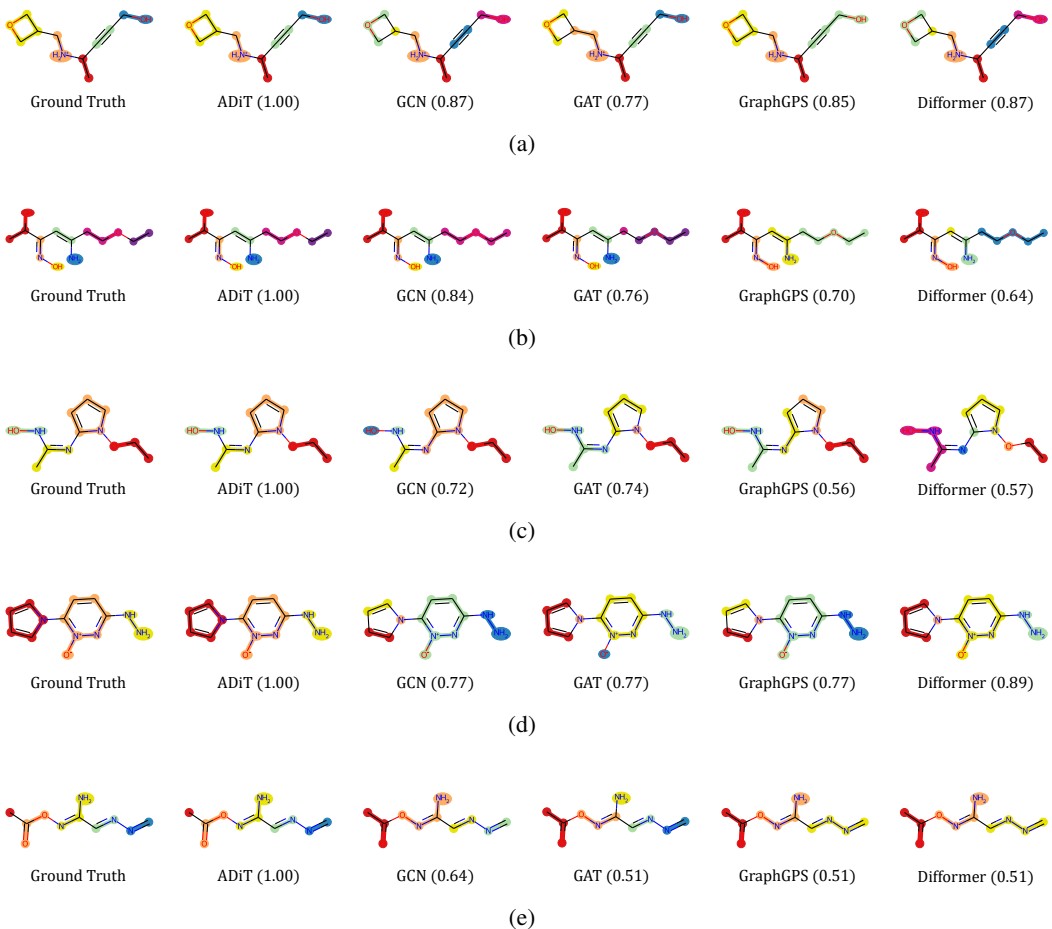

Figure 8: Additional testing cases for molecular mapping operators generated by different models and the expert annotations (ground-truth). For each case, we report the score (the higher is better) that measures the closeness between the generated results and the expert annotations.

