# OpenReview forum: "Advective Diffusion Transformers for Topological Generalization in Graph Learning"
_ICLR.cc/2024/Conference — Submitted to ICLR 2024_

### Official Review · Reviewer_834k · 2023-10-17

**Soundness:** 1 poor
**Presentation:** 4 excellent
**Contribution:** 3 good
**Rating:** 3
**Confidence:** 3

**Summary:**

The paper proposed a new method ADIT which uses graph as advection term whereas diffusion term is computed globally. The authors provide theoretical support on the relation between locality of diffusion term and robustness of model against graph topological changes. The authors then provided experimental support on the performance of their method.

**Strengths:**

- The idea is noval to me.
- The problem it solves is significant. In a lot of cases, graphs are generated with randomness, whereas all state of the art graph diffusion methods I knew are not robust to randomness in graph structure.
- The paper is well presented and easy to read.

**Weaknesses:**

- Figure 3: needs a footnote explaining OOM

**Questions:**

The method is theoretically very nice, but not scalable at all. The global attention is too costly in computation. General graph diffusion methods cost O(E) whereas global diffusion of ADIT cost O(V^2). So I have a few questions on how the authors would think to extend this model to larger graphs.
- Is it possible to use methods from efficient transformer to speed up the global diffuison cost of the model? Are there some suggestions?
- Is it possible to split the graph into batches of vertices / edges, and do it batch-by-batch?
- A lot of linear graph diffusion methods corresponds to a random walk. What is the random walk counterpart of ADIT if there is any?

---

> ### Author Response · Authors · 2023-11-17
> **Response to Reviewer 834k**
>
> Thank you for the nice suggestions. We are encouraged to see the positive comments on our novel ideas, significant research problem and good presentation.
>
> >***Q1: Scalability of the model and solutions for acceleration***
>
> Thanks for this inquiry. As illustrated in Sec. 4.3, our model ADiT-Series only requires $O(N)$ (a.k.a. $O(|\mathcal V|)$) complexity instead of $O(N^2)$. This acceleration is achieved by altering the order of matrix product (the same technique is used by DIFFormer [1]), with the details presented in Appendix E.1.2. Equipped with the linear complexity w.r.t. node numbers, ADiT-Series can smoothly scale to large-sized graphs.
>
> Apart from our used acceleration technique, there also exists other potential choices, such as the random feature map or sliding-window attention, used by efficient Transformers to achieve the similar goal.
>
> [1] Difformer: Scalable (graph) transformers induced by energy constrained diffusion, ICLR 2023.
>
> >***Q2: Is it possible to split the graph into batches of vertices / edges, and do it batch-by-batch?***
>
> Definitely yes. For graphs with larger sizes, e.g., with millions or even billions of nodes, it can be difficult for full-graph training on a GPU with moderate memory. In these situations, splitting the graph into random batches of nodes can help to reduce the overhead, as is done by recent scalable graph Transformers [1, 2]. This technique for scalability is orthogonal to our research focus and agnostic to the model backbone, and we'd like to leave more exploration along this promising direction for future works.
>
> [2] Nodeformer: A scalable graph structure learning transformer for node classification, NeurIPS 2022.
>
> >***Q3: A lot of linear graph diffusion methods corresponds to a random walk. What is the random walk counterpart of ADIT if there is any?***
>
> Thanks for this insightful question. For our ADiT model with the formulation of Eqn. 8, one can treat the coupling matrix $\mathbf C + \beta \tilde{\mathbf A}$ as a random walk propagation matrix, in which case the diffusion process can be considered as a continuous random walk counterpart. The key difference in our model lies in the dense propagation matrix that defines the transition probability over a densely connected latent graph produced by the attention incorporated with the observed structural information.
>
> >***W1: Figure 3: needs a footnote explaining OOM***
>
> We thank the reviewer for the nice suggestion. The OOM indicates out-of-memory error with a 24GB GPU used in our experiment, as illustrated in the caption of Table 1 where it appears at the first time.
>
> Please let us know if you had any further question or feedback.

---

> > ### Comment · Reviewer_834k · 2023-11-21
> >
> > I am lowering my score. Unfortunately the authors did not properly address the other reviewer's doubt on exponential matrix commuting problems. The matrix commuting with perturbation introduces non-negilgible error terms that could be way larger than the error bound the authors want to provide. Taking matrix log implicitly assume that matrix eigenvalues are all positive. The authors should prove why the eigenvalues are all positive, and then proceed with error analysis on how much errors are induced by commuting the perturbations. Note that the perturbation might already been boosted a lot by the log operation. Furthermore, there is an eigenvalue of diffusion operator that is close to (if not equal) 0. If a perturbation lands on that eigenvector, there will be a huge change in the log matrix, if it exists. Afterall, the idea of the paper is nice, I would suggest the authors to revisit their proofs and resubmit to a later conference.

---

> > > ### Author Response · Authors · 2023-11-21
> > >
> > > Thanks for the feedback and suggestion.
> > >
> > > The proofs will be fixed by avoiding the commutation of matrix exponential.
> > >
> > > For the matrix logarithm, our current proof only involves the logarithm of $\tilde{\mathbf A} + \mathbf I$ whose eigenvalues are all non-negative given the normalized adjacency matrix.

---

### Official Review · Reviewer_zCh7 · 2023-10-25

**Soundness:** 1 poor
**Presentation:** 2 fair
**Contribution:** 1 poor
**Rating:** 3
**Confidence:** 5

**Summary:**

This paper introduces Advective Diffusion Transformer to address the graph topologies generalization problems in GNNs under the assumption of a specific data-generating condition. However, there are errors in the proofs of Propositions 1, 2, and 3, as well as Theorem 1 and 2, which constitute the core contributions of this paper.  So the work lacks the necessary solidity for acceptance. These are substantial errors that call into question the paper's reliability. Additionally, this paper lacks a comparison of some highly related work.

**Strengths:**

1. The exploration of graph topological shifts under the assumption of graphon is interesting.
2. The experiments are thorough, encompassing a diverse array of datasets, which enhances the comprehensiveness of the study.
3. The integration of global attention with local message passing within the model presents an intriguing methodology.

**Weaknesses:**

1. I have some big concerns about this paper, especially when looking at the starting Proposition 1 and Proposition 2 in the paper. The authors say that changes in the graph structure affect the graph neural diffusion model a lot. But I don't think they've given enough proof for this, especially when I compare it to what's said in the paper [1]. \
In Proposition 1's proof, they say that $\tilde{A'}$ and $\tilde{A}$ can be swapped around (are commutative), **but that's not always true**. The whole point of Proposition 1 seems to depend on this being right. If it's not right, then Proposition 1 doesn't work. This makes me question the other things the paper says. \
Also, even if we say that their assumption is right, the result they get seems too general. **It's not as specific as the conclusion in the Proposition 1 of the paper [1]**. Proposition 2 also uses the same kind of reasoning as Proposition 1.\
The proofs in the paper have one fundamental flaw that leads to incorrect equations.  For instance, in many equations (Eg. Eq(40), (58), (60), (63), (64), (76) etc.), it uses $e^(X+Y) = e^Xe^Y$ **without considering the commutative between $X$ and $Y$.** \
Because of these problems, especially the mistake in Proposition 1, I don't think this paper should be accepted as it is. They need to go back, check their work and related works such as [1][9], and make it clearer.
2. **Equation (73) in the proof of Proposition 3 is incorrect.** The validity of Equation (73) requires that **$\cal{H}$ be a finite hypothesis class and the loss function $l$ be bounded within $[0,1]$.** Specifically, the penultimate inequality in Equation (73) holds only if $\cal{H}$ is finite, and the last inequality is true when the loss function $l$ is bounded in $[0,1]$. However, this paper does not fulfill the required conditions: the hypothesis class $\cal{H}$  is not finite as it involves neural networks, and the loss functions used like cross-entropy or MSE are not bounded between $[0,1]$. As a result, the conclusions drawn in Proposition 3 are incorrect. **This issue also affects Equation (85) in the proof of Theorem 2, leading to incorrect in the Theorem 2 conclusions.**  In conclusion, the main claim of this paper, Theorem 2, rests on the incorrect assumption. This error is critical enough to question the entire paper's validity, making its findings untrustworthy.
3. Using Advective Diffusion Equations in GNNs is not new, see the related works such as [2] and [3].
4. This paper also lacks a comparison and citation of related works, particularly those focusing on the generalization of topology distribution shift based on the graphon theory, such as references [6] and [7].
5. Theorem 1 and Theorem 2 hold under the data generation hypothesis in Sec. 3.1. How does this data generation hypothesis correspond to the real-world datasets in Sec. 5.2? The citations of the graphon theory are not given in Sec. 3.1. Can you explain more about the graphon?
6. In the proof of Theorem 1, why $C=\bar{C}+m log(I + \tilde{A}) - \beta \tilde{A}$ ?
7. For the model implementation in Section 4.3, why have approximation techniques been chosen over the numerical solvers proposed in [4], which are commonly utilized in graph neural diffusion models such as GRAND, GRAND++, GraphCON[8], CDE[3], as well as GREAD[5] models? Have any ablation studies been conducted to compare the effectiveness of different numerical solvers?
8. Why are there no results for ADIT-INVERSE in Table 1? Additionally, the results for ADIT-SERIES show only marginal improvement compared to GRAND. It would be beneficial to include more baselines of graph neural diffusion models, such as the model in [1][9] and GraphCON, for a more comprehensive comparison.
9. No code has been submitted to reproduce the experiment results.

[1]. Song Y, et al. On the robustness of graph neural diffusion to topology perturbations[J]. Advances in Neural Information Processing Systems, 2022

[2]. Eliasof M, et al. ADR-GNN: Advection-Diffusion-Reaction Graph Neural Networks[J]. arXiv preprint arXiv:2307.16092, 2023.

[3]. K. Zhao, et al.  “Graph neural convection-diffusion with heterophily,”  International Joint Conference on Artificial Intelligence (2023)

[4]. Chen, Ricky TQ, et al. "Neural ordinary differential equations." Advances in neural information processing systems 31 (2018).

[5]. Choi, Jeongwhan, et al. "GREAD: Graph Neural Reaction-Diffusion Equations." arXiv preprint arXiv:2211.14208 (2022).

[6] Ruiz, Luana, et al.  "Graphon neural networks and the transferability of graph neural networks." Advances in Neural Information Processing Systems 33 (2020): 1702-1712.

[7] Ruiz, L., et al.  (2023). Transferability properties of graph neural networks. IEEE Transactions on Signal Processing.

[8] Rusch T K, et al.  Graph-coupled oscillator networks[C]//International Conference on Machine Learning. PMLR, 2022: 18888-18909.

[9] Gravina A, et al.  Anti-symmetric dgn: A stable architecture for deep graph networks[J]. arXiv preprint arXiv:2210.09789, 2022.

**Questions:**

Please refer to the Weaknesses part

---

> ### Author Response · Authors · 2023-11-17
> **Response to Reviewer zCh7 (Part 1/3)**
>
> Thank you for the careful reviewing and checking our proofs in detail. We appreciate the nice suggestions for improvements and the acknowledgement on our problem setting, methodology and experiments. We also would like to clarify some key points that might be potentially misinterpreted in the review.
>
> >***Q1: The communicative property of $\tilde{\mathbf A}$ and $\tilde{\mathbf A}'$***
>
> We'd like to clarify upfront that **our studied topological generalization problem has distinct technical aspects from the adversarial robustness studied in [1]**. The latter situation involves varied topology introduced by the adversarial attack, which can significantly differ from the original graph, while in our case, the new topology with distribution shifts conforms to the underlying data manifold and mechanism. In this regard, we are interested in the extrapolation behavior of the model w.r.t. the small change of $\tilde{\mathbf A}$.
>  Therefore, throughout our analysis sections, we assume $\Delta \tilde{\mathbf A}$ to be a small perturbation, i.e., $||\Delta \tilde{\mathbf A}||_2 \rightarrow 0$.
>
> This paves the way for analyzing the quantity $\lim_{||\Delta \tilde{\mathbf A}||_2 \rightarrow 0} \frac{||\mathbf Z(T; \tilde{\mathbf A}') - \mathbf Z(T; \tilde{\mathbf A})||_2}{||\Delta \tilde{\mathbf A}||_2}$ that measures the change rate of node representations w.r.t. the small change centered at $\tilde{\mathbf A}$. In this regime, it holds that $\tilde{\mathbf A}$ and $\tilde{\mathbf A}' = \tilde{\mathbf A} + \Delta \tilde{\mathbf A}$ share the same eigenspace and are commutative. This makes the property satisfied $e^{\tilde{\mathbf A} \tilde{\mathbf A}'} = e^{\tilde{\mathbf A}' \tilde{\mathbf A}} = e^{\tilde{\mathbf A}}\cdot e^{\tilde{\mathbf A}'}$ that is used in our proof.
>
> Also, we agree with the reviewer that our analysis for Prop 1 and 2 is not as specific as the conclusion in [1]. However, we do not consider this as a disadvantage or major limitation of our work since ** this paper mainly focuses on the methodology side**. The Prop 1 and 2 serve as the preliminary results exploring the problem, and our main goal is not to show the failure of local diffusion but to suggest its potential sensisitity to topological shifts. Our latter analysis for non-local diffusion and proposal of the new model aim at addressing the potential sensisitity and improving the generalization capabilities.
>
> [1] On the robustness of graph neural diffusion to topology perturbations, NeurIPS 2022
>
>
> >***Q2: The validity of Equation (73) requires that $\mathcal H$ be a finite hypothesis class and the loss function be bounded within [0, 1]***
>
> We thank the reviewer for pointing out this issue that helps us to fix our proof for considering more generic neural networks. The original Eqn. 73 is not rigorous enough for $\Gamma$ as a neural network that induces an infinite hypothesis space size, yet **the proof can be easily fixed to accommodate neural networks without impacting the main conclusion of Prop 3 and Thm 2**.
>
> To be specific, if the hypothesis space has infinite size, we can leverage the well-established techniques, such as VC-dimension or Rademacher complexity, to derive the generalization bound. This does not impact the conclusion—the hypothesis space size $|\mathcal H(\Gamma)|$ can be replaced by the Rademacher complexity of $\Gamma$. The latter is a complexity measure of the function class, and Prop 3 and Thm 2 still hold with the original implications unchanged. (We slightly modify the statement in Prop 3 and the proof around Eqn. 73 in the new PDF).

---

> ### Author Response · Authors · 2023-11-17
> **Response to Reviewer zCh7 (Part 2/3)**
>
> >***Q3: Comparison with other works using advective diffusion equations [2, 3]***
>
> We'd like to clarify that advective diffusion equations are a class of diffusion equations serving as a generic mathematical framework. **Though we use this framework as a principled perspective for motivating the model, it remains much room for the originality of model designs.** Compared with [2, 3], **our work targets a distinct problem and our proposed model considerably differs.**
>
> To be specific, [2, 3] target the standard graph learning problem where the training and test graphs are assumed to be from the same distribution. In this regard, the local diffusion/advection terms are both instantiated as local message passing by [2, 3] to improve the model's expressivity and the capability to handle heterophily, respectively. Differently, our work studies the topological generalization problem where the training and test graphs come from disparate distributions, so the main goal is to improve the model's generalization capability. To this end, we consider the non-local diffusion instantiated as global attention. Furthermore, in terms of implementation, prior art including [2, 3] uses the numerical solver for solving the PDE model, while we harness the approximation scheme based on the Padé-Chebyshev theory for computing the closed-form PDE solution. The table below presents a head-to-head comparison.
>
>
> | Aspect | ADR-GNN [2] | CDE [3] | Our work |
> | -------- | -------- | -------- | -------- |
> | Problem Setting     | standard graph learning | standard graph learning | topological generalization |
> | Diffusion Instantiation | local message passing | local message passing | global attention |
> | Implementation | finite-difference iteration | neural PDE solver | Padé-Chebyshev theory |
> | Experiment Scenario | node-level task | node-level task | node-level, edge-level, graph-level tasks |
>
> [2] ADR-GNN: Advection-Diffusion-Reaction Graph Neural Networks. arXiv preprint 2023.
>
> [3] Graph neural convection-diffusion with heterophily, IJCAI 2023
>
>
> >***Q4: Comparison with other works focus on generalization of topology distribution shift based on the graphon theory [6, 7]***
>
> Thanks for this inquiry on the comparison, which allows us to illuminate our contributions.
>
> ***Analysis Target*** First of all, [6, 7] do not study graph diffusion models, and instead they analyze the discrete GNN models. Though graph diffusion equations are intimately related to GNNs, analyzing the generalization behaviors of graph diffusion models and GNNs needs disparate technical aspects and proof techniques.
>
> ***Data Hypothesis*** [6, 7] resort to the conventional graphon model as the data hypothesis that ignores node features and labels in the generation, while our hypothesis in Sec. 3.1 resorts to a more general setting that includes alongside graph adjacency node features and labels in the data generation. Moreover, the analysis in [7] is limited to three specific types of graphs whose distribution shifts are determined by specific edge distributions. These specific graph types are special cases of our framework, where one can flexibly instantiate the pairwise function $h$ in our data-generating mechanism. In this regard, our analysis targets a more general setting that better represents complex real data.
>
> ***Model Designs*** The models proposed in [6, 7] belong to the conventional discrete MPNNs that resort to local message passing, while our proposed model consists of a Transformer-like architecture built on continuous PDE.
>
> [6] Graphon neural networks and the transferability of graph neural networks, NeurIPS 2020.
>
> [7] Transferability properties of graph neural networks. IEEE Transactions on Signal Processing, 2023.
>
> >***Q5: How does this data generation hypothesis correspond to the real-world datasets in Sec. 5.2? The citations of the graphon theory are not given in Sec. 3.1. Can you explain more about the graphon?***
>
> The data generation hypothesis in Sec. 3.1 presents a general setting that can represent the complex real-world data with topological distribution shifts caused by the latent environment $E$. The latter can have different physical meanings in practical cases. For example, the environment for the Arxiv dataset is the publication year which introduces distribution shifts across papers published at different times, and the environment for the Twitch dataset is the geographic domain that leads to distribution shifts across users in different regions. For the molecule datasets, the environment is the molecular scaffold, and for the protein datasets, the environment is the protein identification method. In these situations, the training and test data correspond to disparate environments and distribution shifts are caused by the varied environments in each case.
>
> (See the follow-up response for the graphon issue)

---

> ### Author Response · Authors · 2023-11-17
> **Response to Reviewer zCh7 (Part 3/3)**
>
> For the graphon theory, we have cited the references at the beginning of Sec. 3.1: "inspired by the graph limits (Lovász & Szegedy, 2006; Medvedev, 2014) and random graph models (Snijders & Nowicki, 1997)". Please kindly notice that our data generation hypothesis does not directly follow the graphon, whose original definition ignores node features and labels, and instead, we generalize the data-generating mechanism to include alongside graph adjacency also node features and labels that can better represent the complex real data.
>
> >***Q6: In the proof of Theorem 1, why $C = \overline C + m\log (I + \tilde A) - \beta \tilde A$***
>
> Our proof for Thm 1 is based on construction. In this step, we consider a construction for the global attention matrix $\mathcal C$, according to the universal approximation results that hold for MLPs on the compact set [10].
>
> [10] Kurt Hornik, Maxwell Stinchcombe, and Halbert White. Multilayer feedforward networks are universal approximators. Neural networks, 2(5):359–366, 1989
>
> >***Q7: Why not using the numerical solvers proposed in [4]***
>
> Though the numerical solver proposed in [4] is widely adopted by other graph diffusion models, it is time-consuming and can be non-stable since it requires the backpropagation through a continuous PDE/ODE. In contrast, our diffusion model Eqn. 8 induces a closed-form solution, which enables us to implement the model via computing the closed-form solution rather than solving the PDE with numerical solvers as is done by other diffusion models. To be specific, our model implementation described in Sec. 4.3. is derived from computing the closed-form solution using the approximation of the Padé-Chebyshev theory. This scheme is much faster and more numerically stable. This also fundamentally distinguishes our implementation from other graph diffusion models.
>
> To strengthen our contribution on this point, we supplement ablation study comparing with the counterpart of ADiT (called ADiT-PDESolver) using the numerical solver of [4] on the DDPIN dataset. The results below show the effectiveness of our scheme. Besides, we found ADiT-Inverse and ADiT-Series are about 3 and 20 times faster than ADiT-PDESolver for training.
>
> |  | Node Regression (RMSE) | Edge Regression (RMSE) |
> | -------- | -------- | -------- |
> | ADiT-Inverse     | 1.75 ± 0.02 | 0.147 ± 0.002 |
> | ADiT-Series | 1.49 ± 0.03 | 0.144 ± 0.001 |
> | ADiT-PDESolver | 1.92 ± 0.05 | 0.162 ± 0.008 |
>
> [4] Neural ordinary differential equations, NeurIPS 2018
>
> >***Q8: Why are there no results for ADIT-INVERSE in Table 1? It would be beneficial to include more baselines of graph neural diffusion models.***
>
> As illustrated in Sec. 4.3., ADiT-Series only requires $O(N)$ complexity (by means of the acceleration technique described in Appendix E.1.2) and has better scalability than ADiT-Inverse that requires $O(N^2)$ complexity. For the two node classification datasets in Table 1 that have relatively large graph sizes, it is hard for ADiT-Inverse to be trained on a single GPU with moderate 24GB memory, so we only use ADiT-Series in this case.
>
> Following the suggestion of the reviewer, we supplement more comparison with other graph diffusion models including ADR-GNN [2] and Anti-Symmetric DGN [8], with the results below. We do not compare with [1] since it studies adversarial robustness which has different research focus from us.
>
> | Dataset | Arxiv (2018) | Arxiv (2019) | Arxiv (2020) | Twitch (avg) |
> | -------- | -------- | -------- | -------- | -------- |
> | ADiT-Series     | 53.41 ± 0.48 | 51.53 ± 0.60 | 49.64 ± 0.54 | 62.51 ± 0.07 |
> | ADR-GNN     | 50.12 ± 0.23 | 47.31 ± 0.59 | 45.14 ± 0.31 | 60.87 ± 0.12 |
> | Anti-Symmetric DGN | 50.91 ± 0.41 | 47.54 ± 0.61 | 45.79 ± 0.39 | 60.11 ± 0.09 |
>
> [8] Anti-symmetric dgn: A stable architecture for deep graph networks, ICLR 2023.
>
>
> As stated in our paper, we will release the codes upon publication. We thank the reviewer again for the constructive suggestions that help us strengthen the quality of the paper. And please let us know if you had any further question or feedback.

---

> > ### Comment · Reviewer_zCh7 · 2023-11-21
> >
> > > Q7: Why not using the numerical solvers proposed in [4]?
> >
> > There's a variety of numerical solvers mentioned in [4], including Euler, RK4, Dopri5. Which specific solver is employed in your ADiT-PDESolver?

---

> ### Comment · Reviewer_zCh7 · 2023-11-21
>
> 1, I share the concerns raised by reviewer SG1T in point 1 and 2. The authors have not addressed these, which are fundamental errors in the paper.
>
> 2, Merely substituting the hypothesis space size with Rademacher complexity in Proposition 3 and Theorem 2 is overly simplistic and lacks specificity. For a meaningful analysis of generalization error in GNNs, detailed insights into the Rademacher complexity of your $\Gamma$ are essential.

---

> ### Author Response · Authors · 2023-11-21
>
> Thanks for the feedback.
>
> 1. We will fix the proofs in later version to address the issue caused by the commutation.
>
> 2. For the Rademacher complexity, we thank the reviewer for the suggestion and we agree that more discussions on the complexity of \Gamma will add some value to our work from the side. We'd also like to point out that the big picture of our work focues on the topological generalization, and the main focus of our generalization analysis lies in the influence of topological shifts on the generalization error, i.e., $\mathcal D_2$ in Thm 2. We have discussed how the model can reduce the influence of topological shifts on generalization. In this sense, the analysis is specific to our studied problem and fundamental to the solution in this paper.
>
> 3. We used the widely adopted Euler for the ADiT-PDESolver.

---

### Official Review · Reviewer_SG1T · 2023-10-26

**Soundness:** 1 poor
**Presentation:** 3 good
**Contribution:** 1 poor
**Rating:** 3
**Confidence:** 4

**Summary:**

This paper investigates advective diffusion graph neural network models and study their generalization under graph topology changes.

**Strengths:**

This paper studies the generalization of graph advective diffusion models under graph topology changes.

**Weaknesses:**

1. The authors seem to be unaware of many related works that have already various aspects proposed in this paper. The novelty of the paper is thus unclear.

1. The given bounds in Proposition 1 and  2 are very loose. Furthermore, these are Big-O bounds and do not justify the claim that the label prediction can be highly sensitive to topology perturbations. To make such a claim, a Big-Omega bound should be provided. I am unconvinced by the authors' motivations for this work.

1. There are critical flaws or gaps in the proofs. E.g., in the line before (76). There is no justification why the exponential operator can be factored out. Note that if $A$ and $B$ do not commute, then $e^{A+B} \ne e^A e^B$ or $e^{B}e^{A}$.

**Questions:**

1. The proposed model is very close to the following. The authors need to explain the differences.
  - ACMP: Allen-cahn message passing with attractive and repulsive forces for graph neural networks in ICLR 2023
  - Graph neural convection-diffusion with heterophily in IJCAI 2023

2. The effect of graph structure changes on neural diffusion GNN models have been studied in the following. Generalization results in GNNs have also been proposed. What are the additional new results in this paper?
  - Graphon neural networks and the transferability of graph neural networks in NeurIPS 2020
  - On the robustness of graph neural diffusion to topology perturbations in NeurIPS 2022
  - Transferability properties of graph neural networks in IEEE Transactions on Signal Processing.

3. In the proof of Proposition 1, it is stated that $\tilde{A}$ and $\Delta \tilde{A}$ share the same eigenspace. Why is this true? It seems to be a very critical assumption that needs to be comprehensively justified and stated up front.

4. How does the proposed model perform under heterophily datasets?

---

> ### Author Response · Authors · 2023-11-17
> **Response to Reviewer SG1T (Part 1/2)**
>
> Thank you for reviewing our manuscript and proposing the inquiries that allow us to clarify our differences and properly position our work with existing ones.
>
> >***Q1 & W1: The proposed model is very close to [1, 2]. The authors need to explain the differences.***
>
> Thanks for suggesting these related works, which enables us to clarify our contributions. Though [1,2] also build their models upon continuous PDE diffusion as the foundation, **our work targets distinct problems and proposes considerably different models**.
>
> In terms of problem settings, [1,2] focuses on the standard graph learning problem (where training and test data are assumed to be from the same distribution) and the main goal is to address the heterophily and over-smoothing. In contrast, our work aims at topological generalization (where training and test data are from dissimilar distributions), which has fundamentally different technical aspects.
>
> In terms of models and implementation, [1,2] are both built on local graph diffusion equation. To be specific, [1] extends the diffusion operator to combine the attractive and repulsive effects, while [2] instantiates both the diffusion/advection operators as local message passing. Also, for implementation, these models uses ODE/PDE solver for solving the continous PDE.  Differently, our model resorts to non-local diffusion operator instantiated as global all-pair attention. For implementation, we leverage the Padé-Chebyshev theory for computing the closed-form PDE solution. The table below presents a head-to-head comparison.
>
> | Aspect | ACMP [1] | CDE [2] | Our work |
> | - | - | - | - |
> | Problem Setting     | standard graph learning (oversmoothing, heterophily) | standard graph learning (heterophily) | topological generalization |
> | Diffusion Instantiation | local message passing | local message passing | global attention |
> | Implementation | PDE solver for solving equation | PDE solver for solving equation | Padé-Chebyshev theory for computing closed-form solution |
> | Experiment Scenario | node-level task | node-level task | node-level, edge-level, graph-level tasks |
>
> [1] ACMP: Allen-cahn message passing with attractive and repulsive forces for graph neural networks in ICLR 2023
>
> [2] Graph neural convection-diffusion with heterophily in IJCAI 2023
>
> >***Q2 & W1: The effect of graph structure changes on neural diffusion GNN models have been studied in the following [3,4,5]. Generalization results in GNNs have also been proposed. What are the additional new results in this paper?***
>
> Our work is actually different from [3,4,5], with detailed comparison below.
>
> ***Comparison with [3,5]*** First of all, [3,5] do not study graph diffusion models, and instead they analyze the discrete GNNs. Though graph diffusion equations are intimately related to GNNs, analyzing the generalization behaviors of graph diffusion models and GNNs needs disparate technical aspects and proof techniques. More importantly, [3,5] resort to the conventional graphon model as the data hypothesis that ignores node features and labels in the generation, while our hypothesis in Sec. 3.1 resorts to a more general setting that includes alongside graph adjacency node features and labels in data generation. Also, their analysis focuses on three specific types of graphs whose distribution shifts are determined by specific edge distributions. These specific graph types are special cases of our framework, where one can flexibly instantiate the pairwise function $h$ in our data-generating mechanism. In this regard, our analysis targets a more general and difficult setting that better represents complex real data.
>
> ***Comparison with [4]*** Our work largely differs from [4] in terms of distinct research problems: [4] studies the robustness of graph diffusion models under adversarial attack, while we focus on the topological generalization (see recent works [6,7] highlighting the differences between the two problems). In specific, we have unique contributions on the generalization analysis (e.g. Prop 3, Thm 2). Furthermore, [4] limits the analysis in local graph diffusion, while we also discuss non-local diffusion (Sec. 3.3) as well as the advective diffusion equation (Sec. 4.2) in our framework. The results of [4] lead to a new approach for improving the robustness of GNNs under attack on topology, while our results serve to motivate our new Transformer-like model from the PDE solution.
>
> [3] Graphon neural networks and the transferability of graph neural networks in NeurIPS 2020
>
> [4] On the robustness of graph neural diffusion to topology perturbations in NeurIPS 2022
>
> [5] Transferability properties of graph neural networks in IEEE Transactions on Signal Processing.
>
> [6] David Stutz, Matthias Hein, Bernt Schiele. Disentangling Adversarial Robustness and Generalization, CVPR 2019.
>
> [7] Kimin Lee, Kibok Lee, Honglak Lee, Jinwoo Shin. A Simple Unified Framework for Detecting Out-of-Distribution Samples and Adversarial Attacks, NeurIPS 2018.

---

> ### Author Response · Authors · 2023-11-17
> **Response to Reviewer SG1T (Part 2/2)**
>
> >***Q3 & W3: In the proof of Proposition 1, it is stated that $A$ and $\Delta A$ share the same eigenspace. Why is this true?***
>
> In our analysis for topological generalization, we are mainly interested in the extrapolation behavior of the model w.r.t. the small variation of $\tilde{\mathbf A}$. In this regard, throughout our analysis, we assume a small perturbation $\Delta \tilde{\mathbf A}$, i.e., $||\Delta \tilde{\mathbf A}||_2 \rightarrow 0$.
>
> This allows us to analyze the quantity $\lim_{||\Delta \tilde{\mathbf A}||_2 \rightarrow 0} \frac{||\mathbf Z(T; \tilde{\mathbf A}') - \mathbf Z(T; \tilde{\mathbf A})||_2}{||\Delta \tilde{\mathbf A}||_2}$ that measures the change rate of node representations w.r.t. the tiny change on $\tilde{\mathbf A}$. In this regime, we have the property that $\tilde{\mathbf A}$ and $\tilde{\mathbf A}'$ share the same eigenspace and are commutative. This gives rise to $e^{\tilde{\mathbf A} \tilde{\mathbf A}'} = e^{\tilde{\mathbf A}' \tilde{\mathbf A}} = e^{\tilde{\mathbf A}}\cdot e^{\tilde{\mathbf A}'}$ used in our proof. We add more illustration on this assumption at the beginning of Sec. 3.2.
>
> >***Q4: How does the proposed model perform under heterophily datasets?***
>
> The Twitch dataset we used is essentially a heterophily graph (see [8] for more info on this dataset), which guarantees the diversity of our experimental datasets and shows the competitiveness of our model on heterophily datasets. Having said that the main focus of our paper is topological generalization where training and test graphs come from different distributions, which distinguishes our work from the line of research focusing on heterophily graphs.
>
> [8] New Benchmarks for Learning on Non-Homophilous Graphs, WWW 2021.
>
> >***W2: The given bounds in Proposition 1 and 2 are very loose. I am unconvinced by the authors' motivations for this work***
>
> The Prop 1 and 2 in our paper show that the extrapolation behavior of local diffusion has an exponential upper bound and could lead to undesired significant change on the node representations. The Big-O bounds do not necessarily mean the exponential change rate would definitely happen in practice, but at least suggest the model could be sensisitive to the topological shifts. In contrast, Thm 1 shows that the Big-O bound of advective diffusion can be reduced to arbitrary polynomial orders, which suggests the non-local model is insensisitive to the topological shifts. Therefore, these results together show that non-local diffusion with advection is preferable over local diffusion, since the former has guarantee of bounding the change of node representations. This reasonably motivates our model design in Sec. 4 built upon non-local diffusion with advection.
>
> Also please kindly notice that **our work is not a theoretical paper**. While we agree with the reviewer that the Big-Omega bounds can more accurately reflect the behavior than Big-O bounds, to be honest, it will require deriving the lower bound of matrix exponential norm, which still remains an open question. And, since we mainly focus on the methodology side, it is not necessary to show the negative result of local diffusion, which is not our main target neither. Our goal is to design an effective model with principles and groundings. We thus leave exploration on the Big-Omega bounds as future works along the theoretical line.
>
> We thank the reviewer for the valuable feedback and please let us know if you had any further question or feedback.

---

> > ### Comment · Reviewer_SG1T · 2023-11-18
> >
> > 1. Only very specific kinds of perturbations $\Delta \tilde{A}$ will result in the same eigenspace as $\tilde{A}$. It is not sufficient to assume $\|| \Delta \tilde{A} \||_2 \to 0$. You need to discuss why topology distribution generalization is restricted to only this specific scenario. To show **topological generalization**, it is impractical to assume $\|| \Delta \tilde{A} \||_2 \to 0$ or $\Delta \tilde{A}$ being in the same eigenspace as $\tilde{A}$.
> >
> > 1. There are various places in your proofs that make use of $e^{A+B}=e^A e^B$ without proper justification. I do not see a proper explanation for these.
> >
> > 1. For the Big-Omega vs. Big-O issue, in view of theoretical difficulties, one resolution is to remove the inappropriate theoretical motivation and produce numerical results to illustrate that label predications are indeed exponentially sensitive to topology perturbations.
> >
> > 1. A non-theoretical paper with flawed mathematical derivations, proofs and motivation is not acceptable in view of publication record sanctity. A full revision of the current paper to remove inaccurate parts may be required.

---

> > > ### Author Response · Authors · 2023-11-21
> > >
> > > Thanks for your feedback.
> > >
> > > 1/2. The proofs will be fixed in later version by avoiding the commutation of matrix exponential.
> > >
> > > 3. In our paper, we do not claim that the local diffusion model will necessarily suffer from expopential sensitivity given the Big-O bound. The analysis serves to suggest the potential limitation of local diffusion and the promise of non-local diffusion, rather than to show a guarantee of their failure or success. We agree that more numerical analysis can strengthen this work, but we have to point out that the current results along with the follow-up motivations are properly stated in our paper.

---

### Official Review · Reviewer_Byi6 · 2023-10-31

**Soundness:** 3 good
**Presentation:** 3 good
**Contribution:** 2 fair
**Rating:** 3
**Confidence:** 4

**Summary:**

This paper proposed ADiT that is composed of local message passing and global transformer to tackle topological distribution shift between training and test sets.
The sensitivity of local/non-local model results according to structural perturbation was analyzed.
Based on the analysis, a model using both local and global interactions was proposed, and comparative experiments such as classification and regression were performed.

**Strengths:**

1. Potential of transformer as a solution to topological distribution shift.

**Weaknesses:**

1. The motivation of this paper eventually converges to the composition of mpnn and transformer that already proposed before.
2. There is a lack of concept and experimental comparison with related papers.
3. There is a lack of analysis on out of distribution in experiments.

**Questions:**

1. On the design of PDE based graph learning model, what is the relevance and difference between and where is the experimental comparison with [1]?

&nbsp;&nbsp;&nbsp;&nbsp;&nbsp; [1] "ADR-GNN: Advection-Diffusion-Reaction Graph Neural Networks", Arxiv 2023.

2. The proposed method converges to the composition of local and global message passing. In that respect, compared to [2, 3], what is the difference in terms of model design and performance?

&nbsp;&nbsp;&nbsp;&nbsp;&nbsp; [2] "A generalization of vit/mlp-mixer to graphs", ICML 2023.

&nbsp;&nbsp;&nbsp;&nbsp;&nbsp; [3] "GPS++: An optimised hybrid mpnn/transformer for molecular property prediction", Arxiv 2022.

3. In order to learn a topology agnostic representation, one direction is to consider all possible connections, but it is also possible to ignore all connections. It is necessary to check the results of MLP as baseline performance.

4. What is the definition of hypothesis space size in proposition 3?

5. I would like to know clearly what topological shift was induced through splitting for OGB-Bace and -Sider.

6. Why local diffusion models have comparable performances compared to non-local diffusion models in Table 1?

7. It is necessary to compare with relevant non-local (transformer) models [4, 5], and models that considers out-of-distribution [6, 7, 8, 9].

&nbsp;&nbsp;&nbsp;&nbsp;&nbsp; [4] "EXPHORMER: Sparse Transformers for Graphs", ICML 2023.

&nbsp;&nbsp;&nbsp;&nbsp;&nbsp; [5] "GOAT: A Global Transformer on Large-scale Graphs", ICML 2023.

&nbsp;&nbsp;&nbsp;&nbsp;&nbsp; [6] "Size-invariant graph representations for graph classification extrapolations", ICML 2021.

&nbsp;&nbsp;&nbsp;&nbsp;&nbsp; [7]  "Sizeshiftreg: a regularization method for improving size-generalization in graph neural networks", NeurIPS 2022.

&nbsp;&nbsp;&nbsp;&nbsp;&nbsp;  [8] "From local structures to size generalization in graph neural networks", ICML 2021.

&nbsp;&nbsp;&nbsp;&nbsp;&nbsp;  [9] "Learning substructure invariance for out-of-distribution molecular representations", NeurIPS 2022.

8. Computational time comparison between (non-) local diffusion models is required.

9. Table 9 in the appendix shows that local interactions are more important than non-local interactions. These results may seem to contradict the logic of this paper, which advocates learning topologically agnostic representations to solve topological OOD.

**Details Of Ethics Concerns:**

None.

---

> ### Author Response · Authors · 2023-11-17
> **Response to Reviewer Byi6 (Part 1/3)**
>
> Thank you for the constructive feedback and suggestions for improvement. We address your questions and comments in detail below.
>
> >***Q1 & W2: Comparison with [1]***
>
> First, we would like to note that [1] was posted on ArXiv two months ago and should be considered concurrent work. While both our paper and [1] consider PDE-based graph learning models, **our work is clearly distinct from [1] in terms of the target problem settings and model designs.** First, [1] focuses on the standard graph learning problem where training and test data are assumed to be drawn from the same distribution, while our work focuses on topological generalization where the main goal is to extrapolate to out-of-distribution (OOD) test data with new topologies. The different problem settings lead to different technical challenges and model designs: [1] instantiates the diffusion term as local message passing to improve the expressivity of fitting training data; in contrast, our work considers diffusion as a global attention for better generalization to OOD test data. The implementation of the two models also differs substantially: ADR-GNN follows prior art using finite-difference numerical iterations for solving the PDE model, while our work harnesses the Padé-Chebyshev theory for computing the closed-form PDE solution. Besides the different problem setting, our experiments cover a broader range of tasks than [1]. (See the comparison below).
>
> | Aspect | ADR-GNN | Our work |
> | -------- | -------- | -------- |
> | Problem Setting     | standard graph learning | topological generalization |
> | Diffusion Instantiation | local message passing | global attention |
> | Implementation | finite-difference iteration for solving PDE | Padé-Chebyshev theory for computing closed-form solution |
> | Experiment Scenario | node-level task | node-level, edge-level, graph-level tasks |
>
> That being said, we added new experiments to compare ADiT with ADR-GNN on node classification. Since no code was provided by the authors, we implemented their model according to the description in their paper and obtain the results below, which shows the superiority of our approach ADiT over ADR-GNN:
>
> | Dataset | Arxiv (2018) | Arxiv (2019) | Arxiv (2020) | Twitch (avg) |
> | - | - | - | - | - |
> | ADiT-Series     | 53.41 ± 0.48 | 51.53 ± 0.60 | 49.64 ± 0.54 | 62.51 ± 0.07 |
> | ADR-GNN     | 50.12 ± 0.23 | 47.31 ± 0.59 | 45.14 ± 0.31 | 60.87 ± 0.12 |
>
> [1] "ADR-GNN: Advection-Diffusion-Reaction Graph Neural Networks", Arxiv 2023
>
> >***Q2 & W1/W2: Comparison with other models using local/global message passing [2, 3]***
>
> To be honest, there are a number of recent works considering the hybrid architecture of local and global message passing in the model (e.g., GraphTrans, GraphGPS and DIFFormer that have been compared in our experiments). **These models including [2, 3] suggested by the reviewer have fundamental differences in the problem settings and model designs**. First, similar to [1], they all focus on the standard graph learning problem, wherein the main target aims to improve the model's expressivity instead of the generalization capability (our focus). Second, from the architectural view, these models are composed of discrete feed-forward layers, which is largely different from our model built upon continuous PDE and its closed-form solution.
>
> In the weakness section, the reviewer recognizes the motivation of "the composition of mpnn and transformer" shared by these models and ours as limited originality, but we argue that **given the similar high-level motivation, it is still important and under-determined how to design the effective models for different tasks**. The latter, which actually remains an open question, can lead to different model designs and architectures competent in distinct tasks.
>
> To better justify our contributions, we present a comparison with these models that consider local/global message passing in the table below.
>
> | Aspect | How Local/Global Message Passing is Integrated | Main Target Task | Publication Year | Have Compared in Our Experiments
> | - | - | - | - | - |
> |   GraphTrans   | stacking MPNN and Transformer | standard graph-level task  | 2021 | Yes |
> |   NodeFormer   | Transformer + edge regularization loss | standard node-level task  |  2022 | No |
> |   GraphGPS   | add MPNN and attention at each layer | standard graph-level task |  2022 | Yes |
> |   GPS++ [2]   | add MPNN and attention at each layer | standard graph-level task |  2022 | No |
> |   Graph ViT/MLP-Mixer [3]   | position embedding  | standard graph-level task |  2023 | No |
> |   DIFFormer   | add MPNN and attention at each layer | standard node-level task  |  2023 | Yes |
> | ADiT (ours) | solution of advective diffusion equation | topological generalization in node-level, edge-level, graph-level tasks | 2023 | - |
>
> [2] "A generalization of vit/mlp-mixer to graphs", Arxiv 2023.
>
> [3] "GPS++: An optimised hybrid mpnn/transformer for molecular property prediction", Arxiv 2022.

---

> ### Author Response · Authors · 2023-11-17
> **Response to Reviewer Byi6 (Part 2/3)**
>
> >***Q3: Check the results of MLP as baseline performance***
>
> Thank you for this constructive suggestion, and we supplement new experiments of MLP on all the datasets (see results in Table 1, 2, 3 in the revised paper). It demonstrates the superiority of our model over MLP which does not leverage any inter-dependence among nodes and yields inferior generalization. These results can strengthen our contributions.
>
> >***Q4: What is the definition of hypothesis space size***
>
> In general machine learning context, the hypothesis space refers to the set of all possible hypotheses that a learning algorithm can consider or output. In our case, the hypothesis space particularly refers to the set of candidate functions induced by $\Gamma_\theta$ as approximated mapping from $\mathbf A$, $\mathbf X$ to $\mathbf Y$.
>
> >***Q5 & W3: What topological shift was induced through splitting for OGB-Bace and -Sider***
>
> The public splits of these datasets provided by the OGB team are based on the molecular scaffolds. The latter refers to a pre-defined set of molecular substructures. In this regard, as illustrated in Appendix F.1.3 in our original paper, the molecules in the training and test sets have different scaffolds, introducing the topological distribution shifts.
>
> >***Q6: Why local diffusion models have comparable performances compared to non-local diffusion models in Table 1?***
>
> The real-world datasets are much more complex than the hypothetical regimes used for theoretical analysis. Particularly, given the node-level inter-dependence in these datasets, it is challenging for the model to learn the generalizable topological features that are complex and abstract. Also, the structural information of observed graphs is important for prediction, so the non-local models also need to make good use of the input graphs for desired generalization. The improvements of ADiT over other non-local models (DIFFormer and GraphGPS) suggest that our model can better exploit both the observed structural information and unobserved global interactions for generalization.
>
> >***Q7 & W2: It is necessary to compare with relevant non-local (transformer) models [4, 5], and models that considers out-of-distribution [6,7,8,9]***
>
> While we agree that it's always better to compare with more competitors, we'd like to point out that **[4-9] focus on different problem settings or are orthogonal to our approach**, and in these regards, we did not compare with them in our original experiments.
>
> For the non-local (transformer) models, we have compared with three state-of-the-art graph Transformers (GraphTrans, GraphGPS and DIFFormer) in our experiments. [4, 5] focus on standard graph learning instead of topological generalization, and have different research focus. Even so, we add new comparison with them in node classification datasets.
>
> | Dataset | Arxiv (2018) | Arxiv (2019) | Arxiv (2020) | Twitch (avg) |
> | -------- | -------- | -------- | -------- | -------- |
> | ADiT-Series     | 53.41 ± 0.48 | 51.53 ± 0.60 | 49.64 ± 0.54 | 62.51 ± 0.07 |
> | EXPHORMER     | 51.03 ± 0.31 | 48.18 ± 0.43 | 46.31 ± 0.27 | 61.42 ± 0.17 |
> | GOAT |  52.12 ± 0.11 | 49.41 ± 0.23 | 47.21 ± 0.12 | 61.23 ± 0.06 |
>
> The OOD methods [6,7,8] focus on size generalization where distribution shifts are purely caused by change of graph sizes, which is different from topological generalization. Also, these works are technically orthogonal to our model, as they focus on designing regularization approaches or learning algorithms that are agnostic to the model backbone. In contrast, our model is a new encoder backbone, in parallel to other GNN models. This is why our original comparison focuses on the peer models that can equally serve as the encoder backbones. That being said, we add new comparison with SizeShiftReg [7] and MoleOOD [9] on the molecular datasets, using GCN as the backbone that is used by their original papers. The table below shows the testing ROC-AUC and demonstrates the competitiveness of ADiT.
>
> | Dataset | OGB-BACE | OGB-SIDER |
> | -------- | -------- | -------- |
> | ADiT-Inverse     | 80.38 ± 1.40 | 65.29 ± 0.16 |
> | ADiT-Series | 82.03 ± 0.42 | 62.28 ± 0.36 |
> | SizeShiftReg | 79.19 ± 0.62 | 59.61 ± 0.28     |
> | MoleOOD | 79.62 ± 0.31 | 60.64 ± 0.54     |
>
> >***Q8: Time comparison***
>
> Thanks for the nice suggestion that can help to strengthen our paper. We supplement new results of training/inference time per epoch of all the models on the DDPIN dataset.
>
> | - | training time per epoch (s) | inference time per epoch (s) |
> | -- | -------- | -------- |
> | GCN | 0.44 | 0.30 |
> | GAT | 0.47 | 0.38 |
> | SGC | 0.42 | 0.31 |
> | GraphTrans | OOM | OOM |
> | GraphGPS | 0.84 | 0.67 |
> | DIFFormer | 0.96 | 0.63 |
> | ADiT-Series | 0.94 | 0.76 |
> | ADiT-Inverse | 1.55 | 1.29 |
>
> The results show that our models, in particular ADiT-Series, is as efficient as other Transformer-based models (GraphGPS and DIFFormer). This suggests our model can achieve better performance with the cost of comparable time.

---

> ### Author Response · Authors · 2023-11-17
> **Response to Reviewer Byi6 (Part 3/3)**
>
> >***Q9: Why local interactions are more important than non-local interactions in Table 9?***
>
> This is because in these node classification datasets, the observed graph contains much useful information and is important for generalization. In our paper, we **did not** state that non-local interactions are necessarily more important than local interactions, and actually, this may depend on specific datasets in practice. Instead, our main point is that learning non-local interactions can facilitate generalization (and also notice that the local interactions can be useful as well), which is validated by the improvement of ADiT over ADiT w/o diffusion in Table 9.
>
> Thank you again for reviewing our manuscript, and please let us know if you had any further question or feedback.

---

### Author Response · Authors · 2023-11-17
**General Response to All Reviewers**

We are grateful to the reviewers for the time dedicated to reviewing our paper. We are encouraged to see the reviewer's acknowledgement on the significant research problem we explore and the comprehensive experiments encompassing diverse tasks.

The reviewers have divergent opinions on the novelty of our method, with specific questions on the differences compared to several lines of existing works, which we would like to clarify below and in individual responses:

***1. Our work targets the open challenge of topological generalization, which is an under-explored research question largely ignored by prior works in graph learning.***

> We tackle distinct technical challenges stemming from the topological distribution shifts, which differ from the research focus of **existing graph diffusion models (GRAND, GRAND++, [1] suggested by Reviewer Byi6, [2,3] suggested by Reviewer SG1T and [1-6] suggested by Reviewer zCh7)** that tacitly assume training and test data come from the same distribution and aim to improve the model's expressivity without analyzing its generalization. Also, topological generalization is largely different from the adversarial robustness problem studied in [7] suggested by the reviewers.

***2. We leverage the mathematical framework of graph diffusion equations as a principled perspective for analyzing the model's extrapolation/generalization behaviors under topological shifts.***

> This new perspective and analysis on continuous models distinguish our work from **existing works on topological generalization (e.g., [8, 9] suggested by Reviewer SG1T and zCh7)** based on discrete message passing layers and specific types of graph data.

***3. In light of our analysis, we introduce a new Transformer-like encoder backbone built upon the closed-form solution of diffusion equations for topological generalization.***

> From the architectural view, the novelty of our model is two-fold: 1) compared with **existing graph Transformers (e.g., GraphTrans, GraphGPS and DIFFormer that we have compared and [11-14] suggested by Reviewer Byi6)**, our model is built on a continuous PDE; 2) compared with **existing graph diffusion models (e.g., [1-7])** that resort to local message passing, our model extends to non-local diffusion with global all-pair attentions. Besides, instead of using the differentiable ODE/PDE solver [8] adopted by prior art, our model implementation is derived from computing the closed-form PDE solution by the Padé-Chebyshev expansion, which enables good scalability and stability.

***4. We demonstrate the wide applicability and potential of the model in comprehensive experiments encompassing a broad spectrum of tasks.***

> In the experimental evaluation, our main goal is to showcase the competitiveness of our model as a general backbone in diverse graph learning tasks with the focus on generalization, instead of competing against the task-specific SOTA models that target distinct problems and have specific applicability.

[1] ADR-GNN: Advection-Diffusion-Reaction Graph Neural Networks, Arxiv 2023

[2] Graph neural convection-diffusion with heterophily, IJCAI 2023

[3] ACMP: Allen-cahn message passing with attractive and repulsive forces for graph neural networks, ICLR 2023

[4] GREAD: Graph Neural Reaction-Diffusion Equations, Arxiv 2022.

[5] Graph-coupled oscillator networks, ICML 2022

[6] Anti-symmetric dgn: A stable architecture for deep graph networks, ICLR 2023.

[7] On the robustness of graph neural diffusion to topology perturbations in NeurIPS 2022

[8] Graphon neural networks and the transferability of graph neural networks in NeurIPS 2020

[9] Transferability properties of graph neural networks in IEEE Transactions on Signal Processing 2023.

[10] Neural ordinary differential equations, NeurIPS 2018

[11] A generalization of vit/mlp-mixer to graphs, ICML 2023.

[12] GPS++: An optimised hybrid mpnn/transformer for molecular property prediction, Arxiv 2022

[13] EXPHORMER: Sparse Transformers for Graphs, ICML 2023.

[14] GOAT: A Global Transformer on Large-scale Graphs, ICML 2023.

According to the feedback and suggestions, we update the paper PDF with the modified parts colored blue. We also provide multiple new experiments to answer the questions raised by the reviewers. We hope our responses address the raised concerns and would appreciate the reviewers raising their scores. We will be glad to answer any further questions.

---

### Meta-Review · Area_Chair_EhFo · 2023-12-05

**Metareview:**

The paper introduces the Advective Diffusion Transformer (ADiT), a model combining local message passing and a global transformer, designed to address the challenge of topological distribution shifts in graph neural networks (GNNs). It critically evaluates the sensitivity of existing models to structural perturbations and proposes a hybrid approach that balances local and global interactions, validated through classification and regression experiments.

While the proposed ADiT model shows promising results, the reviewers have identified several weaknesses that need to be addressed:

1. The motivation of the paper is unclear.
2. The correctness of several proofs, including those for Propositions 1 and 2, is questionable.
3. The paper overlooks many related works.

Based on these weaknesses, we recommend rejecting this paper. We hope this feedback helps the authors improve their paper.

**Justification For Why Not Higher Score:**

The reviewers unanimously believe the paper should be rejected.

**Justification For Why Not Lower Score:**

N/A

---

### Decision · Program_Chairs · 2024-01-16

Reject